# Naturality-Guided Hyperedge Disentanglement for Message Passing Hypergraph Neural Network

## Abstract

Hypergraph data structure has been widely used to store information or meaning derived from group interactions, meaning that each hyperedge inherently contains the context of their interactions. For example, a set of genes or a genetic pathway can be represented as a hyperedge to express the interaction of multiple genes that collaboratively perform a biological function (i.e., interaction context). However, most existing hypergraph neural networks cannot reflect the interaction context of each hyperedge due to their limited capability in capturing important or relevant factors therein. In this paper, we propose a **simple but effective** hyperedge disentangling method, **Natural-HNN**, that captures inherent hyperedge types or the interaction context of an hyperedge. We devised a novel guidance for hyperedge disentanglement based on the naturality condition in the category theory. In our experiments, we applied our model to hypergraphs of genetic pathways for the cancer subtype classification task, and showed that our model outperforms baselines by capturing the functional semantic similarity of genetic pathways. Our implementation is available at https://anonymous.4open.science/r/Natural-HNN-E264.

## 1 Introduction

Recently, several Hypergraph Neural Networks (HNNs) (Feng et al., 2019; Chien et al., 2021) have been devised to integrate the complex interactions within hypergraphs, driven by the increasing demand resulting from the prevalence of multiway interactions in reality. In electric circuits (Cockett et al., 2023; Wang et al., 2022a), for instance, many circuit components are connected in parallel, naturally causing multiway interactions. In biology (Nguyen et al., 2022), most biological processes are the result of complex interactions. Specifically, a genetic pathway is a set of genes that collaborate to perform a specific function in a biological process. In other words, as pathways represent functional relations among genes participating in interactions, it is natural to express these interactions as a hypergraph. Application domains of HNNs are progressively expanding to natural language processing, chemistry and recommender systems.

A noteworthy characteristic of a hypergraph is that each hyperedge may contain different interaction contexts. In opinion dynamics (Neuhäuser et al., 2021; Hickok et al., 2022), which is an area exploring how opinions of individuals develop over time in a social network, group discussions can be represented as hyperedges. More precisely, each individual (i.e., node) can participate in different group discussions (i.e., hyperedges) that have their own discussion topics, which we call the *interaction context*, such as social issues or economic policy. When information about context is explicitly available during data collection, it can be expressed as hyperedge types in a heterogeneous hypergraph and can be reflected in message passing by relational HNNs or heterogeneous HNNs. When information about context is not accessible, however, the context information is lost and data is expressed as a homogeneous hypergraph. Thus, capturing interaction context (or inherent hyperedge types) during message passing is needed.

Genetic pathway is an example of a biological network that lacks annotations (Liu & Thomas, 2019) for interaction context (i.e., the function of a pathway or condition such as cell types or tissues). Since genes exhibit different characteristics (i.e., gene expression levels or gene function) depending on the context (Chen et al., 2021), it is important to reflect the context of pathways. For example, FOXO1 in the insulin signaling pathway at hepatocytes (cell type) can activate gluconeogenesis in

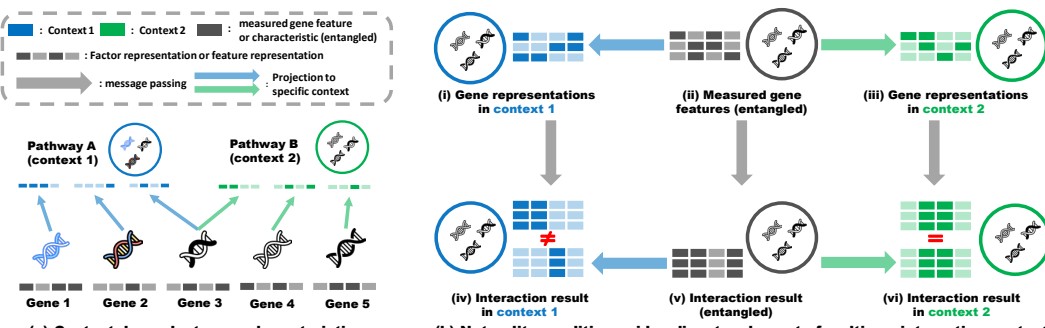

Figure 1: (a) Genes can exhibit different characteristics or gene expression levels depending on the context (Gene 3). Also, genes interacting under the same context can exhibit different charactersitics or gene expression levels (pathway B). (b) Naturality condition (commutativity) guides interaction context disentanglement.

liver (tissue) (Puigserver et al., 2003). On the other hand, FOXO1 acts as an important regulator in FOXO1/lysosome/MVB/GSK3$\beta$ pathway which is related to the maintenance of proteostasis and the control of effector T cell (cell type) differentiation (Jin et al., 2020). This highlights the importance of capturing interaction context during message passing.

However, most HNNs cannot leverage interaction contexts properly. Convolution-based methods cannot perform interaction context-dependent message passing as nodes always propagate the same message to their neighboring hyperedges. Although attention-based methods propagate hyperedge dependent messages by differentiating the importance of neighbors with respect to each factor among multiple factors of the interaction, they cannot determine which factor is more relevant to the interaction context of each hyperedge. Most recently, a sheaf-based method (Duta et al., 2024) that learns a different restriction map (or a transformation matrix) for every (node, hyperedge) pair has been proposed. While the design of sheaf-based methods allows the model to capture the interaction context of a hyperedge by learning a different transformation for every (node, hyperedge) pair, they will hardly do so as there is no guidance that helps the transformation to be related to interaction context.

In this paper, we focus on the fact that establishing a criterion for disentangling the factors of an interaction (e.g., identifying pathway function or condition) is challenging. Existing studies (Ma et al., 2019; Hu et al., 2022) extract information regarding the factors of an interaction by simply adopting factor-specific MLPs to the (entangled) node representation. Then, they consider a factor to be relevant to the interaction context of a hyperedge when a set of nodes have similar factor representations, under the assumption that nodes interact with each other due to their commonality. However, the factor similarity is not always related to the similarity in the interaction context of a hyperedge in reality. In genetic pathway example, as can be seen in the Figure 1 (a), genes participating under the same context can have different characteristics such as different gene expression levels[1]. As a result, the similarity-based criterion might not be effective in capturing the function or condition of pathway, and would hinder a model in effectively integrating contextual information from participants. Hence, a new angle of approach for guiding disentanglement is required to integrate context in hypergraph message passing framework.

To this end, we propose a novel Naturality-guided disentangled Hypergraph Neural Network (**Natural-HNN**) that can inherently reflect the interaction context of an hyperedge. We approach the task with the category theoretical perspective (Fong & Spivak, 2018), and determine the criterion for disentangling factors as the factor representation consistency based on the naturality condition that must be satisfied between entangled and disentangled representations. Figure 1 (b) shows the naturality condition applied to our genetic pathway example. Let's suppose that genes in a pathway interacts under the context 2 and does not interact under context 1. The result of interaction under context 2 must be consistent, regardless of whether interaction was performed only on context 2 (i.e., factor specific message passing, Figure 1 (b) $(ii) \rightarrow (iii) \rightarrow (vi)$) or the interaction was performed for both contexts but only context 2 related result was selected (i.e., factor information extraction after entangled message passing, Figure 1 (b) $(ii) \rightarrow (v) \rightarrow (vi)$). On the other hand, this commutativity does not hold for context 1 (i.e., the result of $(ii) \rightarrow (i) \rightarrow (iv)$ and $(ii) \rightarrow (v) \rightarrow (iv)$ is

---

[1]Specific examples can be found in (Harris & Levine, 2005; Mehdizadeh et al., 2023)

different) as the pathway is not related to context 1. The adoption of consistency constraint derived from category theory allows us to capture context related factors without relying on any assumption on the data. Given that our model can potentially capture inherent heterogeneity of interactions (i.e., various contexts of interactions, Appendix H.2) within homogeneous data, it offers a practical solution to many real-world problems where the types of heterogeneous interactions are unknown.

Our main contributions are summarized as follows:

- To the best of our knowledge, we are the first to propose a hyperedge disentanglement-based method that is systematically designed to capture the context related to the background or condition of multiway interaction .

- We proposed a novel way to guide the hyperedge disentanglement, by focusing on the compositional structure of entities in hypergraph message passing framework. Through a new criterion derived from the category theory, we created a simple but effective model, showing outstanding performance even with a small hyperparameter search space.

- We applied our model to the cancer subtype classification task, and showed our model can actually capture functional semantics of pathways (i.e., interaction context of hyperedges). Also, we showed that capturing such context of interaction is critical in real world hypergraph problems.

## 2   RELATED WORK

**Hypergraph Neural Network.**     Several HNN models have been recently proposed to leverage information contained in multiway interaction. HGNN (Feng et al., 2019) and HCHA (Bai et al., 2021) use a normalized hypergraph Laplacian, which is mathematically equivalent to clique expansion (CE) (Sun et al., 2008), and apply the traditional graph convolution mechanism. HNHN (Dong et al., 2020) additionally adopts nonlinearity when calculating hyperedge representations to differentiate a hypergraph from a clique expanded graph, while UniGNN (Huang & Yang, 2021) unifies HNNs and GNNs into the same framework. Moreover, HyperGAT (Ding et al., 2020) adopts the attention mechanism to HNN for text classification, and SHINE (Luo, 2022) proposes dual attention mechanism for the disease classification task. ED-HNN (Wang et al., 2022b) proposes equivariant message passing HNN, which allows hyperedges to propagate different messages to its incident nodes. AllDeepSets and AllSetTransformer (Chien et al., 2021) consider a hyperedge as a set and apply DeepSets (Zaheer et al., 2017) and Set Transformer (Lee et al., 2018), respectively, to increase expressive power of HNN. All of theses methods, however, cannot give different weights to different heads or factors, limiting their capability of capturing the interaction context of an hyperedge, which is crucial in practice. Sheaf Hypergraph Network (Duta et al., 2024) learns a different restriction map or transformation matrix for every (node, hyperedge) pair. Although this approach may enable the model to capture interaction context with all these learnable transformation matrices, it lacks clear guidance for doing so and requires significant computational resources. WHATSNET (Choe et al., 2023) captures the interaction context shaped by the participants (i.e., the context depends on 'who participates the interaction'). For example, if six students and one teacher participates a discussion, it is highly likely that the teacher takes the role of moderator. However, the context that we are trying to capture is more related to the background or condition of interaction (i.e., the topic of discussion), which is different from the context defined by (Choe et al., 2023). Thus, WHATSNET and our paper aims to solve different problems. More details can be found in Appendix H.8.

**Disentangled Representation Learning.**  Disentangled representation learning (DRL) (Roth et al., 2022; Fumero et al., 2021; Higgins et al., 2018) aims to disentangle the factor of variation of observed data. The effectiveness of DRL has garnered attention of researchers, leading to its expansion into the field of GNN. DisenGCN (Ma et al., 2019) disentangles the factor of variations in nodes to find the factor behind connections, while FactorGCN (Yang et al., 2020) disentangles graphs into several factor graphs. DisGNN (Zhao et al., 2022) recently proposes to disentangle edge types with the self-supervision from label conformity.

Since graph-based disentangling methods cannot model multiway interactions, DRL is also being applied to hypergraphs. HSDN (Hu et al., 2022) attempts to capture structural semantics by disentangling a hypergraph into several factor hypergraphs. Although this method is advantageous when capturing the functional structure in molecules or finding communities in a social network, it is not suitable for capturing the interaction context as this approach captures semantics derived from different connectivity or substructure. DisenHCN (Li et al., 2022) disentangles user embeddings for recommender systems, but is only applicable to hypergraphs with known hyperedge types.

# 3 CATEGORICAL INTERPRETATION OF MESSAGE PASSING HNN AND DISENTANGLEMENT

Prior to the discussion of the naturality condition for hyperedge disentanglement, it is essential to analyze the compositional structure in the hypergraph representation learning. In Section 3.1, we describe the compositional structure of hypergraph message passing neural networks. In Section 3.2, we propose the naturality condition as a guidance for hyperedge disentanglement. The basic concepts in category theory we used are described in Appendix G, and the basic explanation of disentangled representation learning is described in Appendix H.1.

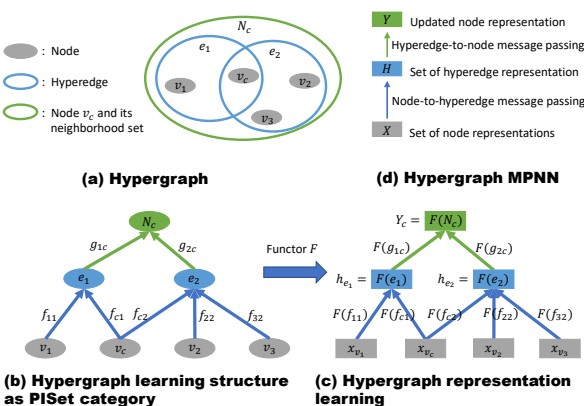

(a) Hypergraph      (d) Hypergraph MPNN

(b) Hypergraph learning structure as PISet category    (c) Hypergraph representation learning

Figure 2: Compositional structure in hypergraph representation learning.

**Notation.** Let $\mathcal{G} = (\mathcal{V}, \mathcal{E})$ denote a hypergraph, where $\mathcal{V} = \{v_1, v_2, ..., v_N\}$ indicates a set of nodes and $\mathcal{E} = \{e_1, e_2, ..., e_M\}$ indicates a set of hyperedges, where $N = |\mathcal{V}|$ and $M = |\mathcal{E}|$ are the number of nodes and the number of hyperedges in a hypergraph $\mathcal{G}$, respectively. A set of node features given as input to each layer of the model is denoted as $X = \{x_{v_1}, ..., x_{v_N}\}$, a set of hyperedge representations (calculated in each layer of the model) is denoted as $H = \{h_{e_1}, ..., h_{e_M}\}$, and a set of representations obtained after message passsing is denoted as $Y = \{y_{v_1}, ..., y_{v_N}\}$. '*en*' denotes an entangled object or morphism and is written in superscript or subscript, while '*dis*' denotes a disentangled object or morphism. The symbol '⨟' is used to denote the composition of morphisms.[2]

## 3.1 COMPOSITIONALITY IN HYPERGRAPH REPRESENTATION LEARNING

Most hypergraph representation learning methods produce the representation of a node by integrating its own representation and its neighbors' representations defined by a hypergraph topology. As an example, in Figure 2 (a), the representation of a center node $v_c$ is updated to the representation that can express the meaning produced by a set of nodes $N_c$, the set whose elements are the node $v_c$ and its one-hop neighbors ($v_1, v_2, v_3$). During the process, the hypergraph topology created by hyperedges are considered.

In this paper, for the first time, we describe the above process of hypergraph representation learning through the lens of the category theory. Specifically, if we consider each node as a set, since a hyperedge contains nodes, there are morphisms (inclusion) between nodes and hyperedges induced by the poset structure. We defined this as **PISet**, the category with **p**oset structure where morphisms are **i**nclusions and objects are **set**s. Thus, we can see nodes ($v_1, v_c, v_2, v_3$) and hyperedges ($e_1, e_2$) constitute **PISet** as shown in Figure 2 (b), where gray-colored nodes and hyperedges are set objects, and inclusions are morphisms (blue arrow) between sets. The same mechanism holds between hyperedges ($e_1, e_2$) and a set $N_c$ that includes node $v_c$ and its neighbors. In Figure 2 (b), for instance, we can see hyperedges ($e_1, e_2$) and $N_c$ constitute **PISet** as they have morphisms (green arrow) induced by the poset structure.

In order to learn and predict with computers, such objects and morphisms must be expressed in numerical values and their transformations. Hence, we define a category of deep learning representations, **DLRep**, where objects are vector representations and morphisms are transformations between them. Figure 2 (c) shows the result of applying a functor $F : \textbf{PISet} \rightarrow \textbf{DLRep}$, which can be simplified to a diagram in Figure 2 (d). Thus, any kind of hypergraph message passing neural networks[3] can be seen as a way of learning representations and their transformations respecting compositional structure of entities.

---

[2]Two notations $f ⨟ g$ and $g \circ f$ have the same meaning : "applying $f$ first, and then applying $g$". We use the notation '⨟' following (Fong & Spivak, 2018).

[3]The message passing types are not only limited to traditional convolution-based or attention-based methods, but also can include complex methods such as general message passing (Papillon et al., 2023).

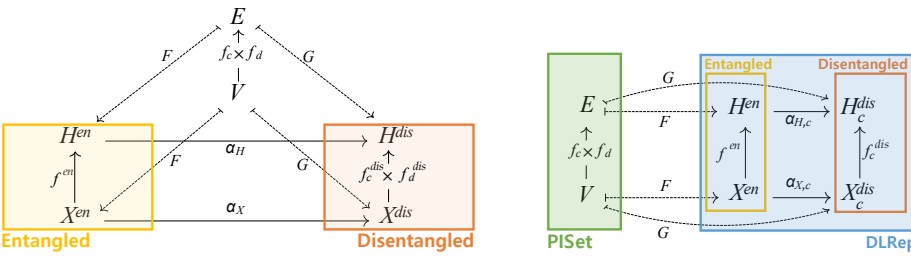

(a) Natural Transformation in HNN        (b) Natural Transformation in HNN, factor perspective

Figure 3: Naturality condition in disentangled representation learning to capture context related factors. $X$ denotes a set of node representations and $H$ denotes hyperedge representation. $V$ and $E$ denote node and hyperedge in **PISet**, respectively.

The most expressive way for a model to accommodate various morphisms would be to assign different learnable parameters to every morphism, which, however, would likely fail in generalizability and scalability perspectives. In this case, providing proper inductive bias is the key to balancing the trade-off between expressive power and generalizability of the model. However, convolution-based methods have a strong assumption that all neighbors can be considered equally regardless of the interaction context of an hyperedge, limiting expressive power of the model. On the other hand, disentangled representation learning can be used as an adequate trade-off by categorizing morphisms into a small number of morphism types, which can be considered as context-dependent message passing. Therefore, we propose a hyperedge disentangling method for context-dependent message passing, which will be introduced in Section 4.

### 3.2 GUIDING DISENTANGLEMENT WITH NATURALITY CONDITION

Since entangled representations and disentangled representations are different ways of representing the same compositional structure, we can regard them as the result of applying two different functors $F$ : **PISet** → **DLRep** (for entangled representations) and $G$ : **PISet** → **DLRep** (for disentangled representations) as shown in Figure 3 (a). Thus, we have the naturality condition between entangled representations and disentangled representations. Figure 3 (b) is equivalent to Figure 3 (a), but only the components related to the factor '$c$' are shown. Note that $\alpha_{X,c} = \alpha_X \mathbin{\mathring{\circ}} p_c$ where $p_c$ : $X^{dis} \to X_c^{dis}$ (refer to Appendix H.3). If factor '$c$' is relevant to the morphism between node set $V$ and hyperedge $E$, the naturality condition must hold for the perspective of factor '$c$'. Thus, factor '$c$' representation of a hyperedge (i.e., $H_c^{dis}$) must be the same (or similar) regardless of applying $f^{en} \mathbin{\mathring{\circ}} \alpha_{H,c}$ (i.e., message passing on entangled representation first, and then disentangling factors) or $\alpha_{X,c} \mathbin{\mathring{\circ}} f_c^{dis}$ (i.e., disentangling factors first, and then message passing on disentangled representation). In other words, the factor representation must be consistent regardless of the sequence of operations if that factor is relevant to the interaction context of an hyperedge[4]. We use this property as a guidance for disentanglement, since it must hold for any kind of hypergraph message passing neural networks, and must work regardless of data characteristics. More precise and detailed explanations are provided in Appendix H.3

## 4 PROPOSED METHOD: NATURAL-HNN

Each layer of Natural-HNN is composed of a message passing lane (left column of Figure 4 (c)), and a non-message passing lane (right column of Figure 4 (c)) as well as their integration with layer normalization (Section 4.3, bottom of Figure 4 (c)). The key component of our model is the message passing lane (Figure 4 (b)) that consists of a Node-to-Hyperedge factor propagation module (Section 4.1), and a Hyperedge-to-Node factor propagation module (Section 4.2). Note that each layer of Natural-HNN has $K$ factors where $K$ is a hyperparamter.

### 4.1 NODE-TO-HYPEREDGE FACTOR PROPAGATION

**Obtaining Two Disentangled Hyperedge Representations.** To validate whether the naturality condition (Figure 4 (a)) holds, we need to get two disentangled hyperedge factor representations

---

[4]The group discussion example in Figure 1 shows this property.

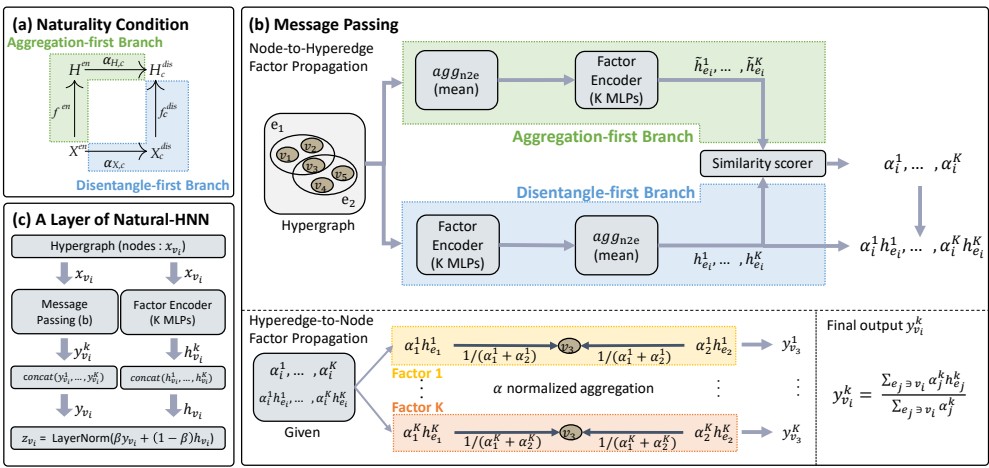

Figure 4: An overview of Natural-HNN. (a) illustrates the naturality condition shown in Figure 3 (b). (b) shows the message passing block of Natural-HNN that consists of a Node-to-Hyperedge and Hyperedge-to-Node factor propagation modules. The Final output of the message passing block is shown at the right bottom corner of (b). (c) shows the composition of each layer of Natural-HNN.

for every factor (i.e., $H_k^{dis}$ for every factor $k \in [1, K]$). The two disentangled representations are obtained through 1) Aggregation-first Branch and 2) Disentalgle-first Branch. In the following, we describe how morphisms in Figure 4 (a) are implemented as operations in the two branches shown in Figure 4 (b).

- **Aggregation-first Branch.** The first disentangled representation is obtained from the aggregation-first branch performing $f^{en} \mathbin{\S} \alpha_{H,k}$ for each factor $k$. This process is implemented as performing aggregation $agg_{n2e}$ (i.e., $f^{en}$ in Figure 4 (a)) first, and then disentangling into hyperedge factor representations using a factor encoder $\alpha_{H,k}$. The factor representations of hyperedge $e_i$ obtained from this branch are denoted as $\tilde{h}_{e_i}^1, \ldots, \tilde{h}_{e_i}^K$.

- **Disentangle-first Branch.** The other one is obtained from the disentangle-first branch performing $\alpha_{X,k} \mathbin{\S} f_k^{dis}$ for each factor $k$. This process is implemented as disentangling into node factor representations with factor encoder $\alpha_{X,k}$ first, and then performing aggregation $agg_{n2e}$ (i.e., $f_c^{dis}$ in Figure 4 (a)). The factor representations of hyperedge $e_i$ obtained from this branch are denoted as $h_{e_i}^1, \ldots, h_{e_i}^K$.

For both branches, we used mean aggregation as $agg_{n2e}$ and $K$ MLPs as factor encoders for disentangling factors. Factor representations are vectors with size $d/K$ (i.e., $h_{e_i}^k, \tilde{h}_{e_i}^k \in \mathbb{R}^{\frac{d}{K}}$), when the desired size for node representations after message passing is $d$. In summary, operations of the two branches regarding factor $k$ can be written as follows:

$$\tilde{h}_{e_j}^k = \text{MLP}_k(\text{mean}(\{x_{v_i}|v_i \in e_j\})), \quad h_{e_j}^k = \text{mean}(\{\text{MLP}_k(x_{v_i})|v_i \in e_j\}) \qquad (1)$$

**Deciding Factors with Consistency.** The extent to which the naturality condition is satisfied can be measured by calculating the similarity between the two disentangled hyperedge factor representations $\tilde{h}_{e_j}^k$ and $h_{e_j}^k$. In other words, we can consider that the naturality condition holds when the two representations are similar (i.e., consistent), and does not hold when the two representations are largely different. We introduce a similarity scorer that calculates the similarity of two $L_2$-normalized vectors. Specifically, we calcualte the relevance or importance of factor $k$ for a hyperedge $e_i$ as $\alpha_i^k = \sigma\left(\frac{h_{e_i}^k}{\|h_{e_i}^k\|_2} W_k \frac{\tilde{h}_{e_i}^{k^T}}{\|\tilde{h}_{e_i}^k\|_2}\right)$, where $W_k \in \mathbb{R}^{\frac{d}{K} \times \frac{d}{K}}$ is a learnable parameter matrix for factor $k$, and $\sigma$ is the sigmoid function. Lastly, we obtain the final hyperedge factor representations by multiplying $\alpha_i^k$ to the corresponding hyperedge factor representations obtained from the disentangle-first branch[5], i.e., $\alpha_i^k h_{e_i}^k$, that reflects the relevance of the factor $k$ for the hyperedge $e_i$.

[5]Although we choose the disentangle-first branch here, we can instead use the output of the aggregation-first branch. Both choices give similar results. Please refer to Appendix E.1.

## 4.2 Hyperedge-to-Node Factor Propagation

When aggregating hyperedge representations (i.e., $\alpha_i^k h_{e_i}^k$) to update node representations, the sum of neighboring hyperedge representations with respect to factor $k$ must be divided by the sum of $\alpha_i^k$ so that hyperedge relevance scores (i.e., $\alpha_i^k$) are normalized during aggregation. Thus, the updated factor $k$ representation of node $v_i$, i.e., $y_{v_i}^k$, can be written as $y_{v_i}^k = \frac{1}{\sum_{e_j \ni v_i} \alpha_j^k} \sum_{e_j \ni v_i} \alpha_j^k h_{e_j}^k$.

## 4.3 Final Output of each Layer of Natural-HNN

We allowed our model to determine its focus between information from neighbors (i.e., $y_{v_i}$) and information of the node itself (i.e., $x_{v_i}$) by introducing hyperparameter $\beta$ that decides interpolation ratio between them. To make sure that interpolation is performed on disentangled representations, we used the factor encoder used in the message passing step (i.e., $h_{v_i}^k = \text{MLP}_k(x_{v_i})$). Specifically, $z_{v_i} = \text{LayerNorm}(\beta y_{v_i} + (1 - \beta) h_{v_i})$, where $y_{v_i} = \text{Concat}(y_{v_i}^1, \ldots, y_{v_i}^K)$, $h_{v_i} = \text{Concat}(h_{v_i}^1, \ldots, h_{v_i}^K)$. Note that to reduce the burden of hyperparameter tuning, we fix $\beta = 0.5$ except for the experiment in Appendix C.4.

## 4.4 Optional: Factor Discrimination Loss

Existing disentangled representation learning methods (Liu et al., 2020; Yang et al., 2020) have widely adopted a factor discrimination loss aiming at promoting factors to contain different information. Following (Zhao et al., 2022), we added a factor discrimination loss $\mathcal{L}_{dis}$ to the final loss, i.e., $\mathcal{L} = \mathcal{L}_{task} + \lambda \mathcal{L}_{dis}$[6]. Details can be found in the Appendix B.2. Using the factor discrimination loss increases the performance of our model as can be seen in Table 9, Table 12 and Table 13. However, **we consider this loss to be an optional component** of our model, as excluding it simplifies the model by reducing the hyperparameter search space, making it more comparable to that of GAT (Appendix B.5). We expect this simplification to allow Natural-HNN to be broadly applicable to various fields that can be modeled by hypergraphs just like popular GCN or GAT in conventional graphs, which are broadly used as the GNN encoder regardless of the field of application. Nevertheless, Natural-HNN shows outstanding performance even without the factor discrimination loss by capturing the interaction context of an hyperedge (Section 5).

## 5 Experiments

Since there is no benchmark dataset verifed to contain useful interaction context that is related to the task, we instead performed cancer subtype classification task, which is to identify a subtype of a specific cancer for each patient (Section 5.2). Interaction context of genes (i.e., functionality of pathway) is directly related to the label (i.e., cancer subtype) in the cancer subtype classification task. We also perform qualitative analysis to validate whether our model captures the context-related factors (Section 5.3). Finally, we performed training time analyses in Section 5.4.

### 5.1 Experimental Setup

**Dataset.** For the cancer subtype classification task, we downloaded clinical data for 8 cancer types (BRCA, STAD, SARC, LGG, CESC, HNSC, KIPAN and NSCLC) and preprocessed data following Pathformer (Liu et al., 2023) (Details in Appendix A.2). Every patient (i.e., a hypergraph) has the same genes (i.e., nodes) and pathways (i.e., hyperedges), but the clinical data (i.e., gene representations) are different. The data statistic of each cancer data is provided in Appendix A.1.

**Compared Methods.** We compared Natural-HNN with HNNs introduced in Section 2. Specifically, HGNN(Feng et al., 2019), HCHA (Bai et al., 2021), HNHN (Dong et al., 2020), UniGCNII (Huang & Yang, 2021), AllDeepSets (Chien et al., 2021), AllSetTransformer (Chien et al., 2021), HyperGAT (Ding et al., 2020), SHINE (Luo, 2022), ED-HNN (Wang et al., 2022b), ED-HNNII (Wang et al., 2022b) and a hypergraph disentangling method HSDN (Hu et al., 2022) are used as baselines. Implementation details of some baselines and their variants are described in App. B.1.

**Evaluation.** We randomly split the data into 50%/25%/25% for training/validation/test set. We measured average and standard deviation of the performances for 10 different data splits. The hyperparameter search space is provided in Appendix B.5.

---

[6]$\mathcal{L}_{task}$ denotes the task related loss calculated from cross-entropy loss with labels and predictions. Details are available at Appendix B.3

Table 1: Model performance on cancer subtype classification task (Macro F1). Top two models are colored by **First**, **Second**. † : the variant of the model using multihead attention. ⋆ : $\mathcal{L}_{dis}$ is not used.

| Method | BRCA | STAD | SARC | LGG | HNSC | CESC | KIPAN | NSCLC |
|---|---|---|---|---|---|---|---|---|
| HGNN | 0.726 ± 0.053 | 0.563 ± 0.040 | 0.684 ± 0.067 | 0.694 ± 0.033 | 0.799 ± 0.053 | 0.835 ± 0.052 | 0.921 ± 0.016 | 0.959 ± 0.016 |
| HCHA | 0.704 ± 0.051 | 0.558 ± 0.044 | 0.675 ± 0.068 | 0.682 ± 0.041 | 0.783 ± 0.055 | 0.844 ± 0.054 | 0.920 ± 0.015 | 0.954 ± 0.009 |
| HNHN | 0.697 ± 0.046 | 0.573 ± 0.072 | 0.688 ± 0.075 | 0.674 ± 0.038 | 0.791 ± 0.035 | 0.837 ± 0.059 | 0.920 ± 0.021 | 0.958 ± 0.016 |
| UniGCNII | 0.697 ± 0.052 | 0.617 ± 0.059 | 0.728 ± 0.066 | 0.663 ± 0.039 | 0.830 ± 0.030 | 0.841 ± 0.046 | 0.935 ± 0.012 | 0.949 ± 0.017 |
| AllDeepSets | 0.716 ± 0.058 | 0.557 ± 0.044 | 0.599 ± 0.058 | 0.665 ± 0.046 | 0.801 ± 0.058 | 0.870 ± 0.044 | 0.912 ± 0.015 | 0.953 ± 0.010 |
| AllSetTransformer | 0.743 ± 0.057 | 0.553 ± 0.046 | 0.719 ± 0.052 | 0.653 ± 0.038 | 0.814 ± 0.036 | 0.847 ± 0.046 | 0.925 ± 0.013 | 0.953 ± 0.014 |
| HyperGAT | 0.637 ± 0.121 | 0.534 ± 0.063 | 0.574 ± 0.153 | 0.665 ± 0.054 | 0.789 ± 0.061 | 0.832 ± 0.046 | 0.899 ± 0.037 | 0.927 ± 0.020 |
| HyperGAT† | 0.641 ± 0.115 | 0.502 ± 0.087 | 0.584 ± 0.150 | 0.646 ± 0.043 | 0.791 ± 0.079 | 0.827 ± 0.041 | 0.896 ± 0.025 | 0.939 ± 0.009 |
| SHINE | 0.446 ± 0.155 | 0.371 ± 0.135 | 0.529 ± 0.160 | 0.628 ± 0.104 | 0.718 ± 0.055 | 0.745 ± 0.159 | 0.837 ± 0.197 | 0.866 ± 0.128 |
| SHINE† | 0.651 ± 0.053 | 0.532 ± 0.064 | 0.673 ± 0.059 | 0.650 ± 0.046 | 0.770 ± 0.040 | 0.837 ± 0.061 | 0.925 ± 0.017 | 0.954 ± 0.013 |
| HSDN | 0.757 ± 0.044 | 0.629 ± 0.045 | 0.726 ± 0.063 | 0.692 ± 0.038 | 0.811 ± 0.044 | 0.867 ± 0.033 | 0.937 ± 0.005 | 0.961 ± 0.013 |
| ED-HNN | 0.735 ± 0.047 | 0.615 ± 0.050 | 0.718 ± 0.071 | 0.700 ± 0.030 | 0.835 ± 0.047 | 0.875 ± 0.053 | 0.931 ± 0.013 | 0.955 ± 0.012 |
| ED-HNNII | 0.722 ± 0.045 | 0.536 ± 0.057 | 0.650 ± 0.087 | 0.695 ± 0.039 | 0.845 ± 0.025 | 0.895 ± 0.044 | 0.930 ± 0.015 | 0.953 ± 0.012 |
| Natural-HNN⋆ (Ours) | 0.804 ± 0.036 | 0.659 ± 0.049 | 0.745 ± 0.045 | 0.707 ± 0.035 | 0.862 ± 0.045 | 0.881 ± 0.042 | 0.934 ± 0.010 | 0.962 ± 0.013 |

## 5.2 RESULTS FOR CANCER SUBTYPE CLASSIFICATION

The cancer subtype classification task can be considered as a hypergraph classification task, since every patient (i.e., a hypergraph) has the same genes (i.e., nodes) and pathways (i.e., hyperedges). Specifically, we generated the representation of a hyperedge by simply concatenating representations of hyperedges in a hypergraph following Pathformer (Liu et al., 2023), due to the lack of an effective pooling method reflecting the hypergraph topology developed to date. Then, we applied one layer MLP as the classifier. We inevitably excluded SheafHyperGNN and SheafHyperGCN (Duta et al., 2024) in cancer subtype classification task due to extensive hyperparameter search space (Appendix C.4) with extremely long training time (Section 5.4). We have the following observations in Table 1. **1)** Natural-HNN shows superior performance in most of the cancers with large performance gap compared with most of the models. Especially, we achieve large performance improvements compared with the convolution-based methods as well as AllDeepSets, which cannot leverage the interaction contexts. In the case of BRCA, we achieve about 5% performance improvement compared with the second best model. This result can be attributed to the following two facts: *First*, pathways contain **"context-dependent interaction"**[7] that reflect various functional semantics (Stoney et al., 2018; 2015). *Second*, cancers are directly related to the functions of multiple pathways (Windels et al., 2022; Stoney et al., 2018). Thus, we can conclude that reflecting various functional context of pathways is important in cancer related tasks and our model benefited by effectively capturing such interaction contexts. **2)** Natural-HNN does not show impressive performance on KIPAN and NSCLC compared to other datasets. This is due to the fact that those cancers are relatively easy to be classified with only the gene features (Wang et al., 2021; Oh et al., 2021). **3)** Natural-HNN outperforms the disentangle-based model, HSDN, with a large performance gap. Although HSDN mainly aimed to capture the structural semantics, it is similar to ours in that it can potentially capture interaction types by giving different factor importance for each hyperedge. They also used similarity-based criterion for disentanglement by comparing similarity between factor representations of a hyperedge and nodes. However, the superior performance of Natural-HNN validates that the naturality-guided disentanglement can better integrate contextual information of interaction.

## 5.3 CAPTURING THE INTERACTION CONTEXT OF HYPEREDGES

**Analysis on Cancer Datasets.** To validate that Natural-HNN can capture the interaction context, we checked whether our model captures functional semantics of genetic pathways. Because the models rely solely on cancer subtype labels during training[8], we expect the interaction contexts of informative hyperedges (such as cancer-related pathways) to be captured by the models, while non-informative hyperedges (such as pathways not relevant to cancer) are not. For this experiment, we first selected top-15 pathways[9] based on the SHAP value for each model (Natural-HNN in Figure 5 top and HSDN in Figure 5 bottom). Note that we rely on the SHAP value since information regarding which pathways are relevant to cancers is not given. Then, after clustering these 15 pathways with CliXO algorithm (Kramer et al., 2014), we calculate the similarity between clusters based on the average similarity of pathways that belong to each cluster. Our goal is to check how well Natural-HNN preserves the functional semantic similarity between pathway clusters compared with the cluster similarity calculated with Lin's method (Lin et al., 1998) (BMA), which

---

[7]**A direct quote from (Stoney et al., 2018)**

[8]This means that models do not use external data related to pathway types or pre-trained models.

[9]Only a few pathways are related to each type of cancer. We can also observe this with the SHAP value distribution in Figure 7

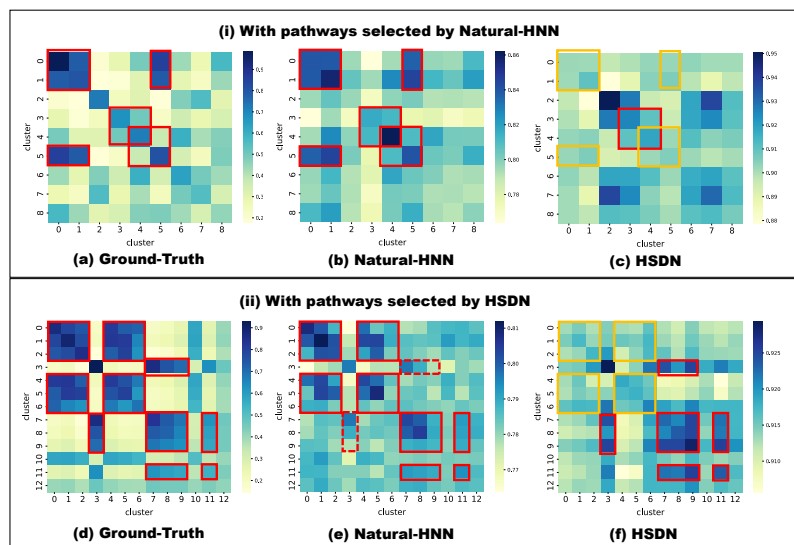

Figure 5: Captured interaction context. Captured patterns are shown in red boxes and not captured patterns are shown with orange boxes. Weakly captured cases are marked as dotted red block.

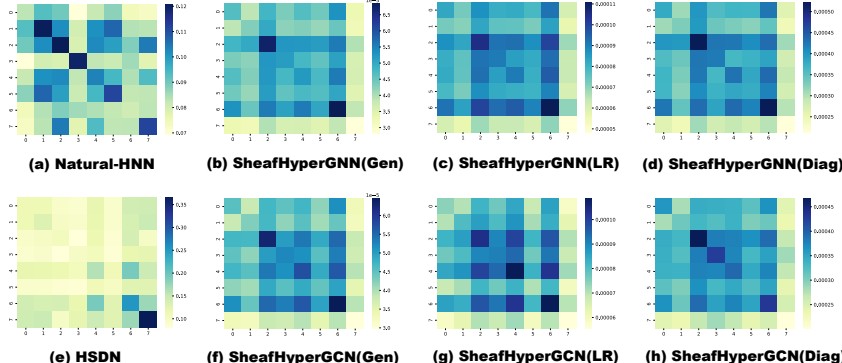

Figure 6: Similarity of transformation matrices between hyperedge types with Natural-HNN (a), HSDN (e), and variants of SheafHyperGNN (b,c,d) / SheafHyperGCN (f,g,h). For (a) and (e), we used 8 as the number of factors.

we consider as the ground-truth. For HSDN and Natural-HNN, cluster similarity is calculated based on the relevance score vector of each hyperedge $e_i$ across all factors, i.e., $\alpha_i = [\alpha_i^1, ..., \alpha_i^K]$, which can be calculated as $1/(1 + \|\alpha_i - \alpha_j\|_2)$. As the experiment setting is somewhat complicated, we described the detailed procedure in Appendix A.3.

The result on the BRCA datset is shown in Figure 5. The row and column of each heatmap is the index of the pathway clusters and color represents similarity between clusters. Figure 5 (a), (b) and (c) shows the measured similarity between clusters with pathways selected by Natural-HNN. Comparing (b) and (c) with (a), we observe that Natural-HNN preserves the functional similarity (red box) better than HSDN, which fails to do so (orange box). Moreover, Figure 5 (d), (e) and (f) shows the measured similarity between clusters with pathways selected by HSDN. An interesting observation is that even with the pathways that were informative to the HSDN, HSDN fails (orange box) to preserve the functional similarity between clusters while Natural-HNN could capture them. The results imply that the naturality condition in category theory is effective in capturing the interaction context of an hyperedge. Additional analyses are described in Appendix H.5

**Analysis on Synthetic Dataset.** Since sheaf-based method can be affected by other contexts such as the context of each individual node (e.g., function of gene in cancer subtype dataset) rather than the actual interaction context, we construct a synthetic dataset that is deliberately generated to only contain the interaction context. To compare the ability of capturing the interaction context with a sheaf-based method, SheafHyperGNN (Duta et al., 2024), we created a synthetic hypergraph with 3200 nodes, 4 node labels, and 2400 hyperedges with 8 hyperedge types (i.e., 300 hyperedges per

hyperedge type). The details for synthetic hypergraph generation is provided in Appendix A.8. First, to validate whether Natural-HNN and HSDN correctly capture the interaction context (or hyperedge type), we calculated similarity of transformation matrices assigned to hyperedges, and then check whether the similaity between hyperedges that belong to the same type is high. Specifically, the similarity between two hyperedges $e_i$ and $e_j$ is computed by $1/(1 + \|W_i - W_j\|_2)$ where $W_i$ is the transformation matrix for hyperedge $e_i$[10]. For sheaf-based methods, as the transformation matrix is defined for each (node, hyperedge) pair, i.e., $\mathcal{F}_{v \trianglelefteq e}$, we compute the similarity between two hyperedges based on the average of the similarity of all possible pairs. For example, given two hyperedges $e_i = (v_1, v_2)$ and $e_j = (v_3, v_4, v_5)$, we calculate the average of the following similarities: $(\mathcal{F}_{v_1 \trianglelefteq e_i}, \mathcal{F}_{v_3 \trianglelefteq e_j})$, $(\mathcal{F}_{v_1 \trianglelefteq e_i}, \mathcal{F}_{v_4 \trianglelefteq e_j})$, $(\mathcal{F}_{v_1 \trianglelefteq e_i}, \mathcal{F}_{v_5 \trianglelefteq e_j})$, $(\mathcal{F}_{v_2 \trianglelefteq e_i}, \mathcal{F}_{v_3 \trianglelefteq e_j})$, $(\mathcal{F}_{v_2 \trianglelefteq e_i}, \mathcal{F}_{v_4 \trianglelefteq e_j})$ and $(\mathcal{F}_{v_2 \trianglelefteq e_i}, \mathcal{F}_{v_5 \trianglelefteq e_j})$. Figure 6 shows the results with top-5 informative hyperedges for each hyperedge type[11], where the ideal result would show dark blue colors in the diagonal[12]. The strong similarities in the diagonal of the heatmap of Natural-HNN (Figure 6(a)) compared with that of HSDN (Figure 6(e)) validates again that Natural-HNN is superior in capturing the interaction context of an hyperedge. Besides, sheaf-based methods show ambiguous result (Figure 6(b-d, f-h)). More results on Appendix D.3 shows that sheaf-based methods hardly capture interaction types.

Table 2: Time took for training 1 epoch, measured in seconds. $d_c$ denotes the dimension of channel (hidden dimension) and $d_s$ denotes stalk dimension for sheaf-based models. # denotes 'number of'.

| $(d_c, d_s$ or # factors) | SheafHyperGNN(Gen) | SheafHyperGNN(LR) | SheafHyperGNN(Diag) | Natural-HNN |
|---|---|---|---|---|
| (16, 2) | 14968.699 (04h 09m) | 15064.670 (04h 11m) | 7691.438 (02h 08m) | 0.544 ± 0.001 |
| (64, 8) | 239376.921 (66h 30m) | 240024.821 (66h 40m) | OOM | 1.853 ± 0.002 |

Table 3: Time took for training 1 epoch, measured in seconds. $d_c$ denotes the dimension of channel (hidden dimension).

| $(d_c$, # heads or factors) | HGNN | AllDeepSets | AllSetTransformer | HSDN | Natural-HNN |
|---|---|---|---|---|---|
| (16, 2) | 0.217 ± 0.000 | 1.195 ± 0.002 | 1.108 ± 0.002 | 0.289 ± 0.000 | 0.544 ± 0.001 |
| (64, 8) | 0.831 ± 0.001 | 2.463 ± 0.005 | 2.671 ± 0.002 | 0.996 ± 0.000 | 1.853 ± 0.002 |

## 5.4 TRAINING TIME ANALYSIS

To validate that Natural-HNN is scalable and efficient, we measured the time taken for training 1 epoch in BRCA dataset. We measured the time 5 times and averaged them. As sheaf-based methods take too long time, we measured them only once. For sheaf-based methods, we calculated the representation of a hyperedge as an average of the transformed node features (i.e., average of $\mathcal{F}_{v \trianglelefteq e} x$). In Table 2, we observe that while Natural-HNN takes only a few seconds, sheaf-based methods take from 2 to 66 hours per epoch depending on $d_c$, $d_s$ and # factors, which makes it not applicable to real-world applications. In Table 3, Natural-HNN is much more efficient than AllDeepSets or AllSetTransformers, while being less efficient than the convolution-based methods. However, considering the superiority of Natural-HNN in terms of the downstream task, we argue that it is acceptable. More results in Appendix F.2 shows that Natural-HNN is scalable and efficient.

## 6 CONCLUSION

In this work, we propose Natural-HNN, which captures the interaction context of nodes within a hyperedge during the message passing process. We analyzed compositional structure in hypergraph message passing and focused on the naturality condition that must be satisfied between entangled and disentangled representations. The power of category theory enabled us to create a **simple but effective model that balances the trade-off between the expressiveness and generalization even with a small hyperparameter search space** (Appendix B.5), which is even comparable to GAT. Moreover, the category theory allowed our model to pursue the intended purpose, capturing the interaction context of nodes within a hyperedge, without the help of external knowledge or a complex objective function. Given the potential of Natural-HNN in capturing inherent heterogenity in the homogeneous data, **we believe that our model will contribute to the domains where heterogenity information is unavailable as we have seen in the biological pathways.**

---

[10]Note that $W_i = [\alpha_i^1 W^1, ..., \alpha_i^K W^K]$, where $W^k$ is the weight of the first layer of $k$-th factor-specific MLP.

[11]Not all hyperedges are equally informative for node classification. More results with different number of hyperedges per type are provided in Appendix D.3

[12]The non-diagonal parts do not need to be bright yellow, as the colors represent relative values.

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

# Appendix

# A DATASET AND EXPERIMENT DETAILS

## A.1 STATISTICS : CANCER SUBTYPE CLASSIFICATION DATASET

The statistics of cancer datasets are shown in the Table 4. Note that every hypergraphs in all 8 cancers have 1497 pathways (hyperedges) and 11552 genes (nodes) with 9 feature dimension. The degree statistics of cancer dataset is shown in the Table 5. When converted to a graph with star-expansion, the graph contains 98013 edges. When converted to a graph with clique-expansion, the graph contains 10114890 edges. Thus, converting the hypergraph into a graph with clique-expansion requires large computation during message passing. The downloading and preprocessing details are provided in Appendix A.2

Table 4: Statistics of 8 cancer datasets used for cancer subtype classification task.

| dataset | summary | class distribution(counts) |
|---|---|---|
| BRCA | 5 class, 769 hypergraphs | Normal-like 33, Her2 44, Basal-like 134, LumB 143, LumA 415 |
| STAD | 5 class, 341 hypergraphs | CIN 200, EBV 29, GS 46, MSI 59, HM-SNV 7 |
| SARC | 4 class, 257 hypergraphs | LMS 104, MFS/UPS 75, DDLPS 57, Other 21 |
| LGG | 2 class, 503 hypergraphs | G2 242, G3 261 |
| HNSC | 2 class, 507 hypergraphs | HPV- 411, HPV+ 96 |
| CESC | 2 class, 280 hypergraphs | AdenoCarcinoma 46, SquamousCarcinoma 234 |
| KIPAN | 3 class, 649 hypergraphs | KICH 65, KIRC 313, KIRP 271 |
| NSCLC | 2 class, 813 hypergraphs | LUAD 451, LUSC 362 |

Table 5: statistics of hypergraphs in cancer subtype classification task

| | min | median | mean | max | std |
|---|---|---|---|---|---|
| node degree | 2 | 5 | 8.485 | 239 | 13.301 |
| hyperedge degree | 13 | 35 | 57 | 1371 | 84.720 |

## A.2 PREPROCESSING : CANCER SUBTYPE CLASSIFICATION DATASET

The overall procedure was adopted from Pathformer (Liu et al., 2023). However, statistics of the data can be slightly different due to the difference of time at which the data was downloaded.

### CREATING HYPERGRAPH

We downloaded pathways from several pathway databases including KEGG (Kanehisa & Goto, 2000), PID (Schaefer et al., 2009), Reactome (Croft et al., 2010) and Biocarta.(Nishimura, 2001). The pathways were selected based on their size and overlap ratio with other pathways. These two conditions must be considered as 1) extremely large pathways do not represent specific functions but rather general functions, 2) small pathways complicate interpretations 3) overlapping pathways cause redundancies. The more detailed explanations can be found in (Reimand et al., 2019). Pathways with too small or too big size or large overlaps are excluded. A specific threshold was chosen following the Pathformer.

### GENERATING HYPERGRAPH LABELS

For BRCA and STAD, we gathered cancer subtypes from TCGA (Weinstein et al., 2013) using TCGAbiolinks (Colaprico et al., 2016; Silva et al., 2016; Mounir et al., 2019) R library. For the rest of 6 cancer datasets we downloaded cancer subtypes from Broad GDAC Firehose (`https://gdac.broadinstitute.org/`)[13]. KIPAN and NSCLC, specifically, was created by integrating KIRC, KICH, KIRP and LUAD, LUSC each as shown in Table 4. This is the reason why it is easy to classify cancer subtypes in KIPAN dataset.

---

[13]Pathformer used labels from pan-cancer atlas study (Sanchez-Vega et al., 2018) for HNSC, CESC and SARC. However, we decided to use the one in Broad GDAC Firehose since it was easier to process the same data

## GENERATING NODE FEATURES

We gathered mRNA/miRNA expression, DNA methylation[14], DNA copy number variation (CNV)[15] using TCGAbiolinks. Gene lengths were acquired from biomaRt R package (Durinck et al., 2009; 2005). The procedure of processing each data with Gistic2 (Mermel et al., 2011), normalization by TPM are adopted from Pathformer. At the end of the processing step, we calculate statistics (mean, min, max, count) of modalities as values for each feature dimension.

### A.3 EXPERIMENT DETAILS OF CAPTURING CONTEXT TYPES

To check whether HNNs could capture functional semantics of pathways (i.e, interaction context of hyperedges), we need context labels for each hyperedge. However, there is no data that annotates the functional semantics of genetic pathways. Instead, we rely on the methods in computational biology to measure and create ground truth.

We clustered functionally similar pathways and measured functional similarity between clusters. Since each cluster is consisted of functionally similar pathways, we can consider each cluster index as a kind of a label that indicates a functional context type. By comparing the functional similarity between clusters earned from model and ground truth, we can check whether the model effectively captured functional semantics of pathways. If the similarity patterns between clusters (i.e., relative similarity scores that are shown as color in heatmap) of predicted result and the ground truth are similar, we can conclude that model could capture functional semantics. We do not directly compare the exact values of prediction and the ground truth since the way of calculating the value is different in prediction (calculation based on relevance scores $\alpha_{e_i}^k$) and ground truth (algorithm used in computational biology).

In order to perform the experiment, we need to consider the followings: **1)** Which pathways need to be analyzed? **2)** How to get ground truth pathway functions **3)** How to calculate ground truth functional similarity between pathways **4)** How to cluster functionally similar pathways in a reliable manner **5)** How to measure ground truth cluster similarity and how to predict cluster similarity with model outputs.

**Which pathways need to be analyzed?** There are two reasons behind selecting pathways : 1) Since CliXO algorithm (Appendix A.6) used for clustering pathways takes a lot of time, the number of pathways to be analyzed must be reduced. 2) The ground truth functional similarity (Appendix A.5) contains vast biological context derived from biological domain knowledge or researches, which might not be present in our dataset. Since our dataset contains only cancer-specific information, there is no way to capture non-existing context (contexts that are not related to cancer) without external supervision. Thus direct comparison between the ground truth and our result is impossible. The most ideal way for fair comparison would be selecting the ground truth that is only relevant to our dataset or task. However, it is impossible since there are no databases with annotated context (cancer or environment) specific pathway functionalities. An alternative way was selecting the pathways that were informative or important in the decision of the model. If a model can correctly capture functional context of pathways, since pathway functions are highly related to the cancers (Windels et al., 2022; Stoney et al., 2018), informative pathways (for the model prediction) are the pathways that contain cancer-specific contexts. Since we only need to check whether functional context are correctly captured under the cancer specific circumstances or condition, by selecting those pathways, we can compare functional similarities that are specific to our data or cancer[16]. The details for selecting pathways are described in Appendix A.4.

**How to get ground truth pathway functions.** Since there is no database that annotates functional similarity scores between pathways, we rely on methods used in computational biology. Hence, we need to get ground truth pathway functions. Similarity calculations and clusterings are based on the annotation of pathway functions. The details are described in Appendix A.5.

**How to calculate ground truth functional similarity between pathways.** Based on the functions of pathways, pathway functional similarity can be calculated. The calculated similarity will be used

---

[14]but we do not use promoter methylation

[15]but we do not use gene level CNV

[16]On the other hand, if the model could not correctly capture pathway functionalities, cancer irrelevant pathways will be selected and will have different result from the ground truth in section 5.3

in clustering and generating ground truth functional similarity between clusters. The details are dealt in Appendix A.5.

**How to cluster functionally similar pathways in a reliable manner.** With functional similarity between pathways, we can cluster functionally similar pathways with CliXO algorithm. The details and example results are shown in Appendix A.6.

**How to measure ground truth cluster similarity and how to predict cluster similarity with model outputs.** Finally, we need to devise a way to measure the similarity between clusters based on the model outputs. Also, we need to measure ground truth functional similarity between clusters. The details are described in Appendix A.7.

In summary, the procedure of experiments can be described as follows. First, we get functional annotation of pathways (hyperedges). Second, we calculate functional similarity between pathways based on annotations. Third, we select pathways to be analyzed based on the model output. Fourth, we cluster the selected pathways with pathway similarity. Finally, we calculate the predicted functional similarity between clusters from model prediction and compare that with the ground truth cluster similarity. The detailed explanation for the result is provided in Appendix H.5.

## A.4 SELECTING PATHWAYS WITH SHAP VALUES

To select pathways that were the most informative for prediction, we provide the final representation of pathways generated by a model, 1 layer classifier (MLP) as well as labels to the DeepExplainer to get SHAP values. Then we select top-k pathways based on the SHAP value. Note that only small number of pathways are relevant to the task as shown in Figure 7. This is due to the fact that not all pathways are related to very specific type of cancer. Although Natural-HNN and HSDN both use the same number of pathways (top-k), the pathways selected by each model can be different. This also leads to different number of clusters in Figure 5 and 19.

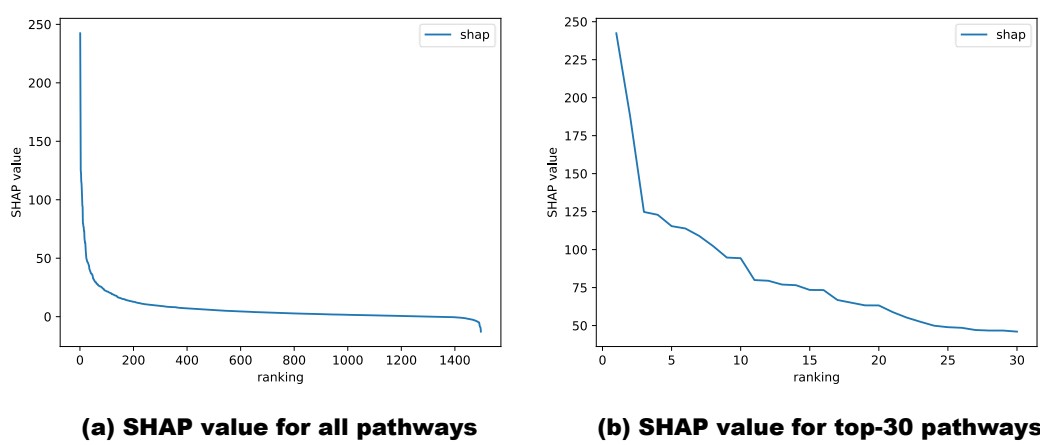

(a) SHAP value for all pathways  (b) SHAP value for top-30 pathways

Figure 7: SHAP value distribution of Natural-HNN on BRCA dataset. X axis represents ranking and Y axis represents SHAP value.

## A.5 CALCULATING FUNCTIONAL SIMILARITY BETWEEN PATHWAYS

This process consists of two steps: 1) assigning pathway level function to pathways and 2) calculating functional semantic similarities between pathways. For both two steps, we adopted the most frequently used and verified methods through several studies. For the assignment of pathway functions, we use GO enrichment analysis. Gene ontology (GO) (Ashburner et al., 2000; Aleksander et al., 2023) is a functional annotation of genes that has a hierarchical structure. Note that, however, the hierarchical structure of functional annotations is close to a directed acyclic graph (DAG) rather than a tree-like hierarchical structure. As an example, we can see DAG structure in the result of CliXO algorithm in the Figure 8. We can computationally annotate pathway functions with GO terms using GO enrichment analysis. We use 'enrichGO' function provided by R package cluster-Profiler (Yu et al., 2012), with pvalue of 0.01 followig the paper (Stoney et al., 2018). Then we

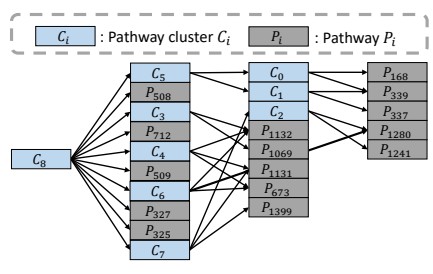
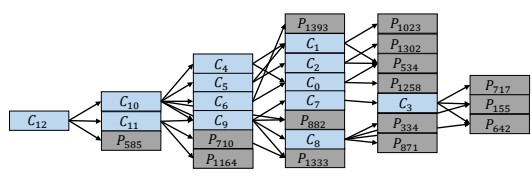
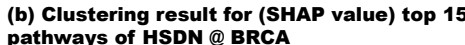

**(a) Clustering result for (SHAP value) top 15 pathways of Natural-HNN @ BRCA**

**(b) Clustering result for (SHAP value) top 15 pathways of HSDN @ BRCA**

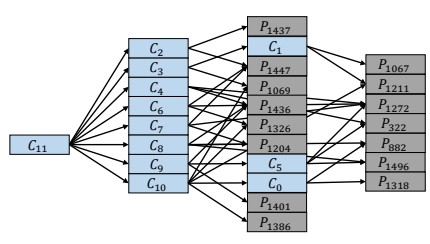
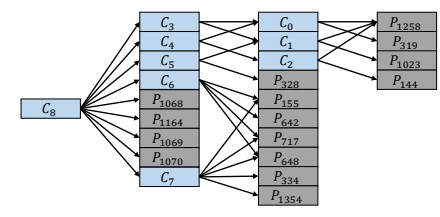

**(c) Clustering result for (SHAP value) top 15 pathways of Natural-HNN @ CESC**

**(d) Clustering result for (SHAP value) top 15 pathways of HSDN @ CESC**

Figure 8: The result of applying CliXO algorithm to top-15 pathways of Natural-HNN and HSDN on BRCA and CESC. The pathway number denotes the index of pathway in our dataset (hyperedge index).

selected the most specific GO terms with set cover algorithm proposed in (Stoney et al., 2018) to assign pathways precise representation of their functions.

The next step is calculating functional semantic similarities between pathways. We used Lin's method (Lin et al., 1998) with best matching average (BMA) as the combination was proven to perform well with CliXO and was proven to be robust in incomplete annotation cases in (Liu & Thomas, 2019). We used mgoSim function in R package GOSemSim (Yu et al., 2010; Yu, 2020) for the calculation of Lin's method.

### A.6 Assigning Pathway Type with CliXO

To cluster functionally similar pathways, we adopted CliXO (Kramer et al., 2014). It was originally designed to cluster gene function annotations (GO) and has been used in multiple biological studies(Kratz et al., 2023; Qin et al., 2020). However, it can also be effectively applied to higher functional semantics such as pathways as in (Zheng et al., 2021). We used official implementation of CliXO 1.0 for our research. We used the following 4 values as hyperparameter of CliXO : a = 0.1, b = 0.6, m = 0.005, s = 0.2.

Since CliXO can cluster functionally similar pathways, we can assign interaction types to pathways by assigning them to the cluster. Figure 8 shows the result of applying CliXO for top-15 pathways selected by Natural-HNN or HSDN for BRCA as well as CESC. Unlike other hierarchical clustering based methods, CliXO created clusters having DAG structure. Considering that GO also has DAG structure, CliXO can be seen as a natural way of reflecting complex structure or relations in biology.

### A.7 Calculating Functional Similarity between Clusters

**Ground Truth** Given a pair of clusters, calculating functional similarity between them is simple. We average the similarity of all possible pathway pairs belonging to different clusters to get functional similarity between clusters.

**Model's prediction** If a model correctly captures functional context of pathways, then the relevance scores $(\alpha_i^k)$ of two similar pathways must be similar for all factors. Thus we define the similarity between pathways as $\frac{1}{1+\|\alpha_i-\alpha_j\|_2}$, where $\alpha_i = [\alpha_i^1, ..., \alpha_i^K]$ is a factor vector of pathway (hyperedge) $e_i$. The cluster similarity can be calculated in the same way as in the ground truth case. We average the similarity of all possible pathway pairs belonging to different clusters to get functional similarity between clusters.

## A.8    SYNTHETIC DATASET GENERATION

**Purpose of creating dataset** The main purpose of synthetic hypergraph experiment is to validate whether each model can capture interaction contexts. Cancer subtype classification dataset will be enough to validate Natural-HNN and HSDN. However, it does not fit for sheaf-based models for two reasons : **1)** SheafHyperGCN and SheafHyperGNN do not scale well, taking too long for training. **2)** Since sheaf-based methods learns projections or transformations for every (node, hyperedge) pair, there is a possibility that sheaf-based methods not only capture interaction contexts but also contexts that are not related to interaction. For example, if genes in genetic pathway have their own context, it might be reflected to transformation matrices of sheaf-based methods. In order to validate sheaf-based methods, we need a synthetic dataset that does not contain contexts other than interaction.

**Conditions to be satisfied** However, generating synthetic hypergraph with meaningful interaction context is difficult. There are several conditions that generation process must satisfy : **1)** Since interaction context must be crucial for predicting labels, raw node features must not have correlation with labels. **2)** Hyperedge types must be highly related to labels. In other words, convolution-based models must not be able to easily predict labels. If one of above conditions fail, a model can easily predict labels without capturing hyperedge types. Generating raw node features, assigning hyperedge types and labels satisfying above condition is complicated.

**Diffculties.** There can be three ways to generate a hypergraph : **1)** Fixing labels, hyperedges with types and then generating node features satisfying condition, **2)** Fixing labels, node features and then creating hyperedges with types, **3)** Fixing hyperedges with types, node features and then generating node features.

- For the **first case**, it is hard to generate node features satisfying conditions. If feature of a node is related to hyperedge type, convolution-based methods will easily predict labels. A model will not rely on hyperedge types for prediction. If feature of a node is related to its label, information from neighbor might not be informative to predict labels. In this condition, it will be hard to verify whether a model captures interaction context as the model will not rely on hyperedges for prediction. Finally, if we randomly create node features by randomly sampling from Gaussian distribution, we cannot know whether hyperedge type is informative for predicting labels.

- For the **second case**, it is hard to create hyperedges with types by just reading through node features and labels. It is hard to know whether created hyperedges with types are informative for label prediction.

- For the **last case**, we can simply generate features and hyperedges with types. Based on the created hypergraph, we can simply assign labels. The detailed explanations are described below.

**Key concept of hypergraph generation.** We brought an idea from a group discussion example described in the introduction as it fits the concept of interaction context. Initial opinions of individuals before discussion can be considered as node features. Individuals can have their own, different ideas and do not necessarily have correlation with interaction types. The group discussion will change the opinion of individuals, which can be implemented as message passing, and will form final opinions of individuals. Based on their final opinions, we can classify individuals or assign labels.

**Generation procedure.** As individuals can have their own opinion before discussion, we generate node features by sampling from Gaussian distribution. We randomly create hyperedges with random size[17] and randomly assign hyperedge types. We made sure that the hypergraph is connected (every node is reachable) and an equal number of hyperedges are assigned to each hyperedge type. The t-SNE result of node features can be seen in Figure 10 (a). We created hyperedge type dependent HNN (Figure 9), which can operate on multiplex hypergraph. We project node features to each type of

---

[17]Size is sampled from Gaussian distribution

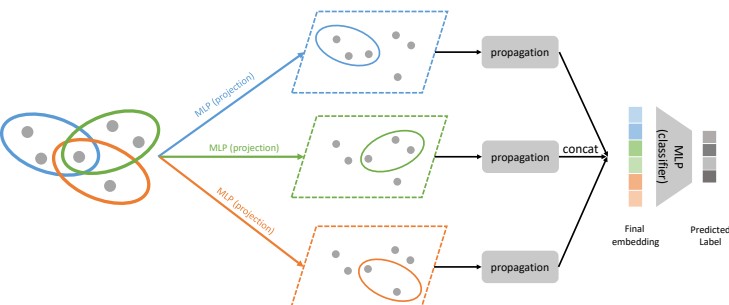

Figure 9: Overall architecture of HNN used for generating synthetic hypergraph.

hypergraphs with 1 layer MLP and perform 1-layer message passing[18] (HCHA without parameter). The concatenation of 1 layer message passing results from hypergraphs of each type will create **final embedding** of nodes. The HNN predicts the label from final embedding with classifier (1 fully connected layer). The t-SNE result of the final embedding without training is provided in Figure 10 (b). Since node features as well as weights of HNN are all random, the final embeddings are not clustered. Thus, it is hard to assign labels in this state.

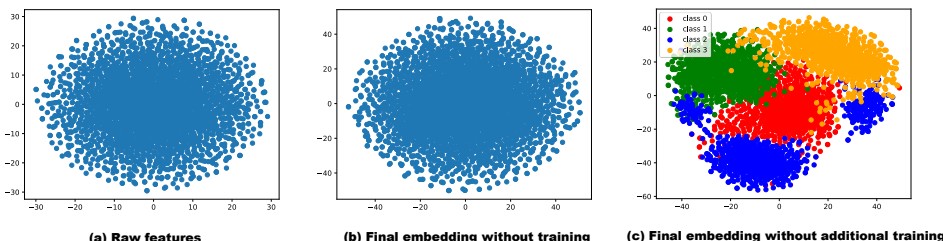

Figure 10: t-SNE result for node features, final embeddings without training, and that without additional training.

Thus, we decided to train HNN so that final embeddings can form clusters. For training, we assign labels based on the prediction of the model. The class with the highest predicted probability is assigned as a label to the node. However, we have to make sure that the number of nodes for each classess are equal to prevent all nodes from having the same class. For loss calculation, we used cross-entropy loss. The result of training HNN with 4000 epochs can be seen in Figure 10 (c). We can observe that not all classes are clustered well. Still, it is much better than the result in Figure 10 (b). We decided to perform additional training with 10000 epochs, but without condition that all classes must have equal number of nodes. The result can be seen in Figure 11. We can observe that the final embeddings are well clustered while raw node features does not. Hence, we believe hyperedge types are important in the generated hypergraph to get well clustered embeddings from noisy raw features[19].

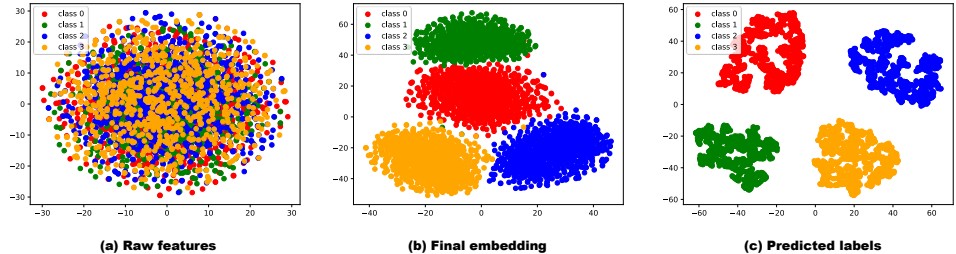

Figure 11: t-SNE result for raw features, final embedding, prediction result with label colored.

---

[18]If we use more than two layers of propagation, it is hard to know whether each hyperedge type in each layer is critically important or informative for labels as the influence of a type in one layer can be influenced by other type in the other layer. Thus, we used only 1 layer of propagation.

[19]When we trained HGNN with 1 layer, 64 hidden dimension, the accuracy was $58.075 \pm 1.908$.

**Process Summary.** Hypergraph generation process can be summarized as follows : **1)** Sample node features from Gaussian distribution, randomly create hyperedge with types. **2)** Train HNN. Labels are defined to be predicted label, but the number of labels need to be the same for every node classes. **3)** Perform additional training. Labels are defined to be predicted label, but the number of labels for each class does not necessarily be the same.

**Hyperparameters for hypergraph generation.** We created a small hypergraph with 3200 nodes with 4 type of labels, 2400 hyperedges with 8 hyperedge types (300 hyperedges per type). Node features with 100 dimension was generated from Gaussian distribution with mean 0 and standard deviation 3. The degree of each hyperedge was sampled from Gaussian distribution with mean 7 and standard deviation 2, but we made sure that all hyperedges contain more than 2 nodes.

# B    IMPLEMENTATION DETAILS

In Appendix B.1, we describes some implementation details of baselines and their variants, which can be different from official implementations. From Appendix B.2 to B.5, we describe implementation details for the components of Natural-HNN.

## B.1    BASELINES AND THEIR VARIANTS

We implemented HyperGAT based on the paper as its official implementation is different from what is explained in the paper. Moreover, as the original version of SHINE and HyperGAT do not involve multihead attention, we implement it for fair comparisons. For SHINE, we also implemented two versions, one without using $\mathcal{L}_{reg}$ and the other with $\mathcal{L}_{reg}$ which is a loss introduced by the paper for the purpose of making node representations to be similar if the nodes are included in the same hyperedge. However, we did not use the version with $\mathcal{L}_{reg}$ in cancer subtype classification task since the loss converts a hypergraph to a graph using clique expansion, which causes tremendous computational cost. For SheafHyperGCN and SheafHyperGNN, we used official implementation of the paper.

## B.2    FACTOR DISCRIMINATION LOSS

We defined a factor discrimination loss $\mathcal{L}_{dis}$ similar to the one used in (Zhao et al., 2022). In order to promote factors to contain different information, we use a factor classifier implemented with one layer MLP. Each factor representation of every hyperedge will be given as input to the factor classifier. The classifier needs to identify to which factor the factor representation belongs. If the classifier can correctly identify the factor with factor representation, i.e. if factor representations of two different factors of a hyperedge are distinguishable, it is highly likely that factors contain different information.

Specifically, we can calculate the loss by creating pseudo labels. For each factor representation of each hyperedge ($h_{e_i}^k$), we assign a pseudo label $Y_{e_i}^k = k$. Then the loss can be defined as follows:

$$\mathcal{L}_{dis} = -\sum_{e_i \in \mathcal{E}} \sum_{k=1}^{K} \sum_{c=1}^{K} \mathbf{1}(Y_{e_i}^k = c) log(softmax(MLP(h_{e_i}^k))) \tag{2}$$

This loss is applied to each layer of Natural-HNN. As described in Section 4.4, the final loss would be $\mathcal{L} = \mathcal{L}_{task} + \lambda \mathcal{L}_{dis}$. As mentioned before, $\mathcal{L}_{dis}$ is an optional part of our model. The hyperparameter search space for $\lambda$ is provided in Appendix B.5

## B.3    LOSS USED FOR TRAINING $\mathcal{L}_{task}$

After the final message passing layer of Natural-HNN, we get the final node embeddings $z_{v_i}$. The classifier of Natural-HNN will predict labels $p_{v_i} \in \mathbb{R}^C$ where $C$ denotes the number of classes. In other words, $p_{v_i,c}$ denotes the probability that node $v_i$ has class $c$ as answer. If we denote $l_{v_i}$ as the label (one-hot vector) for node $v_i$, the task loss can be calculated with cross-entropy loss.

$$\mathcal{L}_{task} = -\sum_{i=1}^{|V|} \sum_{c=1}^{C} l_{v_i,c} \log(p_{v_i,c}) \tag{3}$$

Note that, we use hyperedge embedding of the final layer instead of node embeddings for cancer subtype classification task.

## B.4    FACTOR ENCODER

In Section 4, we explained that we use $K$ number of MLPs to get $K$ factor representations. The resulting factor representation is a vector with size $d/K$ when desired output representation size of a layer is given as $d$. When implementing the factor encoder as a code, we use single MLP that outputs vector with size $d$. As described in H.1, applying $K$ different MLPs (with output vector size $d/K$) is the same as applying one MLP (with output vector size $d$) and chunking the vector

to smaller ones with size $d/K$. (i.e. First $d/K$ values corresponds to the $1^{st}$ factor representation, and following $d/K$ values corresponds to the $2^{nd}$ factor representation and so on.) Hence, in the right lane of Figure 4, the concatenation operation is not performed as the output of a single MLP is equivalent to a concatenated vector. The nonlinear activation function we used for factor encoder is hyperbolic tangent (tanh).

## B.5 HYPERPARAMETER SEARCH SPACE

We report the hyperparameter search space of each model in standard benchmark dataset as well as cancer subtype classification task. We used Adam optimizer for Natural-HNN. For the baselines, we closely followed optimizers or schedulers they used in their paper. Table 6 and Table 7 shows the hyperparameter search space in the standard benchmark dataset and cancer subtype datasets respectively. '♯ Total' denotes the number of all possible hyperparameter combinations that each model needs to search. 'cl' denotes the number of classifier layers. When the number of classifiers is larger than 1, those models have an additional hyperparameter that decides the hidden dimension of the classifier. ♯ MLP layer denotes the number of layers in MLP that was used in AllDeepSets, AllSetTransformer, ED-HNN, ED-HNNII. In the case of ED-HNN and ED-HNNII, there were three types of MLPs and each MLP could have different number of layers. $\lambda$ for $\mathcal{L}_{dis}$ is hyperparameter that changes the reflection ratio of the factor discrimination loss.

Table 6: Hyperparameter search space in standard benchmark dataset. † : MLP layers used in AllDeepSets, AllSetTransformer, ED-HNN, ED-HNNII

| models | ♯ cl | classifier dim | head (factor) | ♯ MLP layer † | $\lambda$ for $\mathcal{L}_{dis}$ | ♯ Total |
|---|---|---|---|---|---|---|
| HGNN | 1 | - | 1 | - | - | 32 |
| HCHA | 1 | - | 1 | - | - | 32 |
| HNHN | 1 | - | 1 | - | - | 32 |
| UniGCNII | 1 | - | 1 | - | - | 32 |
| AllDeepSets | 1,2 | 64,128,256,512 | 1 | 1,2 | - | 320 |
| AllSetTransformer | 1,2 | 64,128,256,512 | 1,2,4,8 | 1,2 | - | 1280 |
| HyperGAT | 1 | - | 1,2,4,8 | - | - | 128 |
| SHINE | 1 | - | 1,2,4,8 | - | - | 128 |
| HSDN | 1 | - | 1,2,4,8 | - | 0.0001, 0.0005, 0.001, 0.005, 0.01, 0.05, 0.1 | 896 |
| ED-HNN | 1,2 | 64,128,256,512 | 1 | $[0,1,2] \times [1,2] \times [0,1,2]$ | - | 2880 |
| ED-HNNII | 1,2 | 64,128,256,512 | 1 | $[0,1,2] \times [1,2] \times [0,1,2]$ | - | 2880 |
| Natural-HNN | 1 | - | 1,2,4,8 | 1 | - | 128 |
| Natural-HNN+$\mathcal{L}_{dis}$ | 1 | - | 2,4,8 | 1 | 0.0001, 0.0005, 0.001, 0.005, 0.01, 0.05, 0.1 | 672 |

For standard hypergraph benchmark datasets, we used [64, 128, 256, 512] as hidden dimension and [0.1, 0.01, 0.001, 0.0001] as learing rate. For weight decay, we used [0, 1e-5]. We fixed the number of layers to 2 unless the paper of a model fixed the number of layers to a specific number. In other words, if the paper of a model tuned the number of layers, we fixed them as 2. For example, we fixed the number of layers of ED-HNN and ED-HNNII as 2 since they tuned the number of layers in [1,2,4,6,8]. Generally, we used 0.5 as dropout. (If the paper of a model specified dropout to a specific value, we used the value following the paper.) As we can see, our model generally has a small hyperparameter search space comparable to GAT (when not using $\mathcal{L}_{dis}$). Although ED-HNN and ED-HNNII had good performance on standard hypergraph benchmark datasets, they had to rely on very large hyperparameter search space.

Table 7: Hyperparameter search space in cancer subtype classification task. † : MLP layers used in AllDeepSets, AllSetTransforer, ED-HNN, ED-HNNII

| models | head (factor) | ♯ MLP layer † | $\lambda$ for $\mathcal{L}_{dis}$ | ♯ Total |
|---|---|---|---|---|
| HGNN | 1 | - | - | 24 |
| HCHA | 1 | - | - | 24 |
| HNHN | 1 | - | - | 24 |
| UniGCNII | 1 | - | - | 24 |
| AllDeepSets | 1 | 1,2 | - | 48 |
| AllSetTransformer | 1,2,4,8 | 1,2 | - | 192 |
| HyperGAT | 1,2,4,8 | - | - | 96 |
| SHINE | 1,2,4,8 | - | - | 96 |
| HSDN | 1,2,4,8 | - | 0.0001, 0.0005, 0.001, 0.005, 0.01, 0.05, 0.1 | 672 |
| ED-HNN | 1 | $[0,1] \times [1] \times [0,1]$ | - | 96 |
| ED-HNNII | 1 | $[0,1] \times [1] \times [0,1]$ | - | 96 |
| Natural-HNN | 1,2,4,8 | - | - | 96 |

For cancer subtype classification tasks, we used [16, 32, 64] as the hidden dimension and [0.1, 0.01, 0.001, 0.0001] as learning rate. For weight decay, we used [0, 1e-5]. We fixed the number of layers to 2 unless the paper of a model fixed the number of layers to a specific number. During training, we set 50 as the batch size. Generally, we used 0.5 as dropout. (If the paper of a model specified dropout to a specific value, we used the value following the paper.) Since we fixed the number of classifiers to 1, the hyperparameter search space of some models are largely reduced when compared to the node classification task. For ED-HNN and ED-HNNII, we reduced the search space of the number of MLPs since it took too much time to get the results.

## C  Standard Hypergraph Benchmark dataset

We performed experiments with standard hypergraph benchmark dataset to check whether Natural-HNN can be applied to the **datasets that are not verified to have meaningful context of interactions. Considering how hyperedges were created for benchmark datasets, it is not likely that those datasets contain meaningful or task related interaction contexts.** In co-citation and co-authorship networks, for example, hyperedges are created by simply connecting all documents cited by a paper or written by an author. Citations between a pair of papers might have context that is related to a reason for citation, however, it is hard to expect that a group of documents (papers) cited by a paper creates a special meaning or have a special context. Even if we assume that hyperedges in co-citation networks contain interaction context, it is still not clear how these interaction contexts are related to the labels of nodes. It is also hard to expect interaction context in co-authorship networks for a similar reason. Thus, **the benchmark dataset experiment will verify whether Natural-HNN can be applied to the datasets where informativeness of interaction context is not known.**

For the node classification task with standard hypergraph benchmark datasets, we randomly split the data into 50%/25%/25% for training/validation/test set. We measured average and standard deviation of the performances for 10 different data splits. The hyperparameter search space is provided in Appendix

### C.1  Statistics : Standard Hypergraph Benchmark Dataset

Cocitaion networks and coauthor networks are adopted from (Yadati et al., 2019). The node features are bag-of-words representation of each documents. NTU2012 and ModelNet40 dataset is computer vision and graphics datasets where features are generated by applying GVCNN(Feng et al., 2018) and MVCNN(Su et al., 2015). Node feature of 20Newsgroups are generated by TF-IDF representations of news. The statistics of standard benchmark dataset is given in Table 8. Homophily ratio was calculated after converting hypergraph into a graph with clique expansion (CE)(Sun et al., 2008) following the method described in the other work (Wang et al., 2022b).

Table 8: Dataset statistics of standard hypergraph benchmark dataset

|  | Cora | Citeseer | Pubmed | Cora-CA | DBLP-CA | NTU2012 | ModelNet40 | 20Newsgroups |
|---|---|---|---|---|---|---|---|---|
| # nodes | 2708 | 3312 | 19717 | 2708 | 41302 | 2012 | 12311 | 16242 |
| # edge | 1579 | 1079 | 7963 | 1072 | 22363 | 2012 | 12311 | 16242 |
| # feature | 1433 | 3703 | 500 | 1433 | 1425 | 100 | 100 | 100 |
| # classes | 7 | 6 | 3 | 7 | 6 | 67 | 40 | 4 |
| avg. $|e|$ | 3.03 | 3.200 | 4.349 | 4.277 | 4.452 | 5 | 5 | 654.51 |
| CE Homophily | 0.897 | 0.893 | 0.952 | 0.803 | 0.869 | 0.753 | 0.853 | 0.461 |

### C.2  Node Classification on Benchmark Datasets

Table 9: Model performance on standard hypergraph benchmark datasets (Accuracy). Top three models are colored by First, Second, Third. † : the variant of the model using multihead attention. ⋆ : the variant of the model using $\mathcal{L}_{reg}$ defined in SHINE(Luo, 2022).

| Method | Cora | Citeseer | Pubmed | Cora-CA | DBLP-CA | NTU2012 | ModelNet40 | 20Newsgroups |
|---|---|---|---|---|---|---|---|---|
| HGNN | 79.453 ± 1.003 | 73.092 ± 1.582 | 87.336 ± 0.443 | 83.383 ± 1.028 | 91.410 ± 0.365 | 88.350 ± 1.082 | 95.567 ± 0.411 | 81.246 ± 0.435 |
| HCHA | 79.276 ± 1.158 | 73.693 ± 1.687 | 87.230 ± 0.511 | 83.191 ± 0.868 | 91.358 ± 0.374 | 88.270 ± 1.304 | 94.703 ± 0.283 | 81.189 ± 0.397 |
| HNHN | 76.765 ± 1.560 | 72.524 ± 1.570 | 87.237 ± 0.523 | 77.480 ± 0.932 | 86.927 ± 0.346 | 88.489 ± 0.878 | 97.811 ± 0.231 | 81.059 ± 0.485 |
| UniGCNII | 79.498 ± 1.508 | 73.514 ± 2.107 | 88.124 ± 0.376 | 83.840 ± 0.693 | 91.728 ± 0.225 | 89.245 ± 0.882 | 97.243 ± 0.334 | 81.687 ± 0.452 |
| AllDeepSets | 79.306 ± 1.627 | 72.959 ± 1.795 | 89.418 ± 0.360 | 84.594 ± 0.793 | 91.594 ± 0.308 | 88.847 ± 0.984 | 97.532 ± 0.185 | 81.721 ± 0.653 |
| AllSetTransformer | 79.749 ± 1.620 | 73.140 ± 1.804 | 88.667 ± 0.388 | 84.786 ± 0.690 | 91.593 ± 0.309 | 89.404 ± 1.074 | 98.217 ± 0.138 | 81.783 ± 0.569 |
| HyperGAT | 55.908 ± 4.128 | 41.751 ± 1.814 | 48.191 ± 0.443 | 73.560 ± 1.829 | 90.292 ± 0.468 | 83.857 ± 1.490 | 92.465 ± 0.387 | 80.997 ± 0.390 |
| HyperGAT† | 58.183 ± 2.079 | 42.246 ± 1.874 | 48.389 ± 0.426 | 73.752 ± 1.508 | 90.394 ± 0.362 | 85.467 ± 1.876 | 92.481 ± 0.463 | 81.083 ± 0.374 |
| SHINE | 57.755 ± 3.198 | 41.413 ± 0.680 | 48.576 ± 0.455 | 75.037 ± 1.912 | 90.759 ± 0.292 | 87.256 ± 1.393 | 93.803 ± 0.395 | 81.061 ± 0.632 |
| SHINE† | 56.307 ± 4.452 | 41.763 ± 0.693 | 48.576 ± 0.433 | 75.613 ± 1.508 | 90.697 ± 0.329 | 87.157 ± 1.426 | 93.878 ± 0.332 | 81.239 ± 0.459 |
| SHINE⋆ | 58.818 ± 1.591 | 41.413 ± 1.563 | 46.682 ± 1.177 | 74.623 ± 1.444 | 61.507 ± 12.169 | 61.451 ± 2.399 | 89.406 ± 0.775 | 61.492 ± 12.666 |
| SHINE†⋆ | 58.065 ± 1.616 | 41.123 ± 1.707 | 43.619 ± 1.402 | 73.087 ± 1.077 | 36.215 ± 17.676 | 70.835 ± 23.388 | 75.956 ± 23.688 | 56.452 ± 13.043 |
| HSDN | 76.632 ± 1.509 | 71.824 ± 1.779 | 87.193 ± 0.323 | 81.595 ± 1.011 | 90.229 ± 0.242 | 89.722 ± 1.196 | 83.439 ± 1.204 | 81.372 ± 0.435 |
| ED-HNN | 80.635 ± 1.670 | 73.696 ± 1.992 | 88.911 ± 0.410 | 85.480 ± 0.828 | 92.151 ± 0.291 | 87.594 ± 0.811 | 97.999 ± 0.199 | 81.608 ± 0.695 |
| ED-HNNII | 78.951 ± 1.445 | 72.524 ± 1.682 | 79.355 ± 0.953 | 83.693 ± 0.839 | 91.702 ± 0.325 | 86.223 ± 0.958 | 95.749 ± 0.335 | 80.150 ± 0.753 |
| Natural-HNN (ours) | 80.709 ± 1.635 | 73.285 ± 1.742 | 87.136 ± 0.450 | 84.993 ± 0.491 | 90.961 ± 0.137 | 89.900 ± 1.017 | 98.558 ± 0.295 | 81.734 ± 0.745 |
| Natural-HNN (ours + $\mathcal{L}_{dis}$) | 80.739 ± 1.570 | 73.551 ± 1.964 | 88.475 ± 0.466 | 85.081 ± 0.583 | 91.032 ± 0.179 | 90.060 ± 1.565 | 98.584 ± 0.254 | 81.827 ± 0.695 |

Table 9 summarizes the node classification performance in standard hypergraph benchmark datasets. We have the following observations: **1)** Our model generally performs well on various datasets by taking the first or second place in terms of accuracy. In the case of Citeseer and Cora-CA, the

performance of our model is comparable to the best performing model. The results indicate that our model can be applied to various circumstances, even when the context variety of hyperedges is not guaranteed. **2)** Attention-based models (i.e., AllSetTransformer, SHINE, and HyperGAT) and disentangle-based model (i.e., HSDN) generally perform similar to or worse than convolution-based models (i.e., HGNN, HCHA, HNHN, UniGCNII) and AllDeepSets (which also does not have heads or factors) on Citeseer, Pubmed and DBLP-CA. Through the results, we can guess that those datasets do not contain various interaction contexts that is helpful for the model performance. This can also be a reason why our model does not perform well on those datasets as much as on other datasets.

## C.3 Training with only 5% of data

Table 10: Model performance on standard hypergraph benchmark datasets (Accuracy) trained with only 5% of data

| Method | Cora | Citeseer | Pubmed | Cora-CA | DBLP-CA | NTU2012 | ModelNet40 | 20Newsgroups |
|---|---|---|---|---|---|---|---|---|
| HGNN | 66.773 ± 2.806 | 61.445 ± 2.465 | 81.161 ± 0.531 | 71.548 ± 2.652 | 89.689 ± 0.384 | 58.884 ± 5.045 | 94.795 ± 0.381 | 79.690 ± 0.675 |
| HCHA | 67.403 ± 2.865 | 61.600 ± 2.279 | 81.135 ± 0.549 | 71.379 ± 2.465 | 89.689 ± 0.274 | 59.032 ± 5.083 | 93.939 ± 0.448 | 79.596 ± 0.652 |
| HNHN | 58.272 ± 1.970 | 58.473 ± 5.296 | 79.793 ± 0.804 | 58.831 ± 2.399 | 82.855 ± 0.499 | 58.737 ± 5.344 | 96.845 ± 0.382 | 78.456 ± 0.602 |
| UniGCNII | 68.212 ± 2.559 | 63.600 ± 1.203 | 83.024 ± 0.820 | 70.799 ± 2.606 | 88.751 ± 0.281 | 60.255 ± 5.022 | 96.584 ± 0.248 | 79.061 ± 0.506 |
| AllDeepSets | 65.694 ± 2.306 | 61.388 ± 4.012 | 84.485 ± 0.647 | 71.319 ± 2.964 | 59.689 ± 0.296 | 59.892 ± 4.833 | 96.055 ± 0.286 | 78.868 ± 0.534 |
| AllSetTransformer | 65.914 ± 2.155 | 62.506 ± 1.720 | 82.942 ± 0.491 | 71.249 ± 2.796 | 89.665 ± 0.216 | 60.444 ± 5.204 | 96.608 ± 0.291 | 79.409 ± 0.590 |
| HSDN | 58.332 ± 2.882 | 57.812 ± 1.808 | 80.195 ± 0.45 | 64.845 ± 4.025 | 87.636 ± 0.243 | 51.949 ± 17.016 | 97.159 ± 0.179 | 79.406 ± 0.594 |
| ED-HNN | 66.433 ± 2.824 | 61.759 ± 2.296 | 82.348 ± 0.559 | 69.809 ± 2.569 | 90.039 ± 0.342 | 57.984 ± 6.477 | 96.698 ± 0.265 | 78.386 ± 0.542 |
| Natural-HNN (ours) | 67.343 ± 1.837 | 62.620 ± 2.277 | 82.393 ± 0.467 | 70.809 ± 2.789 | 88.700 ± 0.251 | 60.511 ± 5.338 | 98.031 ± 0.196 | 79.329 ± 0.666 |
| Natural-HNN (ours + $\mathcal{L}_{dis}$) | 67.393 ± 1.938 | 62.694 ± 2.218 | 82.838 ± 0.609 | 70.909 ± 3.439 | 88.906 ± 0.204 | 61.384 ± 4.570 | 98.141 ± 0.116 | 79.431 ± 0.552 |

To check the generalization power of our model, we performed an experiment of training with only 5% of data. Following the split ratio of HGNN for Cora dataset, we trained with 5% of data, validated with 18.5% and tested with 37% of data. Table 10 shows the result. We have the following observations: **1)** The performance of Natural-HNN tends to be similar or slightly better than convolution-based models. This shows that Natural-HNN has good generalization power that is comparable to convolution-based methods. **2)** Our model performs better than recently introduced model, ED-HNN. Even if ED-HNN has much larger hyperparameter search space, Natural-HNN performs better due to generalization power.

## C.4 Comparison with Sheaf Hypergraph Networks

Sheaf hypergraph networks (Duta et al., 2024) extends idea of Sheaf based graph learning methods (Hansen & Gebhart, 2020; Bodnar et al., 2022). Sheaf hypergraph networks can be explained as a process of minimizing sheaf Dirichlet energy of signal on a hypergraph. It can be also explained as reaching *apparent* consensus, the consensus of expressed opinions. Compared to Natural-HNN which creates consensus on the discussion topic of each hyperedge, sheaf hypergraph network is more flexible and can have stronger expressive power. Since it can assign different transformations for every node-to-hyperedge message passing, it also has the potential to perform context-dependent message passing and can handle more complex interactions.

As the hyperparameter search space of sheaf hypergraph networks is too large, we compared our result with their official performance in their paper (Duta et al., 2024). Only for this experiment, we additionally tuned the hyperparameter $\beta$ ( 0.1 - 0.9) and the dropout rate (0.1 - 0.9). The hyperparameters of sheaf hypergraph networks include : stack dimension (1 - 8), learning rate (0.1, 0.01, 0.001), weight decay (0, 1e-5), dropout rate (0.1 - 0.9), number of layers (1 - 8). Additionally, sheaf neural networks have options that needs to be selected : weight sharing among layers (or not), type of normalization (degree base or sheaf-based ), type of Laplacian (symmetric or asymmetric), way of initializing hyperedge features (4 methods are proposed), non linear activation function (sigmoid or tanh), weight $W_1$ as learnable parameter or not (Identity matrix). The result is provided in Table 11.

In Table 11, we can observe that Natural-HNN always achieves the best performance except for the Citeseer dataset. The result is quite impressive in two aspects. **1)** Natural-HNN achieved better performance even with less expressive power compared to sheaf hypergraph networks. Note that sheaf hypergraph networks have stronger expressive power by allowing each node-to-hyperedge message passing to use different transformation (while Natural-HNN can select only one of $K$ transformations (MLP)). **2)** Natural-HNN got outstanding performance even with smaller hyperparameter search space. Note that Natural-HNN with extended hyperparameter search space still has much smaller hyperparameter search space compared to sheaf hypergraph network. This result migth be attributed

Table 11: Model performance on standard hypergraph benchmark datasets (Accuracy). $\beta$ denotes that we tuned the hyperparameter $\beta$. The best performance is colored in red. The $2^{nd}$ placed is color in blue.

| Method | Cora | Citeseer | Pubmed | Cora-CA | DBLP-CA | NTU2012 | ModelNet40 | 20Newsgroups |
|---|---|---|---|---|---|---|---|---|
| Diag-SheafHyperGCN | 80.06 ± 1.12 | 73.27 ± 0.50 | 87.09 ± 0.71 | 83.26 ± 1.20 | 90.83 ± 0.23 | - | - | - |
| LR-SheafHyperGCN | 78.70 ± 1.14 | 72.14 ± 1.09 | 86.99 ± 0.39 | 82.61 ± 1.28 | 90.84 ± 0.29 | - | - | - |
| Gen-SheafHyperGCN | 79.13 ± 0.85 | 72.54 ± 2.30 | 86.90 ± 0.46 | 82.54 ± 2.08 | 90.57 ± 0.40 | - | - | - |
| Diag-SheafHyperGNN | **81.30** ± 1.70 | **74.71** ± 1.23 | 87.68 ± 0.60 | **85.52** ± 1.28 | 91.59 ± 0.24 | - | - | - |
| LR-SheafHyperGNN | 76.65 ± 1.41 | 74.05 ± 1.34 | 87.09 ± 0.25 | 77.05 ± 1.00 | 85.13 ± 0.29 | - | - | - |
| Gen-SheafHyperGNN | 76.82 ± 1.32 | **74.24** ± 1.05 | 87.35 ± 0.34 | 77.12 ± 1.14 | 84.99 ± 0.39 | - | - | - |
| Natural-HNN (base) | 80.71 ± 1.64 | 73.29 ± 1.74 | 87.14 ± 0.45 | 84.99 ± 0.49 | 90.96 ± 0.14 | 89.90 ± 1.02 | 98.56 ± 0.30 | 81.73 ± 0.75 |
| Natural-HNN (base + $\mathcal{L}_{dis}$) | 80.74 ± 1.57 | 73.55 ± 1.96 | **88.48** ± 0.47 | 85.08 ± 0.58 | 91.03 ± 0.18 | 90.06 ± 1.57 | **98.58** ± 0.25 | 81.83 ± 0.70 |
| Natural-HNN (base + $\beta$) | 80.83 ± 1.37 | 73.33 ± 0.94 | 87.17 ± 0.35 | 85.20 ± 0.52 | **91.72** ± 0.16 | **90.34** ± 1.02 | **98.58** ± 0.22 | 81.79 ± 0.79 |
| Natural-HNN (base + dropout) | 80.86 ± 1.28 | 73.36 ± 1.31 | 87.63 ± 0.45 | 85.05 ± 0.51 | 91.06 ± 0.13 | 90.06 ± 1.57 | 98.56 ± 0.22 | **81.94** ± 0.62 |
| Natural-HNN (base + $\mathcal{L}_{dis}$ + $\beta$ + dropout) | **81.30** ± 1.32 | 74.06 ± 1.34 | **88.75** ± 0.51 | **85.58** ± 0.77 | **91.91** ± 0.19 | **90.42** ± 0.92 | **98.63** ± 0.23 | **82.08** ± 0.74 |

to the fact that Natural-HNN made good trade-off between generalizability and expressivity of the model.

# D  SYNTHETIC DATASET EXPERIMENTS

**The purpose of experiment.** The purpose of the synthetic dataset experiment is to compare the ability of capturing the interaction context. As already described in Appendix A.8, it is nearly impossible to train and validate the ability of sheaf-based methods on cancer subtype dataset. Also, as described before, we need synthetic dataset to ensure that there are no contexts other than context of interaction so that transformation matrics of sheaf-based methods only depend of the interaction context.

**Experiment Setting.** We randomly split the data into 50%/25%/25% for training/validation/test set. We measured average and standard deviation of the performances for 10 different data splits. To simplify the hyperparameter search space, we fixed learning rate to be one of 0.01 and 0.001, weight decay as 0, hidden dimension to be 64 and dropout as 0.5. Since dataset generation process in Appendix A.8 used 1 layer of propagation, all models in this experiment use 1 layer. For sheaf based methods, we fixed the stalk dimension to 8, set initial hyperedge feature as average of node features, activation function as tanh to reduce hyperparameter search space. For the methods of normalization, we used symmetric degree normalization and assymetric degree normalization. For LowRankSheafs, we set rank as 2. For SheafHyperGCN, we used mediators. For experiment, we used official implementation of the paper.

**Notation for sheaf-based methods.** Throught Appendix D, 'sym' denotes symmetric degree normalization and 'assym' denotes assymetric degree normalization. 'Gen' denotes GeneralSheafs, 'LR' denotes LowRankSheafs and 'Diag' denotes DiagSheafs.

## D.1  PERFORMANCE

Table 12: Disentangle-based model performance on a synthetic dataset. Performances are measured by varying the number of factors. $\mathcal{L}_{dis}$ is the factor discrimination loss introduced in the Section 4.4.

| Method | number of factors : 2 | number of factors : 4 | number of factors : 8 |
|---|---|---|---|
| HSDN (without $\mathcal{L}_{dis}$) | 71.363 ± 1.543 | 73.425 ± 1.465 | 72.838 ± 0.988 |
| HSDN (with $\mathcal{L}_{dis}$) | 72.150 ± 1.616 | 73.600 ± 1.556 | 72.850 ± 1.057 |
| Natural-HNN (without $\mathcal{L}_{dis}$) | 75.688 ± 1.350 | 75.813 ± 1.524 | 75.863 ± 1.182 |
| Natural-HNN (with $\mathcal{L}_{dis}$) | 76.425 ± 1.740 | 76.875 ± 1.318 | 77.225 ± 1.263 |

Table 13: Sheaf-based model performance on a synthetic dataset. Performances are measured by varying the type of restriction maps as well as the type of normalization (symmetric or assymetric).

| Method | GeneralSheafs (Gen) | LowRankSheafs (LR) | DiagSheafs (Diag) |
|---|---|---|---|
| SheafHyperGNN (asymmetric) | 75.538 ± 1.334 | 75.513 ± 1.269 | 75.713 ± 1.371 |
| SheafHyperGNN (symmetric) | 75.388 ± 1.171 | 75.325 ± 1.299 | 75.463 ± 1.495 |
| SheafHyperGCN (asymmetric) | 74.863 ± 1.171 | 74.638 ± 1.121 | 74.600 ± 1.179 |
| SheafHyperGCN (symmetric) | 74.713 ± 1.229 | 74.738 ± 1.039 | 74.713 ± 1.185 |

Although the main purpose of synthetic dataset experiment is to validate the ability of capturing interaction context, we report the performance of sheaf-based methods, HSDN and Natural-HNN. Note that HGNN on this dataset showed the accuracy (%) of 58.075 ± 1.908.

**Comparison with HSDN.** Table 12 shows the result of HSDN and Natural-HNN on synthetic dataset with varying number of factors. We have the following observations : **1)** Natural-HNN generally performs better than HSDN. **2)** We can observe that Natural-HNN had the best performance when the number of factors for Natural-HNN matches the number of hyperedge types, which is 8 in our synthetic dataset. On the other hand, HSDN achieves its best performance when the number of factors was 4.

**Comparison with Sheaf-based methods.** Table 13 shows the result of sheaf-based methods. We have the following observations : **1)** Sheaf-based methods generally have similar performance regardless of normalization types and sheaf types. But SheafHyperGNN generally performs better than SheafHyperGCN. **2)** Natural-HNN slightly performs better than sheaf-based methods.

## D.2 SCALABILITY (TRAINING TIME)

We measured the time took for training 10 epochs. We measured the time 5 times each, and averaged them. The results for SheafHyperGNN and disentangle-based methods are provided in Table 14. The results for SheafHyperGCN and disentangle-based methods are provided in Table 15.

Table 14: Time took for training 10 epoch, measured in seconds. $d_s$ denotes stalk dimension for **SheafHyperGNN**. ♯ denotes 'number of'.

| $d_s$ or ♯ factors | Gen, sym | Gen, assym | LR, sym | LR, assym | Diag, sym | Diag, assym | HSDN | Natural-HNN |
|---|---|---|---|---|---|---|---|---|
| 2 | 3.764 ± 0.022 | 3.760 ± 0.021 | 3.756 ± 0.011 | 3.765 ± 0.019 | 2.056 ± 0.035 | 2.039 ± 0.018 | 0.052 ± 0.001 | 0.059 ± 0.003 |
| 4 | 15.561 ± 0.028 | 15.566 ± 0.052 | 15.628 ± 0.099 | 15.535 ± 0.020 | 4.783 ± 0.035 | 4.806 ± 0.014 | 0.058 ± 0.001 | 0.061 ± 0.001 |
| 8 | 60.719 ± 0.019 | 60.694 ± 0.035 | 60.609 ± 0.147 | 60.697 ± 0.013 | 10.458 ± 0.071 | 10.557 ± 0.065 | 0.063 ± 0.008 | 0.057 ± 0.001 |

In Table 14, we can observe the following : **1)** Increase of stalk dimension greatly increases the time required for training. It might be due to the fact that computations for creating Laplacian matrix from restriction maps greatly increases as stalk dimension increases. **2)** SheafHyperGNN still requires extremely long time for training with small synthetic hypergraph when compared to disentangle-based methods.

Table 15: Time took for training 10 epoch, measured in seconds. $d_s$ denotes stalk dimension for **SheafHyperGCN**. ♯ denotes 'number of'.

| $d_s$ or ♯ factors | Gen, sym | Gen, assym | LR, sym | LR, assym | Diag, sym | Diag, assym | HSDN | Natural-HNN |
|---|---|---|---|---|---|---|---|---|
| 2 | 9.480 ± 0.099 | 9.104 ± 0.082 | 9.736 ± 0.445 | 10.498 ± 0.094 | 9.761 ± 0.296 | 9.345 ± 0.159 | 0.052 ± 0.001 | 0.059 ± 0.003 |
| 4 | 9.788 ± 0.090 | 9.590 ± 0.070 | 9.713 ± 0.276 | 10.363 ± 0.309 | 9.925 ± 0.037 | 9.621 ± 0.145 | 0.058 ± 0.001 | 0.061 ± 0.001 |
| 8 | 10.498 ± 0.094 | 10.177 ± 0.103 | 10.312 ± 0.116 | 10.754 ± 0.479 | 9.886 ± 0.113 | 10.059 ± 0.094 | 0.063 ± 0.008 | 0.057 ± 0.001 |

In Table 15, we can observe similar result, however, the training time does not differ a lot by the dimension of stalks. It might be attributed from the fact that SheafHyperGCN converts each hyperedge to an edge[20] with mediators which reduces the amount of computation.

## D.3 CAPTURED CONTEXT RESULT

Since not all hyperedges are equally important for prediction, we selected top-k important hyperedges by calculating the influence of the existence of hyperedge with HNN used for generating synthetic hypergraph. We calculated the prediction results of the model for all nodes when a specific hyperedge exists and when the hyperedge the does not exist. Then we calculated the difference between the predictions to calculate how the existence of a hyperedge changes the prediction results. The change of prediction for the correct class (label) are added and the change of prediction for the wrong class are subtracted. Thus, if the value was positive, it is likely that the hyperedge was informative for the prediction. There were 285 hyperedges with positive influence in total. The hyperedge type 4 had 20 hyperedges that have a positive influence on prediction and it was the minimum value across all hyperedge types. In other words, all hyperedge types contain at least 20 hyperedges that have positive impact on prediction.

From Figure 12 to Figure 17, we have provided the transformation matrix similarity between hyperedge types for sheaf-based methods, HSDN and Natural-HNN by varying the number of top-k influential hyperedges. The number of influential hyperedges were selected to be 2 (Figure 12), 5(Figure 13), 10(Figure 14), 20 (Figure 15), 50 (Figure 16) and 300 (all hyperedges, Figure 17). As already described in Section 5.3, it is ideal to have strong diagonal values (dark blue). We have the following observations: **1)** Sheaf-based methods have similar heatmap regardless of sheaf types, normalization types, or model. **2)** Sheaf-based methods generally do not have strong diagonal values and have relatively higher similarity between different hyperedge types, showing that sheaf-based methods hardly captures interaction context. **3)** Natural-HNN with 8 factors show strong diagonal values from Figure 12 to 15, proving that Natural-HNN captures. **4)** When Natural-HNN uses 2 or 4 factors, the heatmap shows weaker diagonal lines or have relatively higher similarity between different hyperedge types while the model doesn't when using 8 factors. Considering that the synthetic hypergraph has 8 hyperedge types, we can conclude that Natural-HNN correctly captures interaction context. **5)** HSDN generally do not have strong diagonals. In other words, the similarity of transformation matrices between **the same hyperedge type** is also low. This shows that HSDN does not

---

[20]This is the reason why we could not measure the time in Section 5.4. Cancer subtype classification task uses hyperedge representations to predict graph labels. However, as SheafHyperGCN converts a hyperedge to an edge with mediators, it is hard to define hyperedge representation.

capture interaction context. **6)** As number of hyperedges used to calculate transformation similarities increases, the heatmap hardly shows strong diagonals and have relatively stronger similarities between different hyperedge types. This phenomenon is observed to all models. This is an obvious result since hyperedges that are not very informative for label prediction will not be reflected a lot during model training. However, we have another observation that disentangle-based methods tend to have relatively smaller similarities between hyperedge types when compared to sheaf-based methods. This also shows that sheaf-based methods do not effectively capture interaction context.

### D.4 OUR CONCLUSION FOR SHEAF-BASED METHODS

**Interaction Context.** Sheaf-based method has strong expressive power as it allows to have different transformation for every (node, hyperedge) pair. It also means that its design allows the model to capture interaction contexts. However, as can be seen through several experiments, sheaf-based methods do not effectively capture interaction context.

**Performance.** Since we could not experiment with cancer subtype classification task, we can only assume through the results in benchmark dataset (Appendix C.4) as well as synthetic dataset (Appendix D.1). Through the results, we can assume that Natural-HNN has slightly better performance compared to sheaf-based methods. However, as we have already seen in Section 5.4 and Appendix D.3, sheaf-based methods are not scalable (inefficient) and cannot be applied to many practical applications.

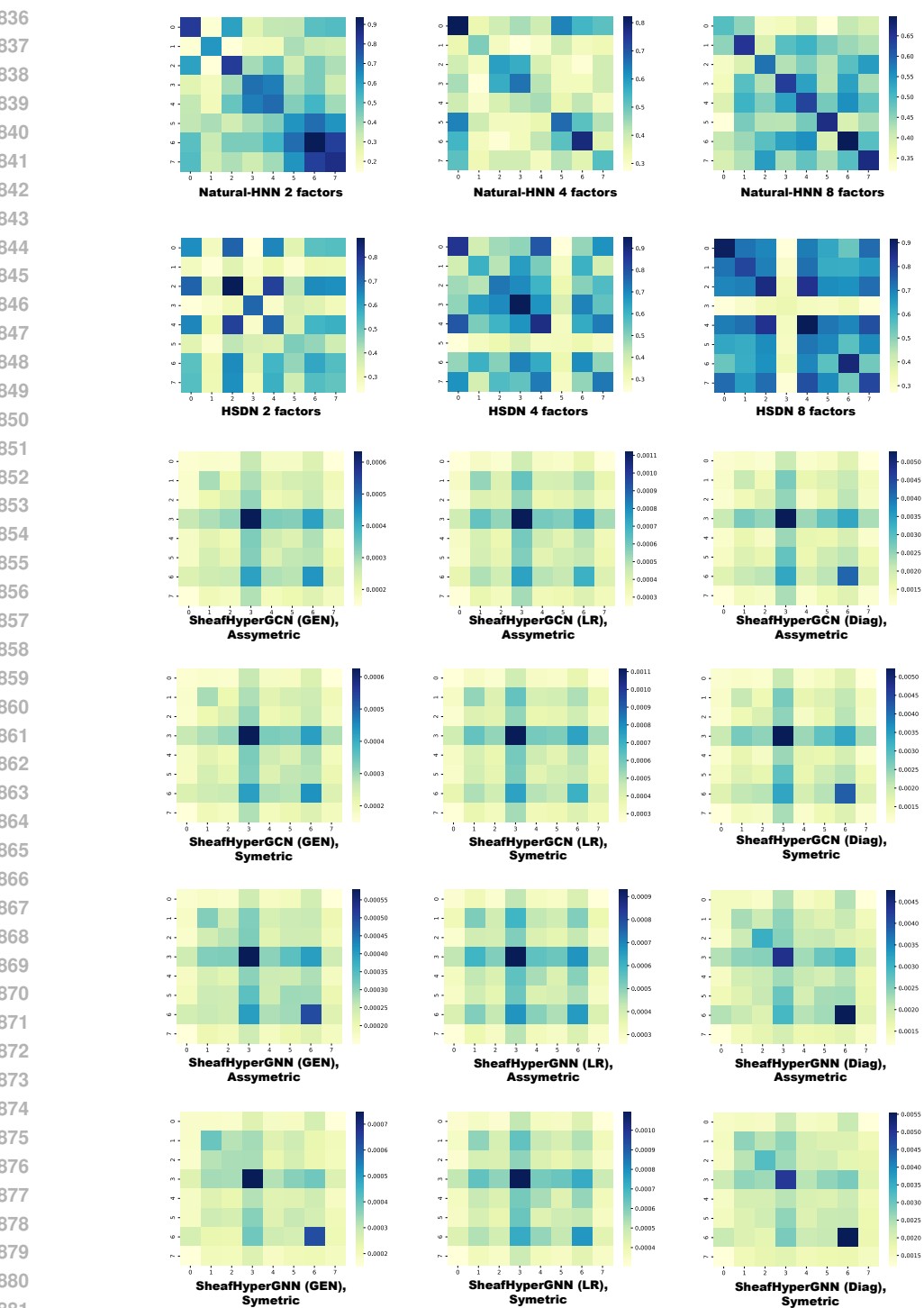

Figure 12: Transformation matrix similarities between different hyperedge types. The results are calculated with top-2 hyperedges based on the influence of hyperedge to label prediction. We can observe that Natural-HNN shows strong diagonal pattern.

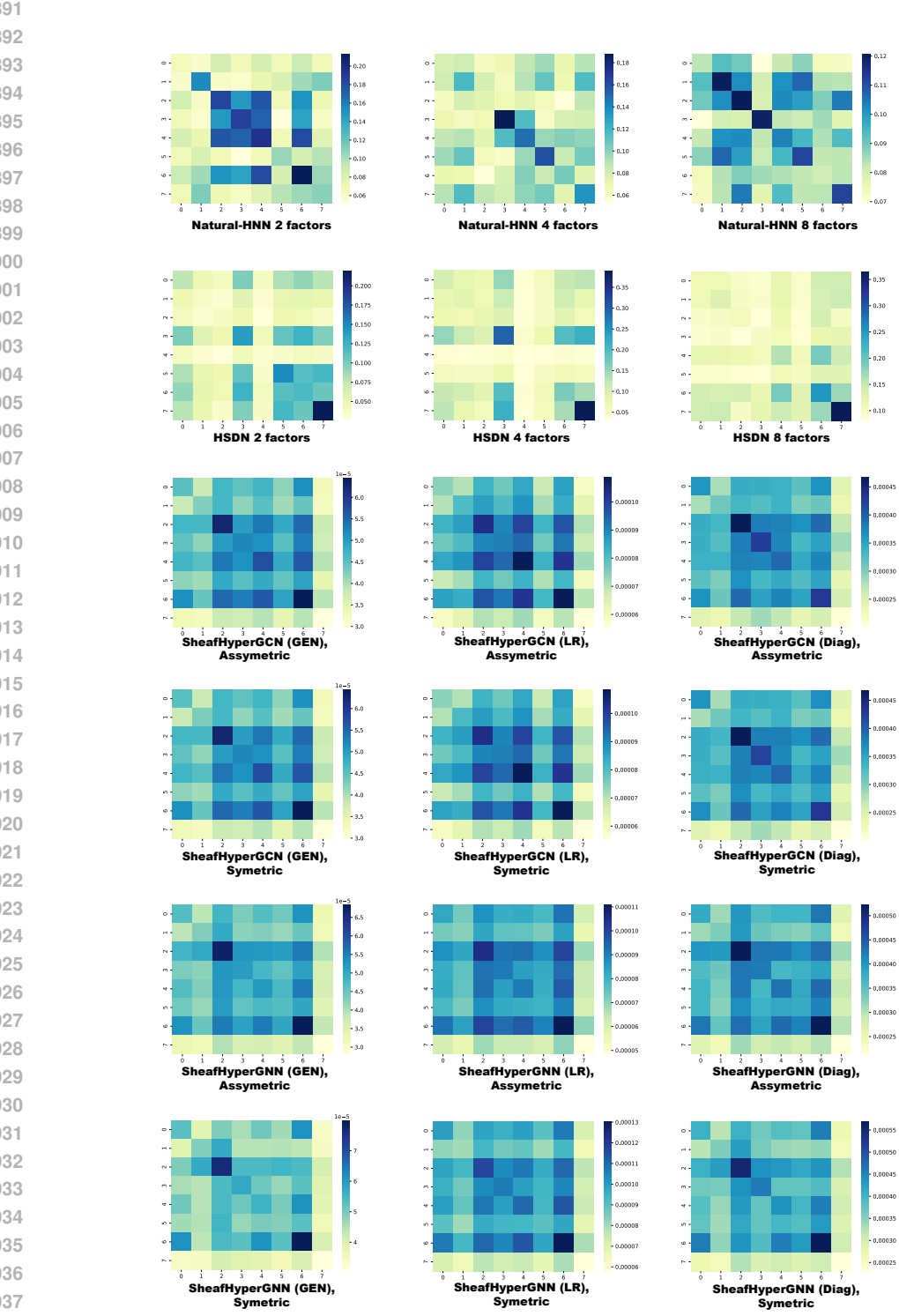

Figure 13: Transformation matrix similarities between different hyperedge types. The results are calculated with top-5 hyperedges based on the influence of hyperedge to label prediction. We can observe that Natural-HNN shows strong diagonal pattern while HSDN fails to. Sheaf-based methods generally shows strong similarity between different hyperedge types (non-diagonal).

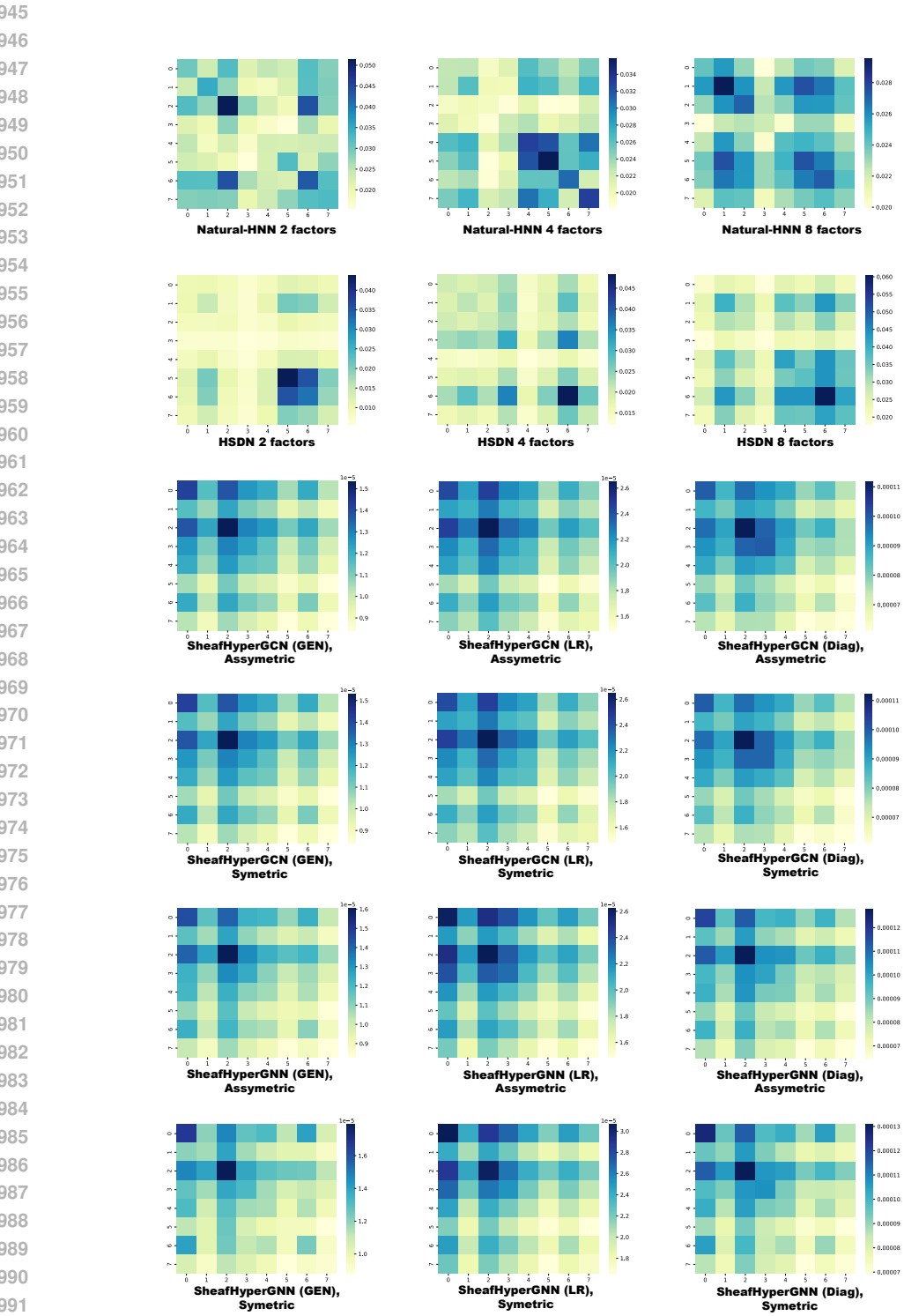

Figure 14: Transformation matrix similarities between different hyperedge types. The results are calculated with top-10 hyperedges based on the influence of hyperedge to label prediction. We can observe that Natural-HNN shows diagonal pattern while HSDN fails to. Sheaf-based methods generally shows strong similarity between different hyperedge types (non-diagonal).

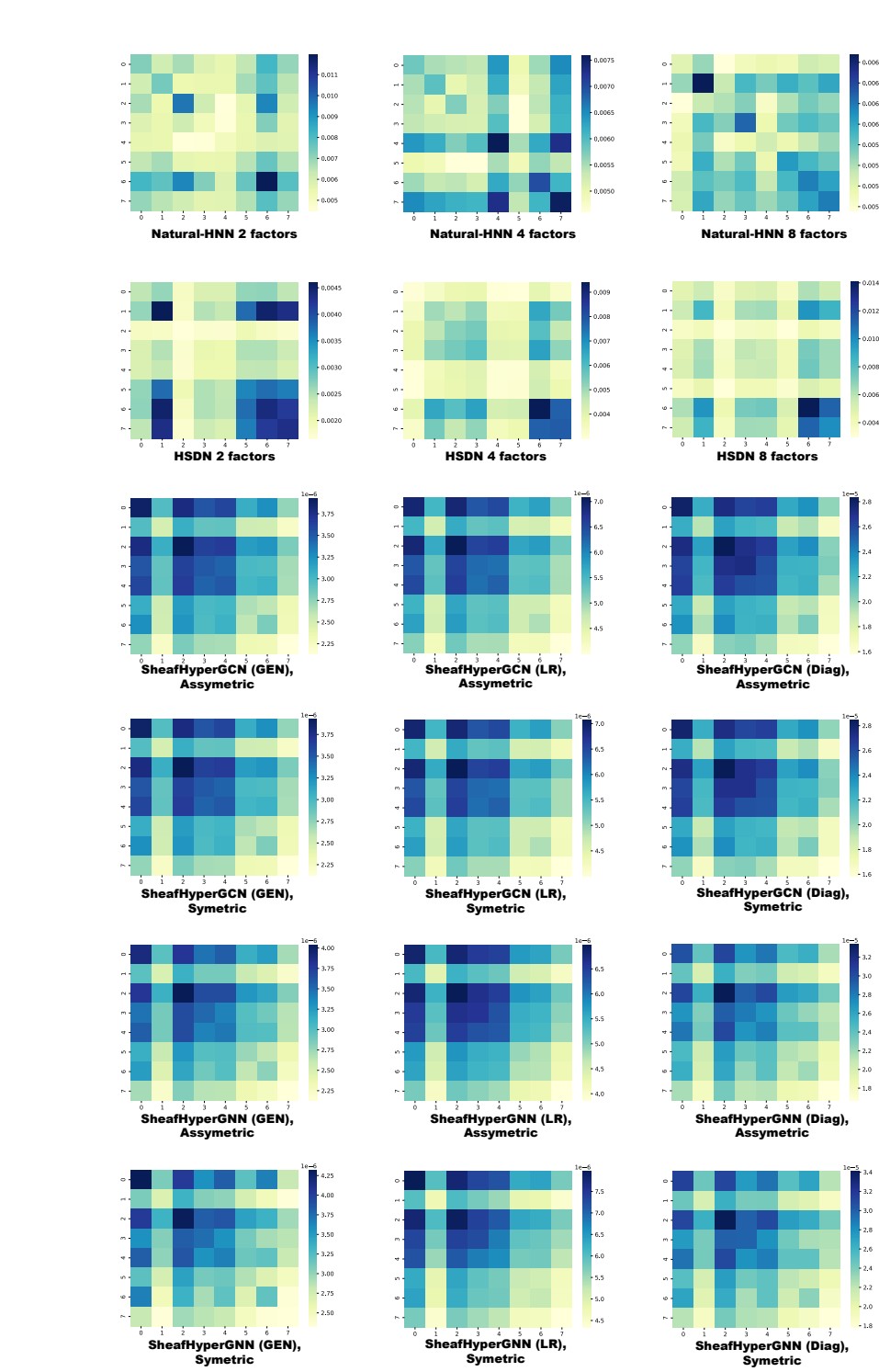

Figure 15: The results are calculated with top-20 hyperedges based on the influence of hyperedge to label prediction. This is the last figure that Natural-HNN shows diagonal pattern. This is because one of the hyperedge types have only 20 hyperedges that has positive influence for prediction.

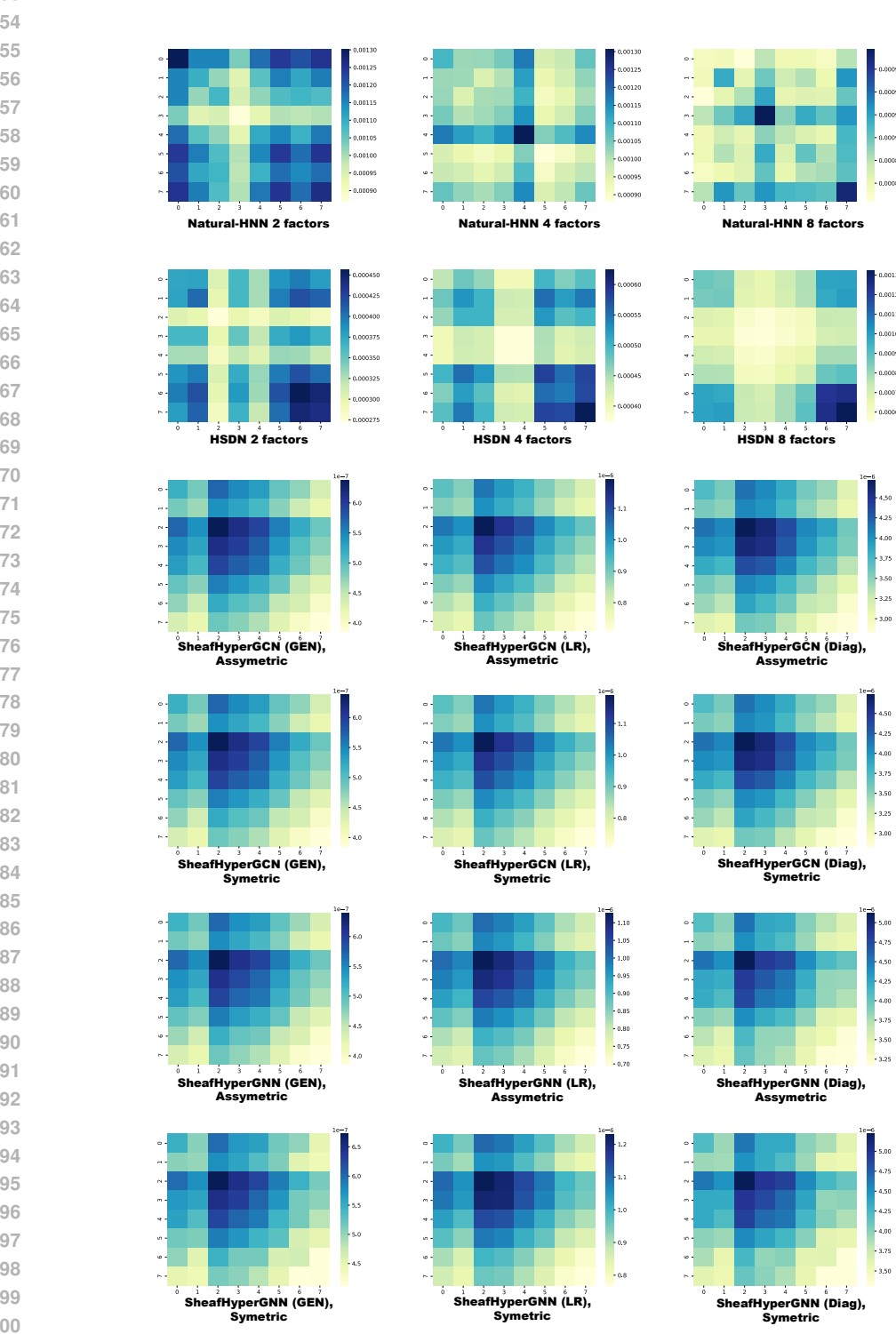

Figure 16: The results are calculated with top-50 hyperedges based on the influence of hyperedge to label prediction. All models fail to capture context. However, we can see that disentangle based models relatively have small similarities between different hyperedge types.

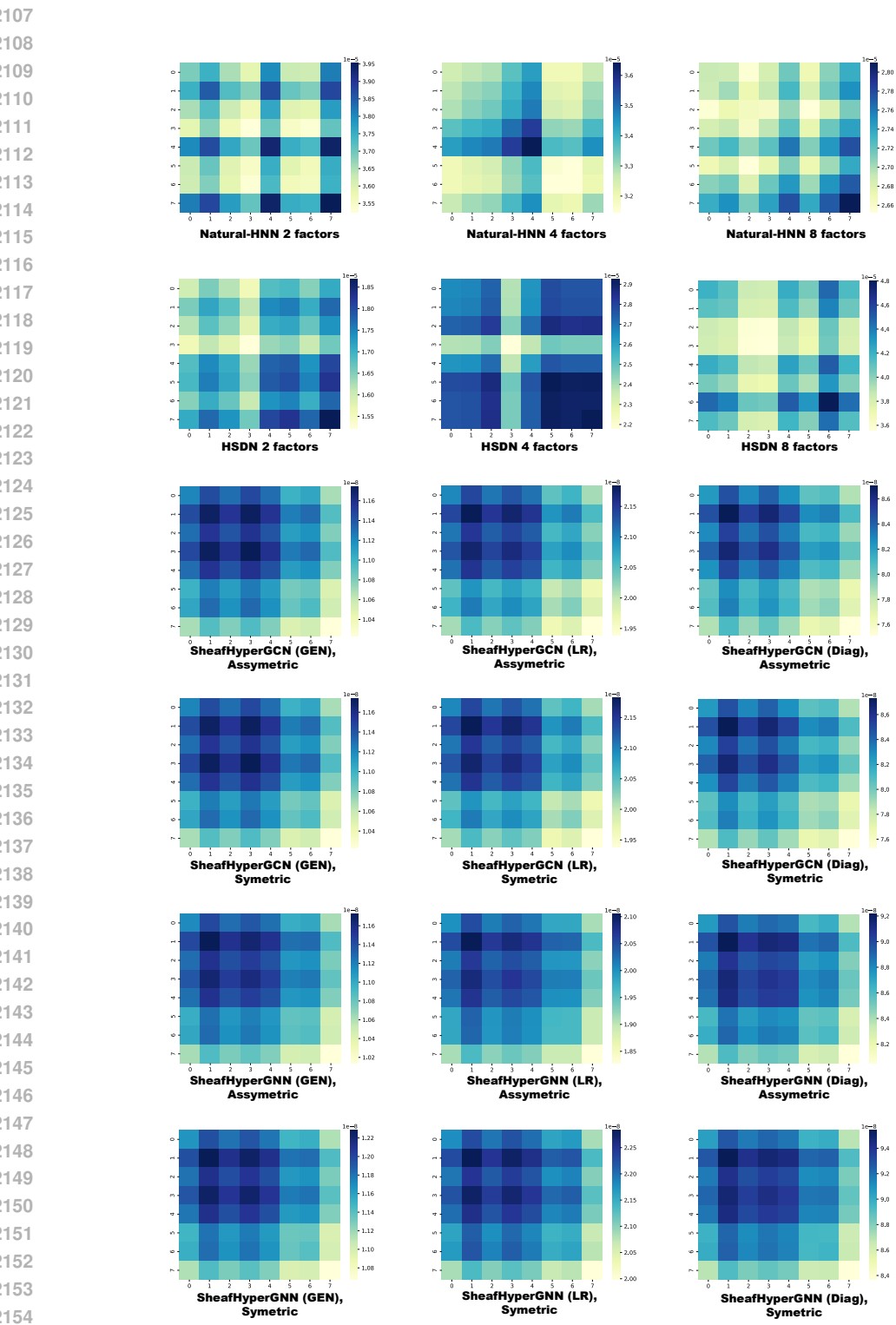

Figure 17: The results are calculated with top-300 hyperedges based on the influence of hyperedge to label prediction. All models fail to capture context. However, we can see that disentangle based models relatively have small similarities between different hyperedge types.

# E ABLATION STUDIES AND HYPERPARAMETER SENSITIVITY

## E.1 SELECTING ALTERNATIVE BRANCH

In Section 4, we used the representation earned from 'Disentangle-first Branch' ($h_{e_i}^k$) when creating final hyperedge factor representations ($\alpha_i^k h_{e_i}^k$). The experiment results below shows the result when using the other branch, 'Aggregation-first Branch' for creating final hyperedge factor representations ($\alpha_i^k \tilde{h}_{e_i}^k$). Table 16 shows the result for standard hypergraph benchmark dataset and Table 17 shows the result for cancer subtype classification task.

Table 16: Comparison of our model (first two rows) with alternative model that uses the other type of hyperedge factor representation (last two rows)

| Method | Cora | Citeseer | Pubmed | Cora-CA | DBLP-CA | NTU2012 | ModelNet40 | 20Newsgroups |
|---|---|---|---|---|---|---|---|---|
| Natural-HNN | 80.709 ± 1.635 | 73.285 ± 1.742 | 87.163 ± 0.450 | 84.993 ± 0.491 | 90.961 ± 0.137 | 89.900 ± 1.017 | 98.558 ± 0.295 | 81.734 ± 0.745 |
| Natural-HNN ($+\mathcal{L}_{dis}$) | 80.739 ± 1.570 | 73.551 ± 1.964 | 88.475 ± 0.466 | 85.081 ± 0.583 | 91.032 ± 0.179 | 90.060 ± 1.565 | 98.584 ± 0.254 | 81.827 ± 0.695 |
| Natural-HNN (other branch) | 80.650 ± 1.684 | 73.237 ± 1.678 | 87.137 ± 0.408 | 84.993 ± 0.434 | 90.968 ± 0.137 | 89.821 ± 0.847 | 98.557 ± 0.232 | 81.729 ± 0.701 |
| Natural-HNN (other branch + $\mathcal{L}_{dis}$) | 80.827 ± 1.157 | 73.575 ± 1.790 | 88.521 ± 0.424 | 85.081 ± 0.503 | 91.030 ± 0.178 | 90.060 ± 0.795 | 98.577 ± 0.227 | 81.837 ± 0.534 |

As we can see in Table 16, there is no big difference in the performance between using 'Disentangle-first Branch' and 'Aggregation-first Branch'.

Table 17: Comparison of our model (first row) with alternative model that uses the other type of hyperedge factor representation (last row).

| Method | BRCA | STAD | SARC | LGG | HNSC | CESC | KIPAN | NSCLC |
|---|---|---|---|---|---|---|---|---|
| Natural-HNN | 0.804 ± 0.036 | 0.659 ± 0.049 | 0.745 ± 0.045 | 0.707 ± 0.035 | 0.860 ± 0.042 | 0.881 ± 0.042 | 0.934 ± 0.010 | 0.962 ± 0.013 |
| Natural-HNN (other branch) | 0.797 ± 0.028 | 0.654 ± 0.041 | 0.747 ± 0.063 | 0.707 ± 0.033 | 0.863 ± 0.022 | 0.875 ± 0.051 | 0.934 ± 0.011 | 0.962 ± 0.012 |

As we can see in Table 17, there is no big difference in the performance between using 'Disentangle-first Branch' and 'Aggregation-first Branch'. The reason for this phenomenon is quite simple. We can consider the two cases: 1) when $h_{e_i}^k$ and $\tilde{h}_{e_i}^k$ are similar and 2) when they are largely different. **1)** When $h_{e_i}^k$ and $\tilde{h}_{e_i}^k$ are similar, the result will not differ a lot between using $h_{e_i}^k$ or $\tilde{h}_{e_i}^k$ as similar representations will be used. **2)** When $h_{e_i}^k$ and $\tilde{h}_{e_i}^k$ are largely different, the result will not be different a lot since relevance score $\alpha_i^k$ will be very small. In other words, $\alpha_i^k h_{e_i}^k - \alpha_i^k \tilde{h}_{e_i}^k = \alpha_i^k (h_{e_i}^k - \tilde{h}_{e_i}^k)$ will be very small for very small $\alpha_i^k$. This case means that the factor representation will not be reflected a lot during message passing since the representation is inconsistent (different result for two branches).

## E.2 NATURAL-HNN WITHOUT NATURALITY CONSTRAINT

We performed another ablation study to check whether naturality condition proposed in the paper is important part that contributes to the model. We created an ablation model that do not satisfies naturality condition by not reflecting relevance score $\alpha_i^k$ during message passing. The results for standard hypergraph benchmark dataset is provided in Table 18. The results for the cancer subtype classification task are provided in Table 19.

Table 18: Model performance on standard hypergraph benchmark datasets (Accuracy). The ablation model does not satisfy the naturality condition.

| Method | Cora | Citeseer | Pubmed | Cora-CA | DBLP-CA | NTU2012 | ModelNet40 | 20Newsgroups |
|---|---|---|---|---|---|---|---|---|
| Natural-HNN (ours) | 80.709 ± 1.635 | 73.285 ± 1.742 | 87.136 ± 0.450 | 84.993 ± 0.491 | 90.961 ± 0.137 | 89.900 ± 1.017 | 98.558 ± 0.295 | 81.734 ± 0.745 |
| Natural-HNN (ours + $\mathcal{L}_{dis}$) | 80.739 ± 1.570 | 73.551 ± 1.964 | 88.475 ± 0.466 | 85.081 ± 0.583 | 91.032 ± 0.179 | 90.060 ± 1.565 | 98.584 ± 0.254 | 81.827 ± 0.695 |
| Natural-HNN (ablation) | 80.220 ± 1.573 | 73.237 ± 1.745 | 87.121 ± 0.170 | 84.874 ± 0.424 | 90.896 ± 0.165 | 89.281 ± 0.718 | 98.144 ± 0.226 | 81.685 ± 0.675 |
| Natural-HNN (ablation + $\mathcal{L}_{dis}$) | 80.250 ± 1.555 | 73.392 ± 1.832 | 88.448 ± 0.407 | 85.022 ± 0.508 | 90.968 ± 0.169 | 89.679 ± 1.129 | 98.177 ± 0.216 | 81.783 ± 0.771 |

In Table 18, we can see that there is a slight to moderate level of performance gap between Natural-HNN and its ablation model. It is not a surprising result that there is not big difference between them since standard benchmark datasets do not seem to have informative interaction contexts related to the task (Appendix C).

In Table 19, we can observe that there is a big difference between Natural-HNN and its ablation model. Since interaction context matters in cancer subtype classification task, naturality condition seems to boost the performance by capturing interaction context.

Table 19: Model performance on cancer subtype classification task (Macro F1). The ablation model does not satisfy the naturality condition.

| Method | BRCA | STAD | SARC | LGG | HNSC | CESC | KIPAN | NSCLC |
|---|---|---|---|---|---|---|---|---|
| Natural-HNN* (ours) | 0.804 ± 0.036 | 0.659 ± 0.049 | 0.745 ± 0.045 | 0.707 ± 0.035 | 0.862 ± 0.045 | 0.881 ± 0.042 | 0.934 ± 0.010 | 0.962 ± 0.013 |
| Natural-HNN* (ablation) | 0.756 ± 0.031 | 0.605 ± 0.039 | 0.713 ± 0.071 | 0.692 ± 0.034 | 0.814 ± 0.037 | 0.852 ± 0.032 | 0.929 ± 0.016 | 0.958 ± 0.016 |

### E.3 HYPERPARAMETER ANALYSIS

Since Natural-HNN does not have many hyperparameters, we analyzed how performance changes by the number of factors. Table 20 shows the result for the standard hypergraph benchmark dataset. Table 21 shows the result for cancer subtype classification task. Note that the tables below show the result of Natural-HNN without $\mathcal{L}_{dis}$.

Table 20: Performance of Natural-HNN with a different number of factors. The best performances (reported in Table 9) are marked in red.

| number of factors | Cora | Citeseer | Pubmed | Cora-CA | DBLP-CA | NTU2012 | ModelNet40 | 20Newsgroups |
|---|---|---|---|---|---|---|---|---|
| 1 | 80.384 ± 1.820 | 73.133 ± 1.767 | 87.063 ± 0.373 | 84.934 ± 0.418 | 90.951 ± 0.139 | 89.622 ± 0.953 | 98.480 ± 0.310 | 81.684 ± 0.725 |
| 2 | 80.532 ± 1.638 | 73.285 ± 1.742 | 87.055 ± 0.401 | 84.904 ± 0.432 | 90.961 ± 0.137 | 89.622 ± 0.759 | 98.513 ± 0.272 | 81.734 ± 0.745 |
| 4 | 80.709 ± 1.652 | 73.188 ± 1.967 | 87.083 ± 0.450 | 84.993 ± 0.491 | 90.939 ± 0.151 | 89.821 ± 1.070 | 98.558 ± 0.295 | 81.635 ± 0.716 |
| 8 | 80.591 ± 1.673 | 73.237 ± 1.783 | 87.136 ± 0.450 | 84.934 ± 0.385 | 90.955 ± 0.131 | 89.900 ± 1.017 | 98.513 ± 0.286 | 81.660 ± 0.714 |

We have interesting observations when we analyze the result in Table 9 with Table 20. **1)** Natural-HNN did not perform well on Citeseer, Pubmed, and DBLP-CA datasets in Table 9. Except for the Pubmed dataset, Natural-HNN used 2 or fewer factors for its own best performance in Table 20. **2)** Natural-HNN showed good performance in remaining 5 datasets in Table 9. Except for 20Newsgroups dataset, Natural-HNN used 4 or more factors for its own best performance in Table 20. From these observations, we can conclude that Natural-HNN generally performed well when captured multiple factors. Also, since Natural-HNN did not have better performance when using more than 2 factors, we suspect that those two datasets do not have various interaction contexts that are beneficial for performance. The results of other attention-based (AllSetTransformer) or disentangle-based (HSDN) models in Table 9 also show a similar tendency. Those models have the potential to capture relational information, however, showed poor performance, even worse than some convolution-based models.

Table 21: Performance of Natural-HNN with different number of factors. The best performance (reported in Table 1) are marked in red.

| number of factors | BRCA | STAD | SARC | LGG | HNSC | CESC | KIPAN | NSCLC |
|---|---|---|---|---|---|---|---|---|
| 1 | 0.789 ± 0.036 | 0.630 ± 0.046 | 0.729 ± 0.055 | 0.695 ± 0.030 | 0.853 ± 0.047 | 0.869 ± 0.048 | 0.926 ± 0.013 | 0.956 ± 0.014 |
| 2 | 0.787 ± 0.038 | 0.642 ± 0.043 | 0.745 ± 0.045 | 0.707 ± 0.035 | 0.858 ± 0.031 | 0.867 ± 0.043 | 0.934 ± 0.010 | 0.959 ± 0.014 |
| 4 | 0.804 ± 0.036 | 0.659 ± 0.049 | 0.725 ± 0.048 | 0.689 ± 0.047 | 0.858 ± 0.036 | 0.881 ± 0.042 | 0.932 ± 0.013 | 0.962 ± 0.013 |
| 8 | 0.785 ± 0.027 | 0.637 ± 0.032 | 0.729 ± 0.058 | 0.691 ± 0.044 | 0.860 ± 0.042 | 0.878 ± 0.034 | 0.924 ± 0.016 | 0.961 ± 0.013 |

We have similar observations when comparing the result in Table 1 and Table 21. **1)** For SARC, LGG and KIPAN in Table 21, Natural-HNN had its best performance when using 2 factors. Except for SARC, Natural-HNN had relatively small increase in performanc in Table 1. **2)** For remaining datasets, Natural-HNN had its best performance when using 4 or more factors. Except for CESC, Natural-HNN had meaningful increase in performance. Thus, we can have similar conclusion that we had when comparing Table 9 and Table 20.

# F  ADDITIONAL EXPERIMENT RESULT

## F.1  COMPUTATIONAL COMPLEXITY

Let $d_i$ be the input embedding dimension, $d_o$ be the output embedding dimension, $K$ be number of factors. $N$ denotes number of nodes and $M$ denotes number of hyperedges, $E$ denotes the number of node($v$)-hyperedge($e$) pair $(v, e)$ satisfying $v \in e$. We will assume that $d_i \geqslant d_o$, $d_o \geqslant K$, $E \geqslant M$ and $E \geqslant N$.

The computational complexity of one layer of Natural-HNN can be calculated by the following:

- Aggregation-first Branch (aggregation + MLP): $O(Ed_i) + O(Md_id_o)$

- Disentangle-first Branch (MLP + aggregation): $O(Nd_id_o) + O(Ed_o)$

- Similarity ($\alpha$) calculation : $O(K(\frac{d_o^2}{K^2} + \frac{d_o}{K})) = O(\frac{d_o^2}{K})$

- propagation back to nodes : $O(KE + Ed_o) = O(Ed_o)$

- other calculations (concat, interpolation by $\beta$) : $O(Nd_o)$ Thus, total computational complexity becomes $O((M + N)d_id_o + E(d_i + d_o + 1) + Nd_o + \frac{d_o^2}{K}) = O((M + N)d_id_o + E(d_i + d_o))$

For HGNN with dimension $d_i \geqslant d_e \geqslant d_o$ ($d_e$ denotes dimension of hyperedge embedding), computational complexity becomes $O(E(d_i + d_e) + (Md_i + Nd_o)d_e)$. The computational complexity of HGNN and Natural-HNN differs only by constant times. It is not surprising since Natural-HNN is quite similar to HGNN but instead use two branches (only) during Node-to-Hyperedge propagation and use factor similarity calculation. Thus, Natural-HNN is as scalable as HGNN.

## F.2  SCALABILITY ANALYSIS (TRAINING TIME)

We measured the time took for training 1 epoch in BRCA dataset. We averaged the values after measuring 5 times each. Also, we conducted the experiment in two settings: one with 2 heads and 16-dimensional vector as hidden representation and the other with 8 heads and 64-dimensional vector as hidden representation. Note that convolution-based models, AllDeepSets and ED-HNN (II) use 1 head as they do not have an attention mechanism. The Table 22 and Table 23 shows the result of our model's scalability. We have the following observations: **1)** Our model is slower than convolution-based models and HSDN. Since convolution-based models use strong inductive bias with simple computations, they are naturally scalable than our model. HSDN took less time since they use only one message passing layer. **2)** Our model is much faster than all attention-based models. Thus, we can conclude that our model scales well with hypergraph and parameter size next to the convolution-based models.

Table 22: Time took for training 1 epoch on BRCA, measured in seconds. $d_c$ denotes hidden dimension. $\sharp$ denotes 'number of'.

| ($d_c$, $\sharp$ heads) | HGNN | HCHA | HNHN | UniGCNII | AllDeepSets | Natural-HNN |
|---|---|---|---|---|---|---|
| (16,2) | 0.217 ± 0.000 | 0.212 ± 0.000 | 0.117 ± 0.000 | 0.237 ± 0.000 | 1.195 ± 0.002 | 0.544 ± 0.001 |
| (64,8) | 0.831 ± 0.001 | 0.813 ± 0.000 | 0.426 ± 0.001 | 0.906 ± 0.001 | 2.463 ± 0.005 | 1.853 ± 0.002 |

Table 23: Time took for training 1 epoch on BRCA, measured in seconds. $d_c$ denotes hidden dimension. $\sharp$ denotes 'number of'.

| ($d_c$, $\sharp$ heads) | AllSetTransformer | HyperGAT | SHINE | HSDN | ED-HNN | ED-HNNII | Natural-HNN |
|---|---|---|---|---|---|---|---|
| (16,2) | 1.108 ± 0.002 | 0.711 ± 0.001 | 0.675 ± 0.001 | 0.289 ± 0.000 | 2.042 ± 0.003 | 3.852 ± 0.006 | 0.544 ± 0.001 |
| (64,8) | 2.671 ± 0.002 | 2.415 ± 0.003 | 2.204 ± 0.002 | 0.996 ± 0.000 | 3.558 ± 0.005 | 6.169 ± 0.014 | 1.853 ± 0.002 |

## F.3  GENERALIZATION POWER OF NATURAL-HNN

To check the generalization power of our model, we experimented with different training set split ratio, while maintaining the validation and test set ratio to 25%. From 50%, we gradually reduced training set proportion to 10% as shown in Figure 18. Figure 18 (a) and (b) are the result of measuring Macro-F1 scores and (c) and (d) are the result of measuring relative degradation of performance to the performance when trained with 50% (i.e., $(F1_{50} - F1_x)/F1_{50} \times 100\%$ where $F1_x$ denotes the Macro-F1 score when trained with x%.). Figure 18 (a) and (c) are the result in Cora-CA dataset, which is standard hypergraph benchmark, (b) and (d) are the result for BRCA dataset, which is

dataset used for cancer subtype classification task. The left figure in each Figure 18 (a,b,c,d) is the result of comparing ours (blue) and convolution of deepset based models. These baselines cannot perform context-dependent message passing. The right figure in each Figure 18 is the result of comparing ours (blue) and other baselines that have potential for context-dependent message passing (i.e. the models that can perform hyperedge dependent or node dependent message passing).

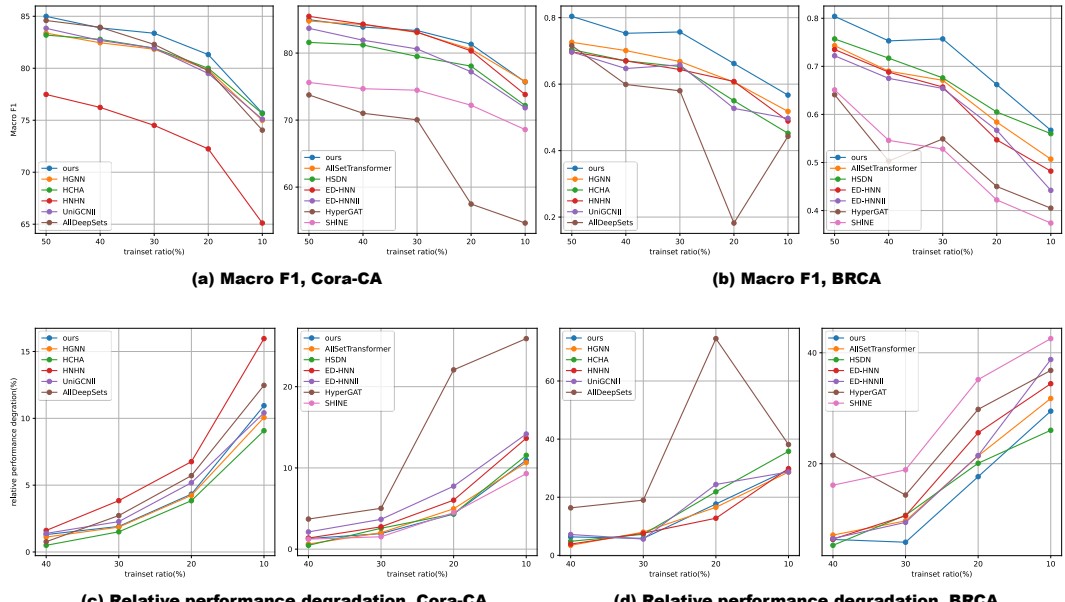

Figure 18: The performance of models when reducing training set proportion. First row shows Macro F1 score and the second row shows relative performance degradation compared to the performance when using 50% of dataset as training set. Ours (blue) maintains best Macro F1 score and small relative performance degradation on both Cora-CA and BRCA dataset.

We have the following observations : **1)** The degradation of performance for Natural-HNN was smaller when compared with most of the baselines in both Cora-CA and BRCA. Specifically, we can see that Natural-HNN has comparable result with convoluation based models in left figures of Figure 18 (c) and (d). Considering that convolutions based models have strong generalization performance due to their strong inductive bias, we can say that our model has good generalization power comparable to convolution based models. When compared with other baselinese in Figure 18 (b) and (d), we can observe that Natural-HNN had very small degradation in performance. In other words, Natural-HNN had nearly the smallest degradation when compared with models that have more expressive power than convolution based methods. We can consider our model had good generalization among baselines with more expressive powers. Specifically, in Figure 18 (d), Natural-HNN showed outstanding result in cancer dataset which has various context of interactions. This might be due to the fact that the inductive bias (context of interaction) that Natural-HNN used matched the actual data characteristics.

**2)** Natural-HNN had the best Macro-F1 score for all different training ratio. Our model always had the best performance compared to convolution or deepset based models in left figures of Figure 18 (a) and (b). Specifically, we can see that Natural-HNN had outstanding performance in BRCA cancer dataset in the left figure of Figure 18 (b). Thus, we can conclude that Natural-HNN is more expressive compared to convolution based models. Also, when inductive bias (interaction context) matches the data characteristics (BRCA), Natural-HNN provides outstanding performances. From the result, we could verify that Natural-HNN can utilize context information to get good performance. When compared with other baselines, in the right figures of Figure 18 (a) and (b), we can see that our model could achieve better, or at least comparable performance when compared with baselines. We can conclude that our model has expressive power comparable to other attention (including Set Transformer) or equivariance based models. Again, we can observe that Natural-HNN achieved outstanding performance in BRCA dataset by capturing context types. Considering that

Natural-HNN had good generalization and expressivity, we argue that our model made a proper trade-off between expressive power and generalization as described in Section 3.1.

## F.4 CANCER SUBTYPE CLASSIFICATION (MICRO F1)

We briefly provide Micro F1 scores of each model in cancer subtype classification task. The Table 24 also shows that our model generally performs well on most of cancer datasets.

Table 24: Micro F1 score of each model with parameter and hyperparameter of the best Macro F1 score. Top two models are colored by **First**, **Second**. †: the variant of the model using multihead attention. ⋆: we did not use $\mathcal{L}_{dis}$.

| Method | BRCA | STAD | SARC | LGG | HNSC | CESC | KIPAN | NSCLC |
|---|---|---|---|---|---|---|---|---|
| HGNN | 0.817 ± 0.027 | 0.727 ± 0.026 | 0.739 ± 0.057 | 0.696 ± 0.034 | 0.888 ± 0.031 | 0.903 ± 0.034 | 0.935 ± 0.010 | **0.960** ± 0.016 |
| HCHA | 0.808 ± 0.024 | 0.725 ± 0.036 | 0.731 ± 0.058 | 0.685 ± 0.039 | 0.876 ± 0.034 | 0.911 ± 0.034 | 0.939 ± 0.014 | 0.954 ± 0.009 |
| HNHN | 0.806 ± 0.027 | 0.729 ± 0.067 | 0.733 ± 0.046 | 0.676 ± 0.037 | 0.884 ± 0.018 | 0.910 ± 0.033 | 0.931 ± 0.020 | 0.958 ± 0.016 |
| UniGCNII | 0.791 ± 0.027 | 0.797 ± 0.038 | **0.761** ± 0.046 | 0.665 ± 0.038 | 0.910 ± 0.013 | 0.911 ± 0.018 | **0.947** ± 0.010 | 0.950 ± 0.017 |
| AllDeepSets | 0.823 ± 0.025 | 0.748 ± 0.039 | 0.657 ± 0.035 | 0.669 ± 0.045 | 0.895 ± 0.025 | 0.927 ± 0.024 | 0.923 ± 0.016 | 0.954 ± 0.010 |
| AllSetTransformer | 0.827 ± 0.031 | 0.710 ± 0.047 | 0.749 ± 0.047 | 0.656 ± 0.037 | 0.898 ± 0.016 | 0.908 ± 0.025 | 0.938 ± 0.011 | 0.954 ± 0.014 |
| HyperGAT | 0.754 ± 0.116 | 0.725 ± 0.050 | 0.645 ± 0.106 | 0.669 ± 0.051 | 0.889 ± 0.030 | 0.900 ± 0.025 | 0.913 ± 0.036 | 0.928 ± 0.019 |
| HyperGAT† | 0.753 ± 0.072 | 0.676 ± 0.108 | 0.643 ± 0.098 | 0.665 ± 0.042 | 0.883 ± 0.053 | 0.896 ± 0.021 | 0.907 ± 0.256 | 0.940 ± 0.009 |
| SHINE | 0.659 ± 0.090 | 0.590 ± 0.127 | 0.618 ± 0.106 | 0.649 ± 0.058 | 0.846 ± 0.032 | 0.890 ± 0.044 | 0.866 ± 0.149 | 0.879 ± 0.098 |
| SHINE† | 0.783 ± 0.027 | 0.711 ± 0.061 | 0.709 ± 0.045 | 0.654 ± 0.044 | 0.873 ± 0.027 | 0.907 ± 0.031 | 0.936 ± 0.012 | 0.954 ± 0.013 |
| HSDN | **0.838** ± 0.022 | **0.801** ± 0.033 | 0.758 ± 0.047 | 0.694 ± 0.036 | 0.892 ± 0.025 | 0.925 ± 0.024 | **0.950** ± 0.008 | **0.962** ± 0.013 |
| ED-HNN | 0.826 ± 0.024 | 0.793 ± 0.047 | **0.761** ± 0.039 | **0.703** ± 0.028 | 0.913 ± 0.021 | 0.925 ± 0.035 | 0.942 ± 0.012 | 0.955 ± 0.012 |
| ED-HNNII | 0.815 ± 0.027 | 0.748 ± 0.024 | 0.694 ± 0.050 | 0.696 ± 0.038 | **0.916** ± 0.013 | **0.942** ± 0.024 | 0.942 ± 0.010 | 0.953 ± 0.012 |
| Natural-HNN⋆ (ours) | **0.869** ± 0.024 | **0.824** ± 0.027 | **0.770** ± 0.040 | **0.709** ± 0.033 | **0.923** ± 0.020 | **0.932** ± 0.024 | **0.944** ± 0.009 | **0.962** ± 0.013 |

## F.5 CAPTURED CONTEXT IN CESC

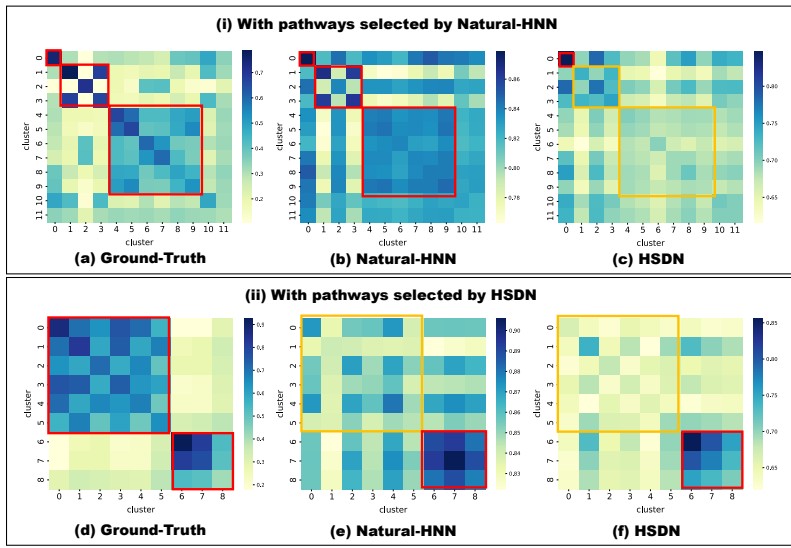

Figure 19: Captured interaction context. Pathways are selected by SHAP value. Captured patterns are shown in red box and not captured patterns are shown with orange box. Weakly captured case is marked as dotted red block.

Figure 19 shows the captured context result in CESC. The evaluation and interpretation method is identical to that of Section 5.3. As we can see in the figure, for pathways selected by Natural-HNN, Natural-HNN correctly captures context similarities between clusters (red box) while HSDN does not (orange box). For the pathways selected by HSDN, Natural-HNN and HSDN partially captures cluster similarity. However, when comparing orange box in (d) and (f), we can observe that Natural-HNN captures interaction context slightly better than HSDN even with the pathways selected by HSDN.

## F.6 FACTOR DISCRIMINATION ANALYSIS

Finally, we perform an experiment to clarify that factors captured by Natural-HNN potentially have different contexts. Since each factor encodes different context and since clusters generated by CliXO algorithm assigns functionally (i.e., context) related hyperedge types, each factor is likely to be related to different clusters. Thus, for each factor and for each cluster, we averaged relevance scores $\alpha_i^k$ of hyperedges that belong to the same cluster. The cluster that is relevant to a specific factor would have high value while irrelevant factors would have small value for that cluster. Figure 20 shows the result of Natural-HNN and HSDN. We have the following observations: **1)** In Natural-HNN, each factor has a different

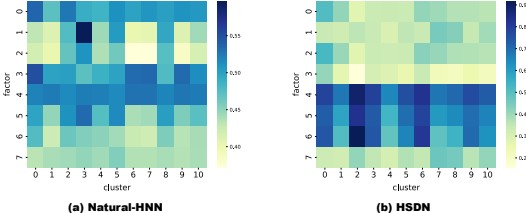

Figure 20: Factor-Cluster Relevance. For the pathways that belongs to the same cluster, we averaged their factor relevance score for each factors. (a) Natural-HNN case shows that each factor contributes to clusters differently. (b) HSDN case shows that some factors have similar contribution over all clusters.

score distribution over clusters. This implies that the factors are contributing to different clusters since they encode different functions. **2)** In HSDN, some factors have similar distribution over clusters. For example, factor 0 and factor 2 of HSDN are similar in every factor. Also, factor 1 and factor 7 have highly similar score distribution over clusters. This implies that some factors of HSDN are correlated. **3)** While scores in (a) are distributed to various clusters and factors, scores in (b) are concentrated on factor 4,5 and 6. Since only few factors are actively reflected while others do not, HSDN fails to utilize different factors effectively. This experiment result is notable since Natural-HNN could capture different context per factor even without factor discrimination loss $\mathcal{L}_{dis}$ while HSDN failed to capture different factors and failed to use them properly even if it adopted factor discrimination loss. Thus, we can consider naturality guidance as an effective criterion for disentanglement.

## F.7 RELIABILITY OF NATURAL-HNN IN BIOLOGY

In order for a model to be reliable, the model should provide consistent output regardless of the choice of hyperparameters. So we conducted an experiment to check whether models consistently rely on the same pathways. If a model consistently rely on the same pathways for prediction regardless of the hyperparameter, biologists might consider the model to be reliable since it potentially captured and used what can be explained with biological domain knowledge. On the other hand, if the model relies on different pathways for different hyperparameters, biologists might not trust the model.

To check whether model relies on the same pathways, we ranked the pathways with SHAP value and selected top-k pathways. These pathways are the ones that models relied most for their prediction. Then, we calculated Jaccard similarity of top-k pathways for different hyperparameters. If top-k pathways earned from each hyperparameter combination is similar, then we can conclude that model always rely on the same pathways regardless of the hyperparameters.

Figure 21 and Figure 22 are the result of calculating Jaccard similarity between different hyperparameter combinations on BRCA dataset. The hyperparameters we changed was the hidden dimension size and the number of factors. Values in each tick of row and column is the pair of the two hyperparameters (i.e., the value in the ticks represent (hidden dimesion, number of factors) pair). Each heatmap shows Jaccard similarity when selecting top 10, 15, 20, 50, 100 and 500 pathways. Figure 21 is the results for Natural-HNN and Figure 22 is the result for HSDN. We also calculated average Jaccard similarity for each heatmap.

The ideal result would show dark blue colors (high similarity) to all cells in the heatmap. It means that top-k pathways that a model relied on are always the same regardless of the hyperparameter. When comparing Figures 21 and 22, we can see that Natural-HNN tends to rely on the same pathway regardless of the hyperparameter while HSDN does not. When comparing average Jaccard similarity scores, we can quantitatively observe that Natural-HNN has better consistency when compared to HSDN. For example, Jaccard similarity with top 15 pathways of Natural-HNN (21 (b)) has average similarity of 0.759 while that of HSDN (22 (b)) has average similarity of 0.555.

From this experiment, we can conclude that Natural-HNN is reliable since it consistently focuses on the same pathways regardless of the choice of hyperparameters. Also, we could again verify that our model captures the functionality of pathways (interaction context of hyperedge) and expect that our model will work reliably in different dataset or different biological applications. Note that similar analysis for Figure 23 and Figure 24 provides similar conclusion.

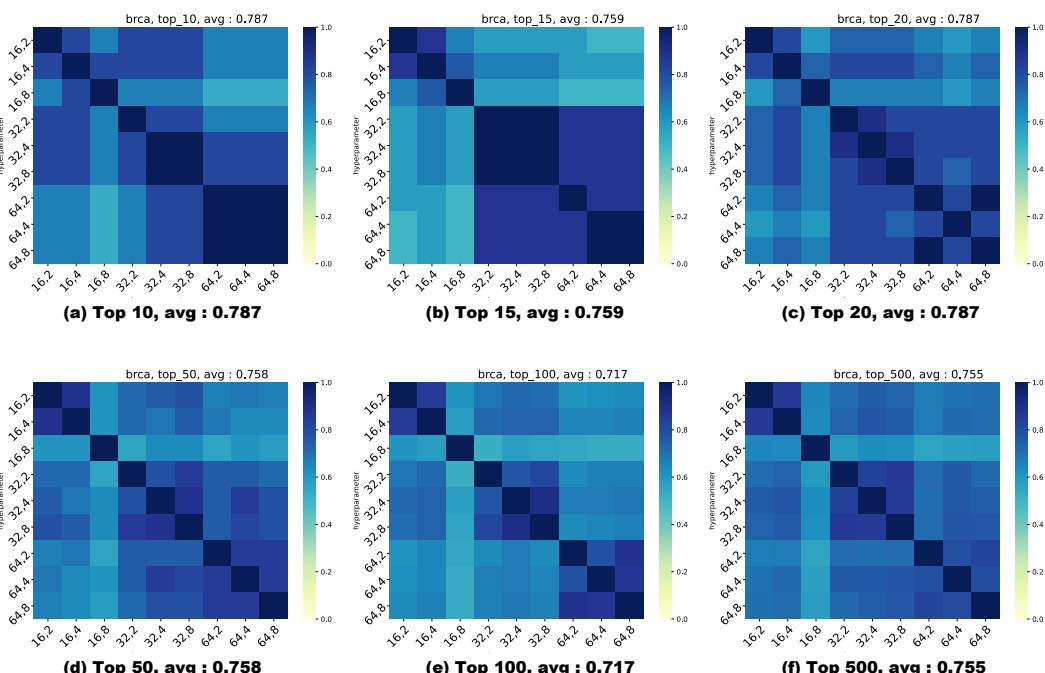

Figure 21: Jaccard similarity calculation result for Natural-HNN on BRCA. We can observe that Natural-HNN generally relies on similar pathways regardless of hyperparameters by showing high Jaccard similarity value.

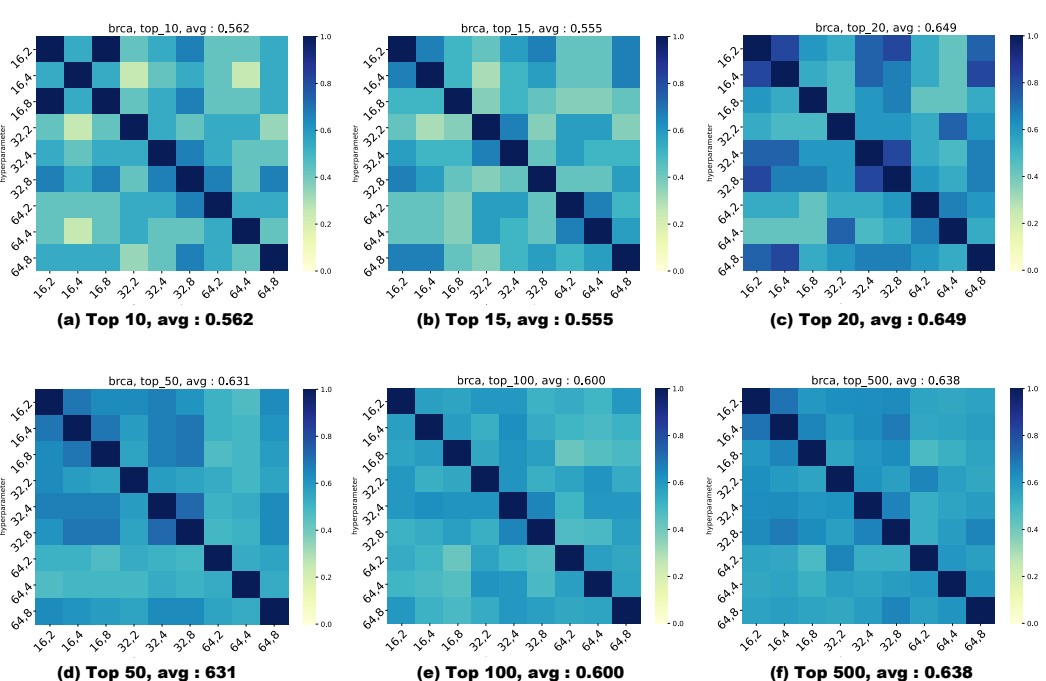

Figure 22: Jaccard similarity calculation result for HSDN on BRCA. We can observe that HSDN relies on different pathways for different hyperparameters by showing strong diagonal pattern. This inconsistency makes HSDN an unreliable model for biology.

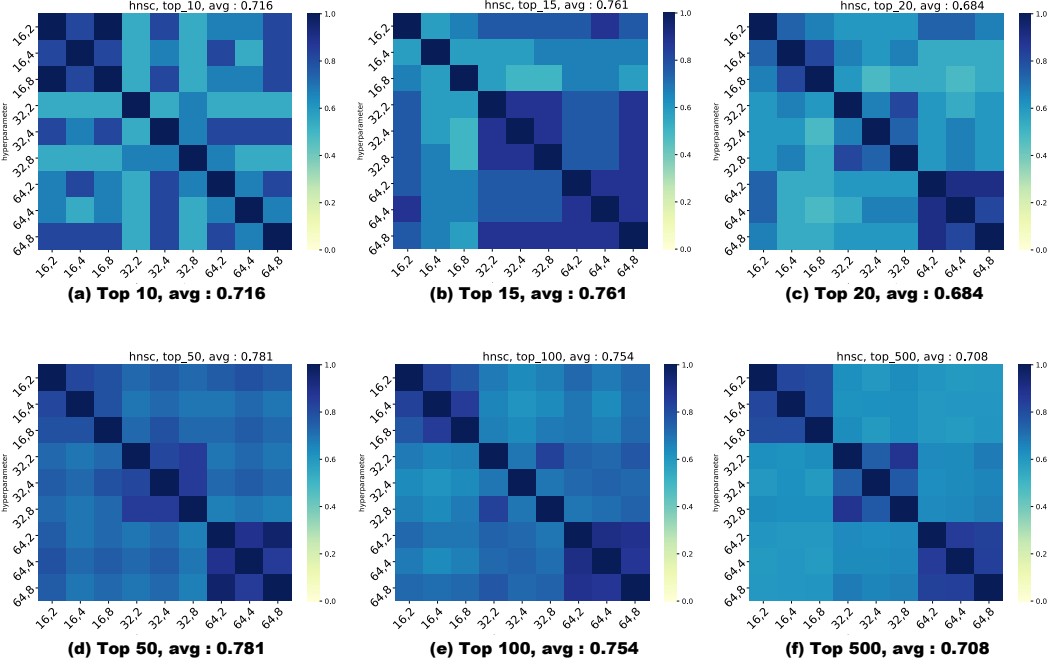

Figure 23: Jaccard similarity calculation result for Natural-HNN on HNSC. We can observe that Natural-HNN generally relies on similar pathways regardless of hyperparameters by showing high Jaccard similarity value.

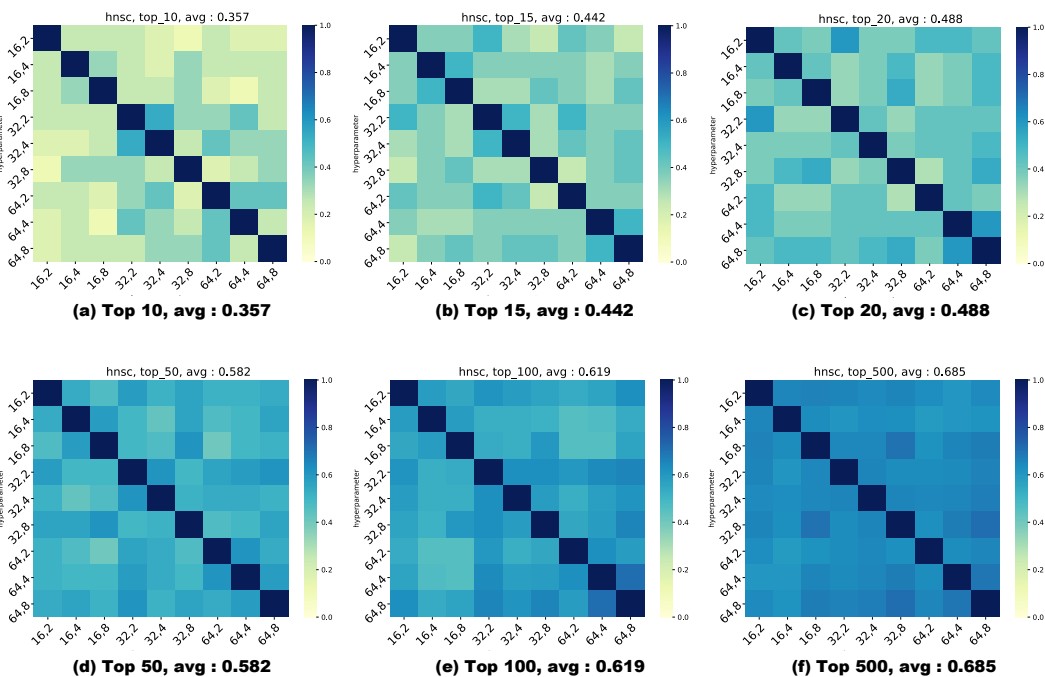

Figure 24: Jaccard similarity calculation result for HSDN on BRCA. We can observe that HSDN relies on different pathways for different hyperparameters by showing strong diagonal pattern. This inconsistency makes HSDN an unreliable model for biology.

## G  Basic Concepts in Category Theory

### G.1  Category Theory

Category theory (Fong & Spivak, 2018; Leinster, 2016) is widely used to represent and analyze the structure or relation of a system. Instead of focusing on the details, category theory takes bird's eye view to see global structure and patterns. Recently, category theory is used to explain learning mechanism of machine learning methods (Bergomi & Vertechi, 2022; Lewis, 2019; Gavranović, 2019; Fong & Johnson, 2019; Fong et al., 2019; Cruttwell et al., 2022; Shiebler et al., 2021; de Haan et al., 2020; Barbiero et al., 2023; Yuan, 2023b; Dudzik et al., 2023; Dudzik & Veličković, 2022; Yuan, 2023a). In this paper, we only use simple, fundamental concepts of category theory: category, functor, natural transformation and product.

### G.2  Category

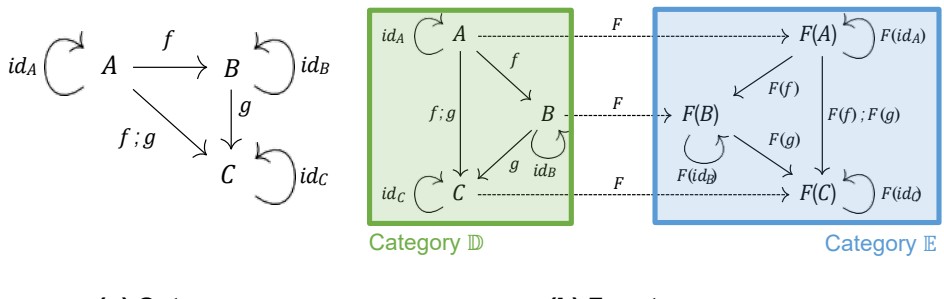

(a) Category                      (b) Functor

Figure 25: Category and Functor

A category $\mathbb{C}$ is contains four components: collection of objects, morphisms, composition rule and identities.

- Collection of objects : $Ob(\mathbb{C})$ (ex : $\{A, B, C\}$ in Figure 25 (a))
- For every pair of objects $A, B \in Ob(\mathbb{C})$, there exists a set $Hom_{\mathbb{C}}(A, B)$. Element of the set is morphism and is denoted as: $f : A \to B$.
- For every three objects $A, B, C \in Ob(\mathbb{C})$, morphisms $f \in Hom_{\mathbb{C}}(A, B)$ (i.e. $f : A \to B$) and $g \in Hom_{\mathbb{C}}(B, C)$ (i.e. $g : B \to C$), **composition rule** holds : $f \mathbin{\overset{\circ}{,}} g = g \circ f \in Hom_{\mathbb{C}}(A, C)^{21}$.
- For every object $A \in Ob(\mathbb{C})$, there exists an identity morphism $id_A \in Hom_{\mathbb{C}}(A, A)$ satisfying the following : $id_A \mathbin{\overset{\circ}{,}} f = f = f \mathbin{\overset{\circ}{,}} id_B$ for morphism $f : A \to B$.

Fig. 25 (a) shows an example of a category with three objects $(A, B, C)$. For each object, there is an identity morphism $(id_A, id_B, id_C)$. For every object pair, there is morphism $(f, g, f \mathbin{\overset{\circ}{,}} g)$ with composition rules.

One of the most important categories is **Set**. In **Set**, the objects are sets and morphisms are functions mapping two sets. The composition rule is satisfied since a composition of two functions becomes a function. Another important category is category of relations, which is denoted as **Rel**. The objects of **Rel** are sets and relations $R \subseteq A \times B$ are morphisms between objects $A$ and $B$. Partially ordered set or poset can be considered as a category where objects are sets and morphisms are partial orders $\leqslant$. Since partial order is a kind of a relation, we can consider this category is a kind of **Rel**.

In Section 3, we analyzed hypergraph message passing framework, and found that, as nodes (considering node as set) are included in hyperedges, hypergraph message passing framework has poset structure with inclusion maps between them. We will define it **PISet**, a category for poset with inclusion morphisms (object is a set, morphisms are inclusions). Since inclusions are partial orders, which is also a relation, we can consider **PISet** as a kind of **Rel** category.

We can define our own category, similar to the one in a prior work (Sheshmani & You, 2021), such that objects are vector representations and their (linear or non-linear) transformations are morphisms. We will call this a 'category of Deep Learning Representations' and denote **DLRep**.

---

[21]Two notations $f \mathbin{\overset{\circ}{,}} g$ and $g \circ f$ have the same meaning : "applying $f$ first, and then applying $g$"

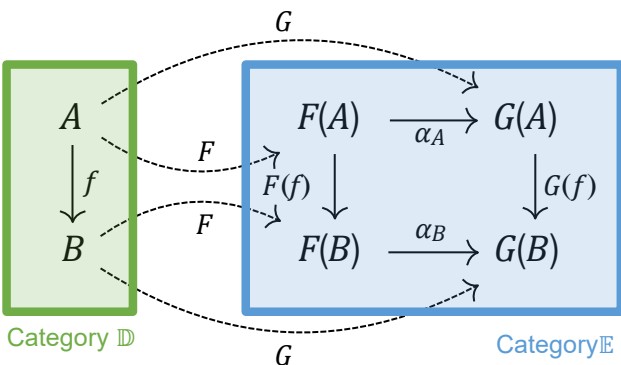

Figure 26: Natural transformation. Identity morphisms are omitted in the figure for simplicity.

### G.3 FUNCTOR

Functor is a structure preserving map between categories. Objects and morphisms in one category are mapped to objects and morphisms in different category, respectively. Figure 25 (b) shows an example of a functor mapping from category $\mathbb{D}$ to category $\mathbb{E}$. Each object in category $\mathbb{D}$ (i.e., $A, B, C$) is mapped to objects in category $\mathbb{E}$ (i.e., $F(A), F(B), F(C)$). The morphisms, including identity morphism, and their compositions in category $\mathbb{D}$ (i.e., $id_A, id_B, id_C, f, g, f \,{}^\circ_\circ\, g$) are also mapped to morphisms in category $\mathbb{E}$ (i.e., $F(id_A), F(id_B), F(id_C), F(f), F(g), F(f) \,{}^\circ_\circ\, F(g)$). In a metaphorical sense, functors serve as bridges that connect two distinct realms while maintaining an identical compositional structure[22].

One example can be a functor mapping from **Set** to **DLRep**. Each set (object) in **Set** is mapped to a vector representation (object) in **DLRep**. Functions (morphisms) in **Set** are mapped to transformations (morphism) between vector representations in **DLRep**. This functor is related to representation learning, since entities (i.e. concept or set) are mapped to their vector representations preserving their compositional structure (relation).

### G.4 NATURAL TRANSFORMATION

Given two functors mapping from one category to another category, i.e., $F$ and $G : \mathbb{D} \to \mathbb{E}$, natural transformation is a way of relating these two functors using morphisms in target category $\mathbb{E}$. Specifically, for each object $A \in \mathbb{D}$, there exists a morphism $\alpha_A : F(A) \to G(A)$ in $\mathbb{E}$. The natural transformation must satisfy the following condition. For every morphism $f : A \to B$ in $\mathbb{D}$,

$$F(f) \,{}^\circ_\circ\, \alpha_B = \alpha_A \,{}^\circ_\circ\, G(f) \tag{4}$$

must hold. This condition is called the ***naturality condition***. Figure 26 shows an example of natural transformation. Functors $F$ and $G$ map objects and morphisms in category $\mathbb{D}$ to category $\mathbb{E}$. Natural transformation $\alpha : F \Rightarrow G$ maps $F(A)$ and $F(B)$ with $\alpha_A$ and maps $G(A)$ and $G(B)$ with $\alpha_B$. The objects and morphisms mapped by two functors as well as natural transformation $\alpha$ all belong to the category $\mathbb{E}$. Thus, natural transformation can be seen as a way of relating different views using morphisms in $\mathbb{E}$[23].

### G.5 PRODUCT

**Product of Objects**

Let $\mathbb{C}$ be a category. For two objects $X_1, X_2 \in Ob(\mathbb{C})$, one can define product of two objects $X_1 \times X_2$ with morphisms $p_1 : X_1 \times X_2 \to X_1$ and $p_2 : X_1 \times X_2 \to X_2$ which are called **projections**. Then, the composition of objects in Figure 27 must be satisfied. Given object $Y \in Ob(\mathbb{C})$ with two morphisms $f_1 : Y \to X_1$ and $f_2 : Y \to X_2$, there exists a unique morphism called **pairing** $\langle f_1, f_2 \rangle : Y \to X_1 \times X_2$

---

[22]The typical example of deep learning method using this concept is sheaf neural network (Hansen & Gebhart, 2020), motivated from cellular sheaf (Hansen & Ghrist, 2019). There are also numerous studies in data science with a similar perspective (Mansourbeigi, 2018; Vepstas, 2019; Kvinge et al., 2021).

[23]One typical example of deep learning method using this concept is Natural Graph Networks (de Haan et al., 2020).

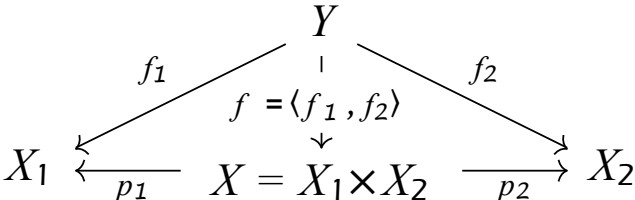

Figure 27: Product of objects.

that satisfies the composition : $f_1 = \langle f_1, f_2 \rangle \mathbin{\text{\textnumero}} p_1$ and $f_2 = \langle f_1, f_2 \rangle \mathbin{\text{\textnumero}} p_2$. Note that pairing $\langle f_1, f_2 \rangle$ is often called as product of morphisms. However to differentiate the concept we introduce below, we will call it pairng, following the recent work (Zhang & Sugiyama, 2023).

**Product of Morphisms**

$$
\begin{array}{ccccc}
X_1 & \xleftarrow{\ p_1\ } & X = X_1 \times X_2 & \xrightarrow{\ p_2\ } & X_2 \\
\downarrow{\scriptstyle f_1} & & \downarrow{\scriptstyle f_1 \times f_2} & & \downarrow{\scriptstyle f_2} \\
Y_1 & \xleftarrow{\ q_1\ } & Y = Y_1 \times Y_2 & \xrightarrow{\ q_2\ } & Y_2
\end{array}
$$

Figure 28: Product morphisms.

Let $\mathbb{C}$ be a category. For objects $X_1, X_2, Y_1, Y_2 \in ob(\mathbb{C})$ and morphisms $f_1 : X_1 \to Y_1$ and $f_2 : X_2 \to Y_2$, we can define **product of morphisms** $f_1 \times f_2 : X_1 \times X_2 \to Y_1 \times Y_2 := \langle p_1 \mathbin{\text{\textnumero}} f_1, p_2 \mathbin{\text{\textnumero}} f_2 \rangle$ satisfying the compositional structure shown in Figure 28.

## H ADDITIONAL EXPLANATION IN DETAILS

Note that the basic concepts in category theory are described in Appendix G.

### H.1 DISENTANGLED REPRESENTATION LEARNING

**Entangled and Disentangled Representation** Disentangled representation learning aims to separate the factor that is related to the variations of the data. For example, some might try to discover the factor that affects the color of an object or the factor that affects the background of an image. In graph neural networks, interactions between entities are usually entangled. In other words, interactions usually contain various factor behind connections but are not explicitly separated. Previous works like DisenGCN (Ma et al., 2019) tried to disentangle the factor behind the connections.

Recently, DisGNN (Zhao et al., 2022) tried to disentangle edge types during message passing process of GNNs. The paper considered interaction types (colleague or neighbors as an example) as factors of edges and tried to integrate disentanglement during message passing process. This kind of disentanglement for message passing is the goal of Natural-HNN.

**Disentangling as product in category theory**

Disentangling methods try to separate an entity into the factors that consists the entity. Thus, it can be analyzed with concept with product in category theory, which was explained in Appendix G. Although recent work (Zhang & Sugiyama, 2023) analyzed the concept of disentanglement, we are going to analyze it in our way, since the paper (Zhang & Sugiyama, 2023) covers disentanglement of generative factors, which does not suit for message passing framework. The difference comes from the fact that, generative factor disentanglement is based on equivariance property, whose morphisms maps an object to itself. Since message passing maps one object to the other object, we need our own way of analyzing disentanglement[24].

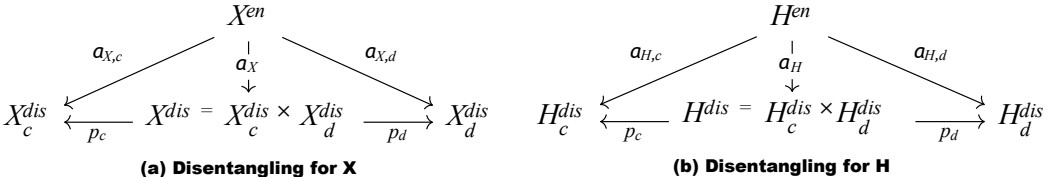



(a) Disentangling for X          (b) Disentangling for H

Figure 29: Disentangling as product of objects.



In section 3, we have seen that disentangling the entangled representation can be seen as a natural transformation between two representations. The Figure 29 shows the disentanglement as product of objects. The entangled representation for $X$ ($X^{en}$) can be converted to disentangled representation $X^{dis}$ through natural transformation $\alpha_X = \langle \alpha_{X,c}, \alpha_{X,d} \rangle$. Since disentangled representation is a collection of factor representations, it can be represented as a product of factor representations $X_c^{dis} \times X_d^{dis}$. The projections $p_c, p_d$ can extract factor representations $X_c^{dis}, X_d^{dis}$. This process is the same as applying $\alpha_{X,c}, \alpha_{X,d}$ respectively. This is the same for disentangling $H$.

Figure 30 shows how morphisms between disentangled node representations and disentangled hyperedge representations are separated. Disentangling morphisms can be explained with the concept of product of morphisms. In the Figure 30, $f_c^{dis}, f_d^{dis}$ represents factor specific morphisms or factor specific message passing. The product of two morphisms, $f_c^{dis} \times f_d^{dis}$, corresponds to message passing for entire factors. What is different from Figure 28 is that we use the same projections $p_c$ instead of using two different projections $p_1, q_1$. This is due to the fact that $X^{dis}$ and $H^{dis}$ both are disentangled representation, meaning that the same projection can extract the same factor for both $X, H$.

**Implementation as MLP**

---

[24]Actually, the biggest difference is that, in generative factor, factor specific morphisms can be independently mapped to itself. However, in message passing, we need to map all factor related morphisms from one object ($X$) to the other ($H$). If only some of them are used independently, it will be mapped to the another object (not $H$).

$$H_c^{dis} \xleftarrow{\ p_c\ } H^{dis} = H_c^{dis} \times H_d^{dis} \xrightarrow{\ p_d\ } H_d^{dis}$$

$$\uparrow f_c^{dis} \qquad f_c^{dis} \times f_d^{dis} \qquad \uparrow f_d^{dis}$$

$$X_c^{dis} \xleftarrow{\ p_c\ } X^{dis} = X_c^{dis} \times X_d^{dis} \xrightarrow{\ p_d\ } X_d^{dis}$$

Figure 30: Morphism of products in disentanglement.

Usually, disentangling entangled representation is implemented with MLP. Let's suppose the desired output size of disentangled representation (i.e., output size of a vector that concatenated every factor representations) is $d$. Usually, $K$ number of factor-specific MLPs (which outputs vector with size $\frac{d}{K}$) are used to extract factor representations. This corresponds to $X_c^{dis}, X_d^{dis}$ in Figure 29. As we have seen above, it is same as applying $\alpha_X \mathbin{\mathring{\circ}} p_c, \alpha_X \mathbin{\mathring{\circ}} p_d$. This can be implemented as using one MLP (which outputs vector with size $d$), which corresponds to $\alpha_X$ and then chunking the disentangled representations into factor representations. Chunking operation can be considered as projections $(p_c, p_d)$. Thus, although we explained as using $K$ factor specific MLPs in Section 4, we actually use one MLP (which outputs vector with size $d$) in actual implementation. Thus, the concatenation operation for $h_v$ is not used in the implementation as applying a single MLP equals to the operation of applying $K$ separate MLPs and then concatenating them.

## H.2 CAPTURING INHERENT HETEROGENEITY

Actually, capturing context of interaction has potential of capturing heterogeneous edge types. Let's consider the case of heterogeneous graph with heterogeneous edges as an example. GNNs reflecting the edge types can be said as considering the context of interactions between entities. Thus, capturing interaction context in hyperedges has potential of capturing heterogenous edge types by considering edge types as categorized result of interaction contexts.

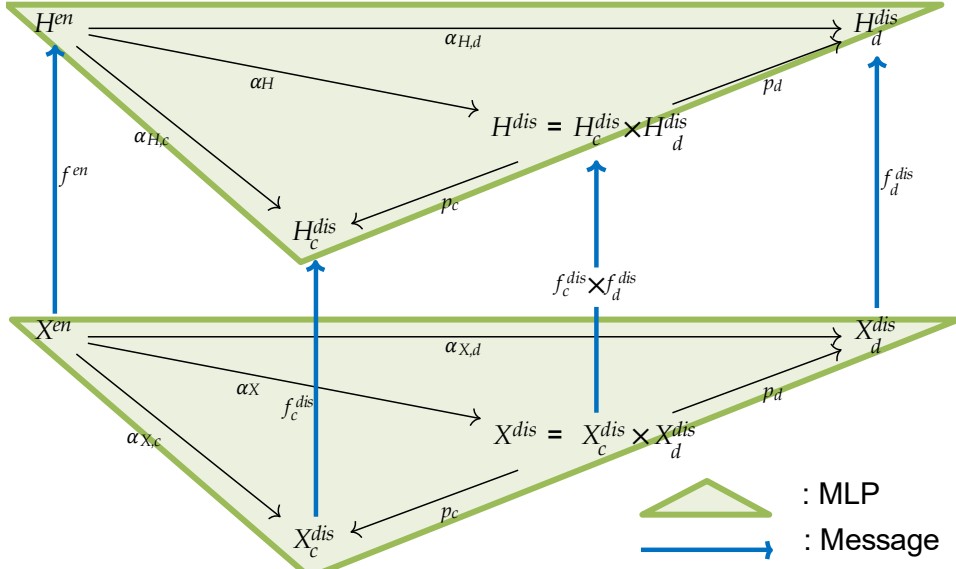

Figure 31: Entire compositional structure. Operations in the implementation are marked with color.

## H.3 Interpretation for Hypergraph MPNN

In Appendix H.1, we have seen how we can analyze disentanglement with concepts of product in the category theory. Applying Figure 29 and Figure 30 to Figure 3 (a) gives the following result (Figure 31). Since this diagram is too complicated, we simplified the figure by extracting factor $c$ related components which resulted Figure 3 (b). The resulting figure is also a natural transformation as it can be seen as a result of applying two different functors. The actual implementation (operation) are marked as the Figure 31.

## H.4 Methodology (How it works)

Since K MLPs are applied to nodes in a hyperedge, it extracts information related to the factors through projection. However, simple projection does not mean that the factor is related to the interaction context. In this section, we will explain how naturality condition guides, although not guarantees, each factors to be related to interaction context. The parameters of factor encoders (K MLPs) are guided to extract interaction context related information during training process. When a specific factor is helpful for performance (predicting labels), the model would try to update parameters of the factor encoder so that the factor information is reflected a lot in hyperedge representation. Since relevance score $\alpha_i^k$ is multiplied to factor representation to get hyperedge represenation ($\alpha_i^k h_{e_i}^k$), the parameters will be updated to increase relevance score $\alpha_i^k$. Considering that relevance score $\alpha_i^k$ is calculated by measuring consistency of factor representation (similarity of hyperedge factor representation learned from two different branches), high relevance score means that the representations are similar. Represenations learned from two branches being similar means that it is highly likely that the naturality condition holds, implying that there exists a morphism between nodes in a hyperedge and the hyperedge under specific projection (type) which means the factor is related to the interaction context. In summary, if a specific context (factor) is informative, the parameter of a factor encoder will be updated to the direction of satisfying naturality. Thus, the factor encoder will eventually encode context-related information. When a specific factor is harmful for performance, the opposite would happen. Since naturality condition guides in which direction to update parameters for each factor, although not guaranteed, it is highly likely that each factor contains different context information.

## H.5 Result analysis of capturing context

Actually, Figure 8 (a) and (c) can explain the experiment result shown in Figure 5 (a,b) and Figure 19 (a,b). For example, in Figure 8 (a), we can see that cluster $C_0$ and $C_1$ both have common parent ($C_5$) and common child ($P_{339}$). That's the reason why Figure 5 (a) and (b) both detected high similarity between those clusters. Also, in Figure 8 (a), $C_3$ and $C_4$ has common child. This can explain why Figure 5 (a) and (b) both detected high similarity between two clusters. When applying these analysis with Figure 5 (c) and Figure 19 (c), we can clearly see that HSDN failed to capture functional similarity or hierarchy of pathways.

On the other hand, when comparing Figure 8 (b) and Figure 5 (f), we can see some similarities are not captured. For example, in Figure 8 (b), clusters $C_0, C_1, C_2$ need to have functional similarity since they contain common children or have common parent. However, in Figure 5 (f), we can see that HSDN failed to capture the functional similarities of those clusters. Through this result, we can again conclude that HSDN failed to capture functional context while Natural-HNN could capture it.

Additionally, we can explain why some diagonals of heatmap do not have high value. For example, $C_8$ in Figure 5 (a) and (b) cannot have high similarity between pathways within cluster $C_8$ as $C_8$ contains all pathways. Note that performing the same analysis with Figure 8 (c), (d), Figure 19 gives the similar result.

## H.6 Message Passing as Opinion Dynamics

Opinion dynamics (Hansen & Ghrist, 2021; Jackson, 2011; Siegel, 2009) is a research field studying how opinions or preferences change over time. Each entity has their preferences or opinions. The interactions among entities can change their opinion over time. As interactions can change entities, opinion dynamics have large similarities with the mechanism of Graph Neural Networks. Message passing mechanisms generates messages and sends to its neighbors. The neighbor node receives

message and update its representation. Mean aggregation process can be considered as minimizing the difference between messages and its own representation. This mechanism can be expressed as the concept of consensus in the opinion dynamics. In opinion dynamics, after the interaction, the difference of opinions of people can be reduced over time. When their opinions becomes similar through interactions, we consider the case as reaching the consensus.

In the group discussion example in Section 1, we considered an hyperedge as group discussion. The group discussion will eventually reach a conclusion or a consensus among entities. This is actually the same as node-to-hyperedge message passing process. The hyperedge representation (consensus) are calculated by aggregating messages of nodes (opinions of participants). After the discussion, the consensus will change the opinion of participants, which corresponds to node representation update in message passing framework.

### H.7 ABOUT CONSENSUS

In reality, there can be much more complex cases than what is explained above. Recent opinion dynamics tackles various cases. For example, there can be some cases where some participants actively participate the discussion while other participate passively. In message passing framework, we can think of attention based models as some nodes have higher importance over others. In some cases, some people can partially lie during the discussion to reach consensus. This can be considered as reaching an apparent consensus. Sheaf hypergraph networks (Duta et al., 2024) can be explained with this concept. Participants (nodes) express transformed opinion (transformation of node feature) during discussion rather than directly expressing their opinions (node feature). In our group discussion example, we considered the case where discussion topics can be differed by discussions (hyperedge). In this case, participants express their topic related opinions (factor representation) on a specific topic (context).

As we can see, we can think of various cases of interactions which can be modeled as various message passing neural networks. Since the goal of Natural-HNN is to capture context of interaction, our group discussion example focused on explaining the concept of context as topic of discussion. When reading our example, some might think of other cases that could happen in group discussion, such as not reaching a consensus (community cleavage problem, (Friedkin, 2015)). However, such additional cases are not related to the concept of interaction context. Thus, we did not considered those cases in the example and Natural-HNN. If we want to accommodate additional cases or constraints, we need to add additional module to our model, which is not the goal of this paper. For example, in the case of some nodes not agreeing on opinion of majority (consensus failure), we can think of a model that can disconnect some nodes from a hyperedge.

### H.8 COMPARISON WITH WHATSNET

#### H.8.1 DEFINITION OF CONTEXT AND MODEL

**WHATSNET.** WHATSNET (Choe et al., 2023) tries to explicitly consider hyperedge-dependent relationship among nodes participating the hyperedge. They focus on the insight that importance or role of a node is shaped by the other participants (nodes) in the hyperedge. They adopt Set Transformer to get hyperedge-dependent node embeddings and then get hyperedge embeddings. During this process, they use relative centrality (ranking) as positional encoding, with a motivation that relative position of nodes within hyperedge is closely related to the edge-dependent node labels or characteristics.

**Natural-HNN.** We tried to capture the context that is related to the background or condition of interaction. In other words, Natural-HNN tries to capture 'why or in which condition this interaction occurs'. We used the naturality condition that must be satisfied when the interaction is related to a specific context. Since Natural-HNN focuses on the context that works as backround or condition, and WHATSNET tries to capture context shaped by participants, the models aim to capture different contexts.

**Paper author & Group discussion example.** WHATSNET provided a paper author example. When a paper is written by four students and one professor, it is highly likely that the professor becomes the last author considering the participants (context). The context defined by Natural-HNN is related to the topic or field of the research paper. These two contexts are independent. For example, the topic of research is not necessarily related to the relationship of participants. The

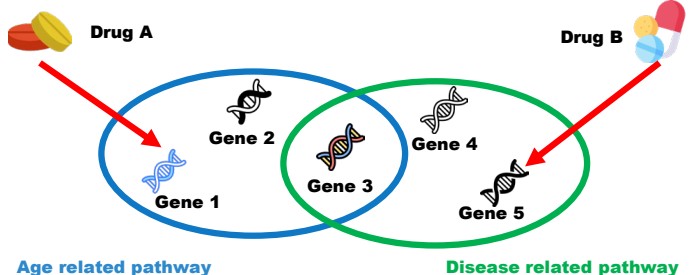

Figure 32: Example of combination therapy. Two different drugs target different proteins in different pathways to get the synergetic effect.

research topic can be economics, politics or science regardless of the relationship of participants. Also, relationship of participants are not shaped from the topic. A similar metaphor can be applied to our group discussion example.

### H.8.2 ADVANTAGES OF NATURAL-HNN AND FUTURE WORK

**Advantage of capturing context.** In reality, many interactions have purposes and are often affected by environment. For example, genetic pathways have specific functional purpose behind gene interaction and its characteristics or gene expressions are affected by environment such as hypoxia (Liang et al., 2018). Also, interaction occurs under specific conditions. Most human cells have the same genes and pathways but they are expressed differently depending on cell type, tissue, disease or age condition. For example, activation of growth hormone-related pathway is likely to depend on the age of a person. Reflecting such contexts in gene or protein representations help better predicting the influence of mutation or treatment.

**Advantage in cancer subtype classification.** Many biological mechanisms operate through pathways. When some pathways are not working properly, it affects the function of important biological process such as cell proliferation. When cell proliferation is not controlled, it can cause tumor growth. Thus, gene related diseases such as cancer are highly affected by malfunctions of pathways. In other words, the functions that are affected by malfunctioning pathways are highly related to the type of cancer. Hence, the status of a pathway with respect to its function (performing function properly or not) is important for cancer subtype classification. Since features of genes are statistics of gene expression levels (how many times a gene is activated, **measured in context-independent manner**), the pathway representation learned from gene interaction likely includes the status of a pathway. Since genes exhibit different characteristics or gene expression levels under different context, it is important to get interaction context-specific representations to properly get the status of pathways.

**More Applications.** One application can be **drug synergy prediction** (Tang & Gottlieb, 2022) for combination therapy (Figure 32). To get better efficacy and cytotoxicity in chemotherapy, combination of two or more drugs are often prescribed for the patients. In Figure 32, drug A targets gene 1 that participates age-related pathway and drug B targets gene 5 that participates disease-related pathway. Since gene 1 interacts with gene 3 through one pathway and gene 5 interacts with gene 3 through the other pathway, targeting gene 1 and gene 5 at the same time can have effects on gene 3. However, the synergy can depend on the conditon or context of interaction. Since age-related pathway is activated (interaction occurs) only for certain age (period of human growth for example), the synergy depends on the patient's age. Thus, reflecting such context for gene or protein representation is important. Another application can be **drug repurposing** task (Han et al., 2024) which seeks new uses for existing drugs. Other example can be predicting the **influence of mutation** of a gene. Since genes participate pathways to perform biological function, reflecting functional semantic (purpose of interaction) or condition can be helpful for predicting influence of mutation. The provided examples are all related to biology domain since biology is the field where multiway interactions with contexts are easily found. However, considering that many complex systems contain interaction with contexts (condition or purpose), we expect to encounter more examples in other domains in the future.

**Future work 1.** Since Natural-HNN averages factor representations of nodes in node-to-hyperedge propagation, it cannot capture context-specific node importance. As our future work, we are plan-

ning to devise a model that can give different importance to nodes per factor without relying on similarity criterion. Note that the importance captured in this context is different from WHATsNET as the importance is decided by the context of interaction rather than who particiaptes the interaction. For example, if economy is the topic of a discussion, it might be better to give more importance to a person who majored economics (or assign the person as moderator). This importance is not dependent on 'who participates the discussion' as the importance of participants can change if the topic differs even with the same participants.

**Future work 2.** As we have seen through several examples in Appendix H.8.1, the definition of contexts are independent. In other words, the context shaped by 'who participates the interaction' and context related to condition or background can compose richer context. For example, let's suppose one grown-up researcher majored in biology, another grown-up researcher majored in AI, two students from biology domain and other two students from AI domain wrote a research paper. If the topic is related to biology or the paper is submitted to biology journal, the last author might be the grown-up researcher majored in biology. On the other hand, if the topic is more focused on AI or submitted to AI conference, the last author might be the grown-up researcher majored in AI. Thus, we believe it is possible to integrate two contexts to capture rich context of interaction.

