# OpenReview forum: "Naturality-Guided Hyperedge Disentanglement for Message Passing Hypergraph Neural Network"
_ICLR.cc/2025/Conference — Submitted to ICLR 2025_

### Official Review · Reviewer_gjHi · 2024-11-01

**Soundness:** 2
**Presentation:** 3
**Contribution:** 2
**Rating:** 5
**Confidence:** 3

**Summary:**

The paper introduces Natural-HNN, a hypergraph neural network (HNN) designed to effectively capture the context of interactions within hyperedges. In experiments, it outperforms baseline models in cancer subtype classification.

**Strengths:**

The paper is clearly presented and easy to follow, with the proposed model demonstrating superior empirical performance compared to existing baseline methods.

**Weaknesses:**

This paper overlooks an important related work [1], which also discusses the idea that different hyperedges provide distinct contextual information to different nodes. In the introduction, the authors highlight one of their key contributions, stating, "To the best of our knowledge, we are the first to propose a hyperedge disentanglement-based method that is systematically designed to capture the context of multiway interaction." Therefore, it is crucial to clarify the differences and advantages of the proposed method compared to the approach presented in [1].

[1] Choe, Minyoung, et al. "Classification of edge-dependent labels of nodes in hypergraphs." Proceedings of the 29th ACM SIGKDD Conference on Knowledge Discovery and Data Mining. 2023.

**Questions:**

[1] applies the proposed method to the edge-dependent node classification task, where node labels are determined by the hyperedge context. This makes it clear and intuitive why modelling the hyperedge context is beneficial for enhancing model performance. However, in this paper, the authors apply their method to a more general node classification setting, which raises questions about the importance of modelling interaction context. It would be helpful if the authors included a dedicated section discussing why capturing interaction context is crucial for improving model performance.

Minor: It seems that the link to the code provided in the paper does not work. Could the authors please double-check the accessibility of the code?

[1] Choe, Minyoung, et al. "Classification of edge-dependent labels of nodes in hypergraphs." Proceedings of the 29th ACM SIGKDD Conference on Knowledge Discovery and Data Mining. 2023.

---

> ### Author Response · Authors · 2024-11-23
>
> We revised introduction section to clearly deliver the definition of context. we added related work section to briefly explain WHATsNet. We added detailed explanations in Appendix H.8 (last three pages)
>
> ## Weakness
> We want to clarify that the definition of context in our paper and that of WHATsNet is different. WHATsNet focuses on the context shaped by the participants by considering hyperedge dependent relation of nodes and relative positions between them. However, the context we focused on our paper is more related to the background or condition of interaction. For a group discussion example, we are interested in the ‘reason for holding this discussion (purpose)’ or ‘the topic of the discussion’. This kind of context is clearly different from the context defined in WHATsNet. We cannot know whether participants will discuss economics or science just by checking the participants of the discussion. On the other hand, we cannot know who will lead the discussion as a moderator if we do not consider who participates in the interaction. We modified the introduction and related work to clearly deliver the definition of context we are interested. More details can be found in Appendix H.8.1 and ‘Future work’ in Appendix H.8.2 of our revised version.
>
> We would like to emphasize that considering the context (background or condition) of interaction is indeed important for learning good representations. For example, even if the same genes and pathways exist for most human cells, they are only activated under specific conditions such as cell type, tissue or age. If a pathway is related to human growth (growth hormone), it shows high activation when a person is a teenager. That pathway is not likely to be activated for grown-ups. If someone wants to predict the effects of drugs or the influence of gene mutation, the contexts (functions or conditions of interaction) needs to be considered. Thus, reflecting interaction context (background or condition) for node representation is beneficial. More explanations are written in Appendix H.8.2.
>
> ## Question
> For hypergraph classification task (cancer subtype classification, Table 1), the importance of capturing context can be described as bellow.
>
> Many biological process works through the functions of pathways. When a pathway is not working properly, it affects the function of important biological process such as cell proliferation. Note that uncontrolled cell proliferation often leads to tumor growth. Thus, gene related diseases such as cancer are highly affected by malfunctions of pathways. In other words, the functions that are affected by malfunctioning pathways are highly related to the type of cancer. Hence, the status of a pathway with respect to its function (performing function properly or not) is important for cancer subtype classification. Since features of genes are statistics of gene expression levels (how many times a gene is activated, measured in context-independent manner), the pathway representation learned from gene interaction likely includes the status of a pathway. Since genes exhibit different characteristics or gene expression levels under different context, it is important to get interaction context-specific representations to properly get the status of pathways.
>
> We added this explanation at ‘Advantage in cancer subtype classification’ in Appendix H.8.2.
>
>
> For the task that reflecting interaction context is important for node representation, we added an example of drug synergy prediction at ‘More Applications’ in Appenidx H.8.2.
>
> ## Question Minor
> We double-checked the link of the code provided in the paper. We checked that our link works for both Chrome or Microsoft Edge browser.

---

> > ### Comment · Reviewer_gjHi · 2024-11-26
> > **Thank you for the rebuttal**
> >
> > Thank you for the rebuttal. However, the contribution compared to WhatsNet [1] is still vague. What are the designs that make the model more efficient in capturing background information compared to WhatsNet? Could you provide empirical and mathematical proof for this?
> >
> > [1] Choe, Minyoung, et al. "Classification of edge-dependent labels of nodes in hypergraphs." Proceedings of the 29th ACM SIGKDD Conference on Knowledge Discovery and Data Mining. 2023.

---

> > > ### Author Response · Authors · 2024-11-28
> > >
> > > We added more explanations on Appendix H.4 to describe how naturality condition guides (although not guarantee) to capture interaction context. We briefly explain it in below.
> > > Since K MLPs are applied to nodes in a hyperedge, it extracts information related to the factors through projection. However, simple projection does not mean that the factor is related to the interaction context. In this section, we will explain how naturality condition guides, although not guarantees, each factor to be related to interaction context. The parameters of factor encoders (K MLPs) are guided to extract interaction context related information during training process. When a specific factor is helpful for performance (predicting labels), the model would try to update parameters of the factor encoder so that the factor information is reflected a lot in hyperedge representation. Since relevance score $\alpha_{i}^{k}$ is multiplied to factor representation to get hyperedge represenation ($\alpha_{i}^{k}h_{e_{i}}^{k}$), the parameters will be updated to increase relevance score $\alpha_{i}^{k}$. Considering that relevance score $\alpha_{i}^{k}$ is calculated by measuring consistency of factor representation (similarity of hyperedge factor representation learned from two different branches), high relevance score means that the representations are similar. Represenations learned from two branches being similar means that it is highly likely that the naturality condition holds, implying that there exists a morphism between nodes in a hyperedge and the hyperedge under specific projection (type) which means the factor is related to the interaction context. In summary, if a specific context (factor) is informative, the parameter of a factor encoder will be updated to the direction of satisfying naturality. Thus, the factor encoder will eventually encode context-related information. When a specific factor is harmful for performance, the opposite would happen. Since naturality condition guides in which direction to update parameters for each factor, although not guaranteed, it is highly likely that each factor contains different context information.
> > >
> > > The naturality condition (relevance score $\alpha_{i}^{k}$ calculated from similarity of two factor representations) is the key component of our method. To check whether this condition helps capturing context, ablation model that does not use this condition needs to be implemented. However, it is difficult to create such variation of our model as there is no way to get $\alpha_{i}^{k}$ if we remove consistency (similarity) calculation. If we remove $\alpha_{i}^{k}$, it cannot be called ‘disentangle-based method’ as there is no way to give different weights to each factor. (The performance of this variation is reported in Appendix E.2. However, transformation matrices will be the same for all hyperedges since it cannot give different weight to factors.) Instead, we can see that the other hypergraph disentangle method (HSDN) that calculates $\alpha_{i}^{k}$ based on factor similarity assumption failed to capture interaction context in Figure 5 and Figure 6.
> > >
> > > For WHATsNet, it is hard to know whether the model captures the context we defined due to the architecture of the model. Since WHATsNet uses multihead attention (MAB(Q,K)), which multiplies two different transformation results to get weights ($QK^{T}$), we cannot know what each transformation (Q,K) would extract during learning process. If we want to guess what kind of information that each transformation extracts, at least one of them must be fixed. The reason we could analyze ours is that we used constraints derived from the analysis of ‘context’ in category theory perspective (the theory that can be used to map syntax and semantics). Also, performing an experiment to check whether WHATsNet captures our context or not is nearly impossible as we need a dataset that has context we defined but does not have context defined by WHATsNet. It is even hard to create a synthetic dataset that does not have context defined by ‘who participates interaction’ as we cannot know whether WHATsNet’s context of each hyperedge affects labels or not. If there exists a slight chance that ‘who participates interaction’ affects the labels (informative to labels), we cannot know whether hyperedge-dependent node representation of WHATsNet is the result of capturing the context we defined or the result of capturing both contexts.

---

> > > ### Author Response · Authors · 2024-11-28
> > >
> > > As a final note, we want to empahsize that the goal of WHATsNet and Natural-HNN is different. WHATsNet tries to capture context generated by ‘who participates interaction’. Thus, every hyperedge (if not duplicate) contains different context. On the other hand, Natural-HNN tries to capture context behind the interaction. It means two different hyperedges can have same interaction context. (For example, one group can have ‘economics’ as their discussion topic but the other group can also have the same discussion topic.) Thus, factors (context) in Natural-HNN can contain semantics that is shared by several hyperedges while hyperedges of WHATsNet have their own context.

---

> > > > ### Comment · Reviewer_gjHi · 2024-12-01
> > > > **Thank you for the reply.**
> > > >
> > > > Dear authors,
> > > >
> > > > Thank you for the reply. However, the novelty compared with WhatsNet still appears vague. As demonstrated in WhatsNet, this work studies how to generate node features conditioned on hyperedges within the classifier, which could be interpreted as a mechanism for reflecting the interaction context of each hyperedge. Consequently, I have decided to maintain my score.

---

### Official Review · Reviewer_CEHf · 2024-11-02

**Soundness:** 2
**Presentation:** 2
**Contribution:** 2
**Rating:** 3
**Confidence:** 5

**Summary:**

This paper presents Natural-HNN, a hyperedge disentangling method designed to identify hyperedge types or interaction contexts within hypergraphs. Natural-HNN incorporates a message-passing layer that leverages node-to-hyperedge interactions, allowing it to capture the underlying structure of hyperedges more effectively. The authors validate their approach on a hypergraph of genetic pathways, showing that Natural-HNN outperforms established baseline methods.

**Strengths:**

- This paper addresses the novel and important problem of hyperedge disentangling
- The authors applied their method on a hypergraph of genetic pathway which is an interesting application.

**Weaknesses:**

- The proposed method lacks novelty.
- The primary issue addressed by this paper—namely, that convolution-based methods cannot perform interaction context-dependent message passing—could be resolved by considering the bipartite representation of a hypergraph and applying a simple message-passing mechanism on it.
- The example provided in Figure 1 is confusing, and none of the datasets are related to it. It would be more compelling if Figure 1 were related to the genetic pathways experiment.
- The authors assert that a drawback of sheaf-based methods is that "there is no guidance that helps the transformation to be related to interaction context." However, they do not explain why this is a drawback or why such guidance is important.
- The introduction is verbose, and portions of it could be more appropriately placed in the related works section.
- Section 3 and Figure 2 are positioned too closely, making the text difficult to follow.
- In Section 5, the authors state, "there is no benchmark dataset verified to contain useful interaction context that is related to the task." If this is the case, the motivation for the proposed method is unclear.
- The model performance shown in Table 1 does not significantly surpass the other baselines.
- Although the authors claim to have an effective hyperedge disentangling method, they need to demonstrate that they have addressed the disentanglement problem in their experiments.

**Questions:**

- What is the motivation behind this work?
- Could this method be applied to standard hypergraph datasets (as presented in previous work), and could you provide results for those datasets as well?

---

> ### Author Response · Authors · 2024-11-23
>
> We revised our paper to clearly deliver the motivation and mechanism of our work : we revised introduction section and changed example in Figure 1 so that it is related to our experiment (genetic pathway). For our comment, we rearranged the sequence of reviewer’s questions and weaknesses considering the flow of our explanation.
>
> ## Question 1
> Simply put, we want to perform relational message passing (similar concept in graph : multiplex GNN, GNN in heterogeneous graph ) even in homogeneous hypergraph that does not have hyperedge type annotations. Although we have revised the introduction section to clearly deliver motivation, we will briefly describe it here. In reality, many interactions have their background (i.e., purpose or motivation of interaction) or conditions of interactions (i.e., condition that interaction occurs). When such context information (background or condition) is accessible during data collection process, it can be expressed as hyperedge types in heterogeneous hypergraph. Relational HNNs or heterogeneous HNNs might be able to reflect context information. However, since context information is not always available during data collection process, the context information is often lost and is expressed as homogeneous hypergraph. We wanted to devise a model that can reflect context (background or condition), or perform relational message passing on hypergraphs, even when it is not explicitly available as hyperedge types.
>
> A genetic pathway  is one example that context information is important but not available (no annotation of hyperedge types). Genes in a pathway interact to perform a biological function (context: purpose, background) and interact under certain conditions such as cell type or age (context: condition). Since a gene exhibits different characteristics such as gene expression levels (amount of gene activity) depending on context, it is important to capture and reflect such contexts during message passing. Note that the cancer subtype classification task (Table 1) uses genetic pathways as a hypergraph.
>
> ## Weakness 2
> The definition of context is not about ‘who participated in the interaction’. Thus, simple convolution-based methods cannot reflect interaction contexts (similar to relational message passing) on hypergraphs (or bipartite graphs).
>
> ## Weakness 4
> Even if sheaf-based methods learn different transformations for every (node, hyperedge) pair, it will need fortune to make all transformations to be related to contexts without any information or guidance about context. We want to note that the context (background or condition) we are interested in is not about ‘who participates in a interaction’. Thus, providing guidance (criterion that transformation matrices must satisfy) will help transformation matrices to be related to interaction context.
>
> ## Weakness 1
> To provide a criterion that transformations (factor projections) must satisfy, we analyzed message passing procedure of HNNs in category theory perspective (we are the first to do so) and found a simple commutativity condition that factor representation and entangled representation must satisfy. This simple criterion allowed our model to capture context even without complex techniques. The simplicity is the strength of our model. Please note that [1] (model for GNN, not HNN) had to rely on several self-supervisions and had to rely on many hyperparameters to disentangle edge types during message passing procedure. In contrast, we could achieve it in HNNs with simple guidance with a small hyperparamter search space that is even comparable to GAT.
>
> [1] Exploring Edge Disentanglement for Node Classification
>
> ## Weakness 7
> Benchmark datasets such as citation networks or co-authorship networks are not verified to contain informative interaction context. In other words, we do not know whether labels are related to context of interaction. If a dataset does not contain meaningful context, it is hard to validate our model with those datasets. The dataset we used for validation (Table 1, cancer subtype classification task) is a case where functions of pathways (context of interaction) is largely related to cancer subtypes. Although it is a dataset that is not a widely used ‘benchmark’, it is a practical application that needs capturing interaction context.

---

> ### Author Response · Authors · 2024-11-23
>
> ## Weakness 9
> Both in our initial and revised submission, we showed that our model could disentangle hyperedge types or interaction context (functionality of pathways) in Section 5.3 (Figure 5, 6).
>
> Figure 5 (cancer dataset) : Since we do not have labels for hyperedge types, we assigned types to hyperedges with a clustering algorithm based on the ground truth functional similarity between pathways. Thus, pathways assigned to the same type have similar function (context). To check whether functions (context) are well disentangled, we compared context similarity between hyperedge types (heatmap). As can be seen in Figure 5, the context similarity captured by ours (Natural-HNN) is quite similar to that of the ground truth, while it is not the case for HSDN. Thus, this experiment shows that our model could capture interaction context (disentanglement).
>
> Figure 6 (synthetic dataset): We created a synthetic heterogeneous hypergraph with 8 hyperedge types. We trained Natural-HNN, HSDN and a sheaf-based model without giving any hyperedge type information. We drew a heatmap that shows the transformation matrix similarity between hyperedge types. (Since factor-related information is extracted through transformation, similar transformation matrices means that they extracted a similar factor or context). We showed that Natural-HNN had a relatively stronger diagonal-like heatmap when compared to other models. It shows that Natural-HNN could inherently capture hyperedge types even if hyperedge type information was not provided.
>
>
> ## Weakness 8
> We believe that for BRCA, STAD, SARC and HNSC, the performance improvement is not small. Our model showed 1.7% ~ 4.7% improvement when compared to the best model among baselines.
> In Section 5.2, we provided prior studies explaining that subtypes (labels) of KIPAN and NSCLC have distinguishable characteristics. It means other models that do not reflect interaction context can also easily classify subtypes. That’s the reason why many models in those datasets have similar performance. Since a small improvement of performance in those datasets originates from the dataset characteristic, we believe it is not a drawback of our model. Finally, we want to note that our model achieved the state-of-the-art performance for 6 datasets out of 8 datasets. Stably outperforming baselines shows that our model is superior to existing baselines.
>
> ## Weakness 5
> In our revised submission, we removed the first paragraph of our initial submission (It was about GNN, not HNN). We think explanations about previous HNNs (fourth paragraph in revised version) is necessary to explain previous models were not designed to capture interaction context.
>
> ## Weakness 3
> We changed Figure 1 in the introduction section to the one related to genetic pathways. We also changed explanations related to Figure 1 in introduction section.
>
> ## Weakness 6
> We reflected your concerns in our revised submission. We added space between Figure 2 and text of Section 3.
>
>
> ## Question 2
> Both in our initial and revised submission, the result for benchmark datasets is available in Appendix C. Part of the experiments in Appendix E (ablation) and F.3 are also related to benchmark datasets.

---

> > ### Comment · Reviewer_CEHf · 2024-12-02
> >
> > Thank you for your rebuttal. Your work shows potential; however, in its current form, it is not yet ready for publication at ICLR.
> >
> > Regarding your rebuttal, I found it challenging to follow your comments. The points are not presented in a clear and organized manner, and it is difficult to discern which weaknesses or questions you are addressing. At this time, I will maintain my current score.

---

### Official Review · Reviewer_8TCo · 2024-11-03

**Soundness:** 3
**Presentation:** 3
**Contribution:** 3
**Rating:** 8
**Confidence:** 3

**Summary:**

This paper presents a hyperedge disentangling method, called Natural-HNN, that captures the inherent hyperedge types or the interaction context of hyperedge, based on the naturality condition in category theory.

**Strengths:**

1. The authors have identified an interesting issue in hypergraph representations and the solution that guidance for disentanglement hyperedge is pretty novel.
2. The experiments are comprehensive, including 8 clinical datasets, 8 benchmark datasets, and a synthetic dataset.
3. This paper is well-organized and easy to follow.

**Weaknesses:**

1. I am  a little bit concern with the overfitting issue, as it contains $k$ MLP for each layer. Does the parameters shared across layers?
2. How does the heterophilic level for the clinial datasets look like? From the model design, it may works well for heterophilic hypergraphs. Can you also report performance for heterophilic hypergraphs, e.g. Congress, Senate, and Walmart in EDHNN [1]?

[1] Equivariant Hypergraph Diffusion Neural Operators. ICLR'23

**Questions:**

Please see the weakness.

---

> ### Author Response · Authors · 2024-11-23
>
> ## Weakness 1
> The parameters of $K$ MLPs are not shared across layers. However, it will not overfit as the number of parameters will be the same regardless of the number of factors $K$. (number of parameters does not increase with $K$) When $d$ is the dimension of hidden representation, each factor representation will have dimension $d \over K$.  Actually, using $K$ MLPs with each output vector dimension $d \over K$ is the same as applying single MLP with output vector dimension $d$ and dividing the vector into $K$ vectors with dimension $d \over K$ (ex: torch.chunk() ). Since most HNNs use at least one MLP per layer with output dimension size $d$, we think Natural-HNN does not use many parameters to overfit.
>
> ## Weakness 2
> Since clinical dataset (cancer subtype dataset) was a hypergraph classification task, we do not have labels to the nodes. It means we cannot measure the heterophilic level of a hypergraph.
>
> When we perform an experiment on heterophily datasets, we have the following results. We got 68.447 ± 2.970 (house), 61.249 ± 3.258 (senate) and 63.129 ± 0.347 (walmart). The result for congress dataset is under experiment. We had reproducibility issue in house dataset. We got 56.087 ± 2.485 for HGNN in house dataset, which is far from reported performance 61.39 ± 2.96.
>
> We calculated the average of hyperedge homophily ratio by the following:
> ${\sum_{e \in \mathcal{E}} {{\sum_{i \in labels} c_{i,e}^2} \over { \vert e \vert }^{2}}} / {\vert \mathcal{E} \vert}$
>
> where $c_{i,e}$ denotes the number of nodes in hyperedge $e$ with label $i$.
> It approximately calculates the average of ‘ratio of homophilious edge within a hyperedge when a hyperedge is converted to a fully connected graph’. We got the following result :
> 0.5206 (house), 0.5076 (senate) and 0.6925(walmart).
>
> We can see that Natural-HNN had largely increased performance on senate and house dataset when compared to HGNN. (As described before, HGNN had 56.087 ± 2.485 performance in our measurement.) We guess that capturing hyperedge context was helpful for a dataset with low hyperedge homophily ratio.  However, when hyperedge homophily ratio was high (almost 0.7), our model had relatively small improvement over HGNN.
>
> When compared to ED-HNN, Natural-HNN had lower performance. Since Natural-HNN uses mean aggregation during node-to-hyperedge propagation (similar to HGNN), Natural-HNN cannot give different importance to nodes in each hyperedge. Under heterophilc setting, giving different importance can be important to give more weight to homophilious neighbors. Since learning node importance without similarity assumption (not relying on attention mechanism) is not trivial, we leave it as our future work.

---

### Meta-Review · Area_Chair_3nQb · 2024-12-20

**Metareview:**

The paper presents Natural-HNN, a hyperedge disentangling method that captures the inherent hyperedge types or the interaction context of hyperedge. The problem studied in this work is important. The comprehensive experiments show that Natural-HNN outperforms baseline methods. The paper presentation is easy to follow. Reviewers raised several key issues, including the incremental novelty of Natural-HNN, the marginal improvement of performance, missing important related works, and so on. After discussion, reviewers reach a consensus of rejection.

**Additional Comments On Reviewer Discussion:**

After discussion, reviewers reach a consensus of rejection. The reviewer who gave the positive score agreed to the review of other reviewers.

---

### Decision · Program_Chairs · 2025-01-22

Reject